# mTORC2-driven chromatin cGAS mediates chemoresistance through epigenetic reprogramming in colorectal cancer

Guoqing Lv[1,2,10], Qian Wang [3,10], Lin Lin [4,10], Qiao Ye[5,10], Xi Li[1], Qian Zhou [6], Xiangzhen Kong[7], Hongxia Deng [8], Fuping You[9], Hebing Chen [4], Song Wu [3] ✉ & Lin Yuan [1,2,10] ✉

Cyclic GMP–AMP synthase (cGAS), a cytosolic DNA sensor that initiates a STING-dependent innate immune response, binds tightly to chromatin, where its catalytic activity is inhibited; however, mechanisms underlying cGAS recruitment to chromatin and functions of chromatin-bound cGAS (ccGAS) remain unclear. Here we show that mTORC2-mediated phosphorylation of human cGAS serine 37 promotes its chromatin localization in colorectal cancer cells, regulating cell growth and drug resistance independently of STING. We discovered that ccGAS recruits the SWI/SNF complex at specific chromatin regions, modifying expression of genes linked to glutaminolysis and DNA replication. Although ccGAS depletion inhibited cell growth, it induced chemoresistance to fluorouracil treatment in vitro and in vivo. Moreover, blocking kidney-type glutaminase, a downstream ccGAS target, overcame chemoresistance caused by ccGAS loss. Thus, ccGAS coordinates colorectal cancer plasticity and acquired chemoresistance through epigenetic patterning. Targeting both mTORC2–ccGAS and glutaminase provides a promising strategy to eliminate quiescent resistant cancer cells.

Cyclic GMP–AMP synthase (cGAS) catalytically forms the molecule cyclic GMP–AMP (cGAMP) upon sensing cytosolic double-stranded (ds) DNA from invading pathogens[1] or damage within the host genome[2,3], activating the cGAS–STING pathway[4]. This pathway plays an established role in immune surveillance; however, cancer cells commonly dampen STING activity while retaining cGAS's sentinel function[5,6], suggesting that cGAS may perform additional stealthy roles independent of STING. cGAS has been reported to be recruited to dsDNA breaks through interacting with poly(ADP-ribosyl)ated PARP1 (ref. 7), thereby suppressing homologous recombination repair[7] and accelerating genomic instability[8]. However, cGAS has also been shown to slow cell proliferation and reduce genomic instability by binding to DNA and interacting with replication fork proteins[9]. Recent studies found that the N terminus of cGAS is critical for sensing nuclear chromatin, though this is blocked by hyperphosphorylation during mitosis[10]. Notably, cGAS does not directly contact nucleosomal DNA but tightly anchors to

[1]Institute of Biomedical Sciences, Peking University Shenzhen Hospital, Shenzhen, China. [2]Department of Biochemistry, University of Texas Southwestern Medical Center, Dallas, TX, USA. [3]Department of Urology, The Third Affiliated Hospital & South China Hospital of Shenzhen University, Shenzhen, China. [4]Institute of Health Service and Transfusion Medicine, Beijing, China. [5]Clinical Medicine Laboratory, Air Force Medical Center, Beijing, China. [6]Department of Computer Science, City University of Hong Kong, Hong Kong, China. [7]Department of Pharmacy, The First Affiliated Hospital of Zhengzhou University, Zhengzhou, China. [8]Department of Otorhinolaryngology Head and Neck Surgery, Ningbo Medical Center Lihuili Hospital, Ningbo University, Ningbo, China. [9]Institute of Systems Biomedicine, Department of Pathology, School of Basic Medical Sciences, Peking University Health Science Center, Beijing, China. [10]These authors contributed equally: Guoqing Lv, Qian Wang, Lin Lin, Qiao Ye, Lin Yuan. ✉e-mail: wusong@szu.edu.cn; yuan_lin@bjmu.edu.cn

the 'acidic patch' on histone 2A–histone 2B instead[11–15]. This chromatin tethering blocks the formation of active cGAS dimers[16] and prevents autoreactivity to self-DNA. Nonetheless, mechanisms underlying cGAS recruitment to chromatin and its chromatin functions remain elusive.

The mechanistic target of rapamycin (mTOR) pathway integrates extracellular signals to regulate cellular responses to environmental changes[17]. mTOR forms two complexes, mTORC1 and mTORC2. While mTORC1 is well studied, the functions of mTORC2 are less clear, although aberrant activity contributes to several cancer types[18–21], implicating oncogenic roles for mTORC2 (ref. 22). Recent evidence links mTOR to drug resistance[23–25]. Inhibiting mTOR induces a dormant state in mammalian embryos and a reversible diapause-like state in cancer cells[26], conferring tolerance against chemotherapy[24,27]. Such mTOR blockade protects against chemotherapeutic agents in colorectal cancer and leukaemia[28–30]; however, mechanisms linking mTOR inhibition to acquired chemoresistance remain unclear. Here, through high-throughput compound screening, we discover that mTOR inhibition disrupts cGAS–chromatin tethering. mTORC2 directly phosphorylates cGAS at serine 37, promoting chromatin anchoring. Of note, ccGAS knockdown inhibits colorectal cancer cell growth under normal conditions but induces acquired resistance to fluorouracil, a front-line therapeutic. Mechanistically, ccGAS recruits SWI/SNF complexes at specific gene loci to regulate DNA replication and glutaminolysis genes, and inhibiting the downstream kidney-type glutaminase (KGA) overcomes resistance. Our study provides mechanistic insight into mTOR inhibition-mediated chemoresistance and implies targeting the mTORC2–ccGAS–KGA axis could improve cancer therapy.

## Results

### High-throughput screening identifies PI3K–mTOR pathway regulation of cGAS–chromatin localization

To explore how cGAS localization is regulated, we analysed its subnuclear distribution in cancer cells. We found that cGAS localized constitutively to the nucleus independent of cell cycle timing or nucleotidyltransferase activity (Extended Data Fig. 1a–c). When fractionating HCT116 cell lysates, we found cGAS to be predominantly chromatin-bound (Fig. 1a). To identify potential regulatory molecules, we performed high-throughput compound screening in HCT116 cells to monitor cGAS–chromatin interaction using bioluminescence resonance energy transfer (BRET) technology[31] (Fig. 1b). In a library of 8,326 compounds, BEZ-235, GDC-0941 and KU-0063794 significantly inhibited this interaction (Fig. 1c), indicating involvement of the PI3K–mTOR pathway. Stable isotope labelling identified cGAS and lymphoid-specific helicase HELLS most significantly reduced on chromatin by the PI3Ki GDC-0941 treatment (Extended Data Fig. 1d). Unlike HELLS fluctuated on chromatin after GDC-0941 exposure, cGAS levels on chromatin consistently decreased with increasing treatment time (Fig. 1d and Extended Data Fig. 1e), suggesting that cGAS is a specific PI3K–mTOR chromatin target.

To determine the specific components driving cGAS–chromatin localization, we treated HCT116 cells with PI3K–mTOR pathway inhibitors (Extended Data Fig. 2a). Immunofluorescence analysis showed that inhibition of PI3K and mTOR, but not AKT, mTORC1 or S6K, reduced cGAS–chromatin association (Fig. 1e and Extended Data Fig. 2b). This was confirmed using the extended bioluminescence resonance energy transfer (eBRET2) assay (Fig. 1f). We then used short hairpin RNA (shRNA) to knockdown RAPTOR and RICTOR, essential subunits of mTORC1 and mTORC2, respectively (Extended Data Fig. 2c). RICTOR knockdown, but not RAPTOR, inhibited cGAS–chromatin localization (Fig. 2a and Extended Data Fig. 2d). BRET (Fig. 2b) and chromatin fractionation (Fig. 2c,d) assays also showed that mTOR and RICTOR knockdown, but not RAPTOR, disrupted the cGAS–chromatin interaction. Moreover, the interaction between the enzyme-inactive S213D cGAS mutant and chromatin was regulated by mTORC2 (Fig. 2b,d). Collectively, these data demonstrate that cGAS–chromatin association is dependent on mTORC2, but not its nucleotidyltransferase activity.

### mTORC2-induced phosphorylation at serine 37 promotes cGAS–chromatin localization

We investigated whether cGAS is a direct target of mTORC2 phosphorylation, given mTORC2's ability to phosphorylate substrates. Co-immunoprecipitation (Co-IP) (Extended Data Fig. 2e,f) and BRET assay (Extended Data Fig. 2g) detected interactions between cGAS and SIN1, an essential mTORC2 component for substrate recognition[22]. Using an in vitro cell-free phosphorylation system, we found that activated mTORC2 directly phosphorylates cGAS at serine 37 as demonstrated by mass spectrometry (MS) (Fig. 2e, Extended Data Fig. 2h and Supplementary Table 1). In contrast, inactive mTORC2 or mTORC1 did not mediate this phosphorylation (Extended Data Figs. 2i and 3a), validating serine 37 as the specific mTORC2 target site. Consistently, serine 37 phosphorylation of cGAS (p-Ser[37]-cGAS) were reduced in RICTOR-deficient HCT116 cells and rescued by RICTOR reintroduction (Fig. 2f,g and Extended Data Fig. 3b–d). Moreover, p-Ser[37]-cGAS levels, not total cGAS, consistently decreased over time with GDC-0941 treatment (Extended Data Fig. 3e).

Serine 37 is located within the N-terminal nuclear localization signal (NLS) of human cGAS[7], which contains a common consensus motif for mTOR phosphorylation[19] (Extended Data Fig. 3f). A second NLS in the NTase core region[7] contains potential phosphorylation sites for both AKT1 and mTOR. In vitro phosphorylation assays showed that AKT1 phosphorylated serine 305 (ref. 32), whereas mTORC2 did not (Extended Data Fig. 3f,g). BRET assays found that cGAS mutants S305D and S305A did not bind chromatin (Extended Data Fig. 4a). Additional mutant experiments demonstrated that cGAS nuclear localization is independent of serine 305 phosphorylation, but serine 37 phosphorylation promotes nuclear localization of mutants (Extended Data Fig. 4b). These results indicate that mTORC2 specifically phosphorylates the N-terminal NLS at serine 37, which may facilitate human cGAS–chromatin localization.

Immunofluorescence analysis revealed that p-Ser[37]-cGAS co-localizes with histone H2A in the nucleus of HCT116 cells (Extended Data Fig. 4c). MS analysis showed that chromatin-bound but not extranuclear cGAS was phosphorylated at serine 37 (Extended Data Fig. 4d). Immunofluorescence further showed that the S37D phosphomimetic mutant was exclusively localized to chromatin, whereas the non-phosphorylatable S37A mutant was cytoplasmic (Fig. 2h and Extended Data Fig. 4e), despite both retaining chromatin binding capacity during mitosis (Extended Data Fig. 4f). Chromatin fractionation confirmed that the S37D mutation promoted chromatin association, whereas S37A disrupted it (Fig. 2i). In addition, inhibiting mTORC2 by JR-AB2-011 gradually reduced Ser37 phosphorylation and cGAS–chromatin levels over time (Fig. 2j and Extended Data Fig. 4g,h). These results indicate that mTORC2-mediated serine 37 phosphorylation is essential for cGAS–chromatin localization.

### ccGAS recruits the SWI/SNF complex at specific chromatin regions to regulate gene expression

To explore ccGAS functions, we characterized its interacting proteins (Extended Data Fig. 5a,b and Supplementary Table 2). In extranuclear fractions, cGAS interacts with ribosomal subunits (Extended Data Fig. 5c). ccGAS recruits several components of the SWI/SNF chromatin remodelling complex, including SMARCC2, SMARCA4, PBRM1, ARID1A, SMARCA5, SMARCD1 and UCHL5 (Fig. 3a and Extended Data Fig. 5d–f). SWI/SNF alters nucleosome positioning and chromatin accessibility to control gene expression[33]. The nuclease benzonase was included in all buffers to minimize post-lysis DNA binding by cGAS, then Co-IP validated interactions between cGAS and SWI/SNF components such as ARID1A, SMARCA4 and SMARCC2 in colorectal cancer cells (Fig. 3b and Extended Data Fig. 5g). These results suggest that ccGAS recruits the SWI/SNF complex to regulate gene expression.

To investigate whether ccGAS binds preferentially to certain genomic regions, we performed CUT&Tag[34] assays in replicates (Extended Data Fig. 6a and Supplementary Table 3). Analysis of peaks

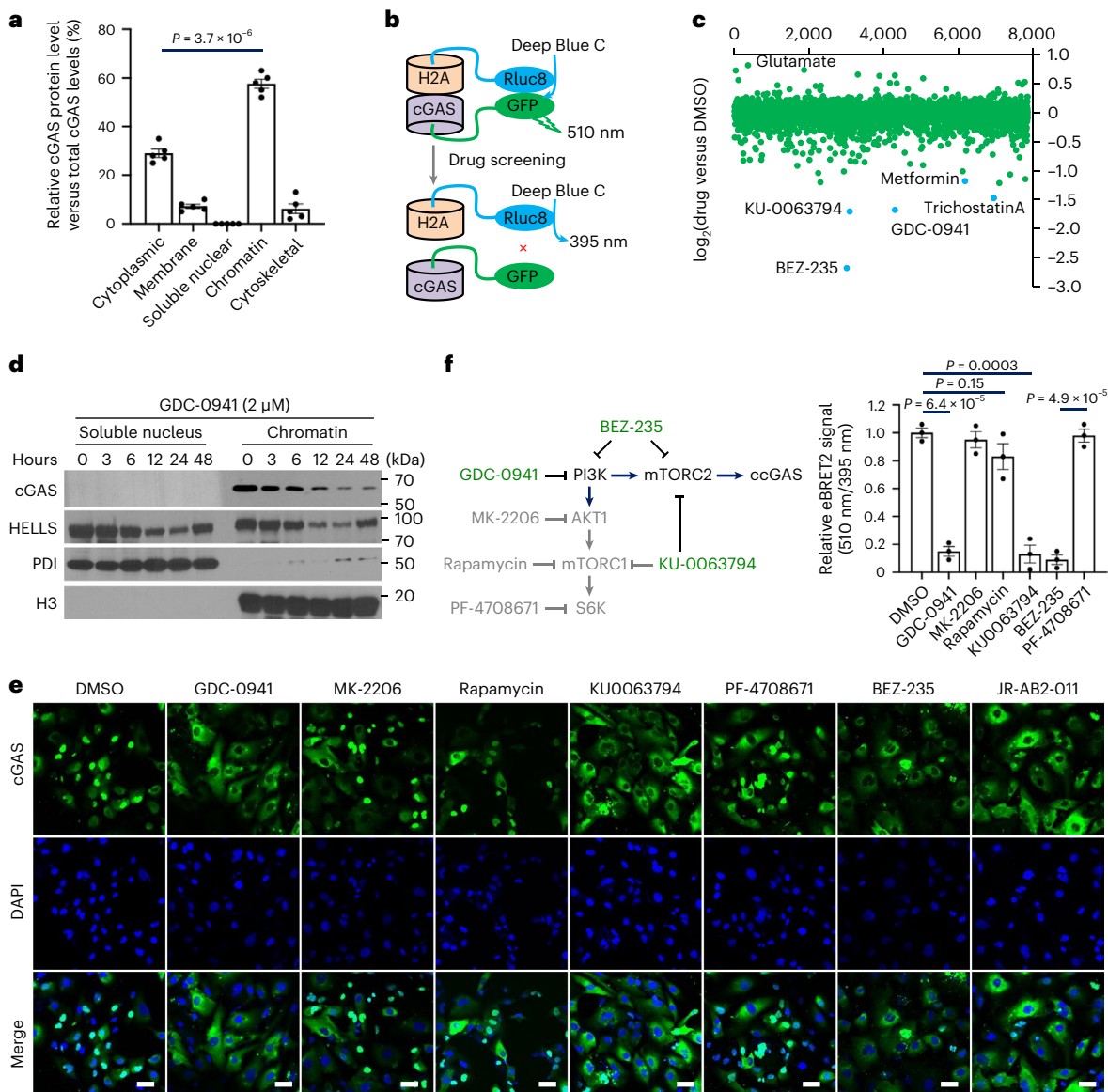

**Fig. 1 | High-throughput screening identifies PI3K–mTOR pathway regulation of cGAS–chromatin localization. a**, Cell fractionation and ELISA were used to quantify cGAS protein levels in subcellular fractions of HCT116 cells. $n = 5$ independent experiments per group. **b**, A chromatin cGAS biosensor composed of pcDNA3.1(+)-GFP2-cGAS and pcDNA3.1(+)-Rluc8-H2A was generated. The eBRET2 signal ratios indicate cGAS–H2A interactions. **c**, The biosensor underwent high-throughput drug screening in HCT116 cells. Ratios of eBRET2 signals with drugs (Bioactive Compound Library) versus dimethylsulfoxide (DMSO) indicate effects on cGAS–chromatin interactions. **d**, HCT116 cells were treated with GDC-0941 for the times indicated, then soluble nuclear and chromatin fractions were analysed by immunoblot (IB) with the antibodies shown. **e**, Immunofluorescence analysis of cGAS subcellular distribution in HCT116 cells treated with PI3K–mTOR inhibitors (2 µM, 12 h). MK-2206, a highly selective AKT inhibitor; rapamycin, an allosteric mTORC1 inhibitor; PF-4708671, a p70 ribosomal S6 kinase inhibitor; Scale bars, 20 µm. **f**, HCT116 cells transfected with the biosensor were treated with 2 µM of the indicated compounds for 12 h, then eBRET2 signals were measured. Ratios of compounds versus control indicate effects on cGAS–chromatin interactions. $n = 3$ independent experiments per group. Data are shown as mean ± s.e.m. Unpaired two-tailed $t$-test. Experiments were repeated three times (**d**, **e**) or twice (**c**) with similar results. Numerical data and unprocessed blots are available as source data.

near genes showed high correlation between replicates ($r = 0.79$) (Extended Data Fig. 6b–d) and ccGAS enrichment near essential gene transcription start sites in purine deoxyribonucleotide (G/A)-rich regions (Extended Data Fig. 6e–g). KEGG analysis showed that ccGAS-bound genes enriched for the PI3K–AKT (insulin) pathway (Extended Data Fig. 6h). Sequencing data located 16.6% of ccGAS peaks at gene promoters (Fig. 3c and Extended Data Fig. 6i). KEGG analysis of promoter-bound genes linked them to cell cycle regulation and amino acid metabolism (Fig. 3d and Extended Data Fig. 6j).

To characterize ccGAS-regulated proteins, we used TMT labelling and quantitative proteomics in cGAS-knockdown HCT116 cells (Extended Data Fig. 7a,b and Supplementary Table 4). Among 6,523 identified proteins, 53 decreased significantly (fold < 0.5) in the knockdowns (Extended Data Fig. 7c,d). Gene Ontology analysis showed that these proteins relate to DNA replication (Fig. 3e and Extended Data Fig. 7e). Levels of components of the CDC45–MCM–GINS (CMG) helicase complex, including MCM2-7, GINS3 and GINS4, were decreased in cGAS-knockdown cells (Fig. 3f and Extended Data Fig. 8a,b). Replication fork proteins such as WDHD1, UHRF1, POLA2, PRIM1, PRIM2 and SMARCAL1 also decreased (Fig. 3f and Extended Data Fig. 7c,e). Collectively, these results suggest that ccGAS recruits SWI/SNF at specific chromatin regions to positively regulate DNA replication protein expression.

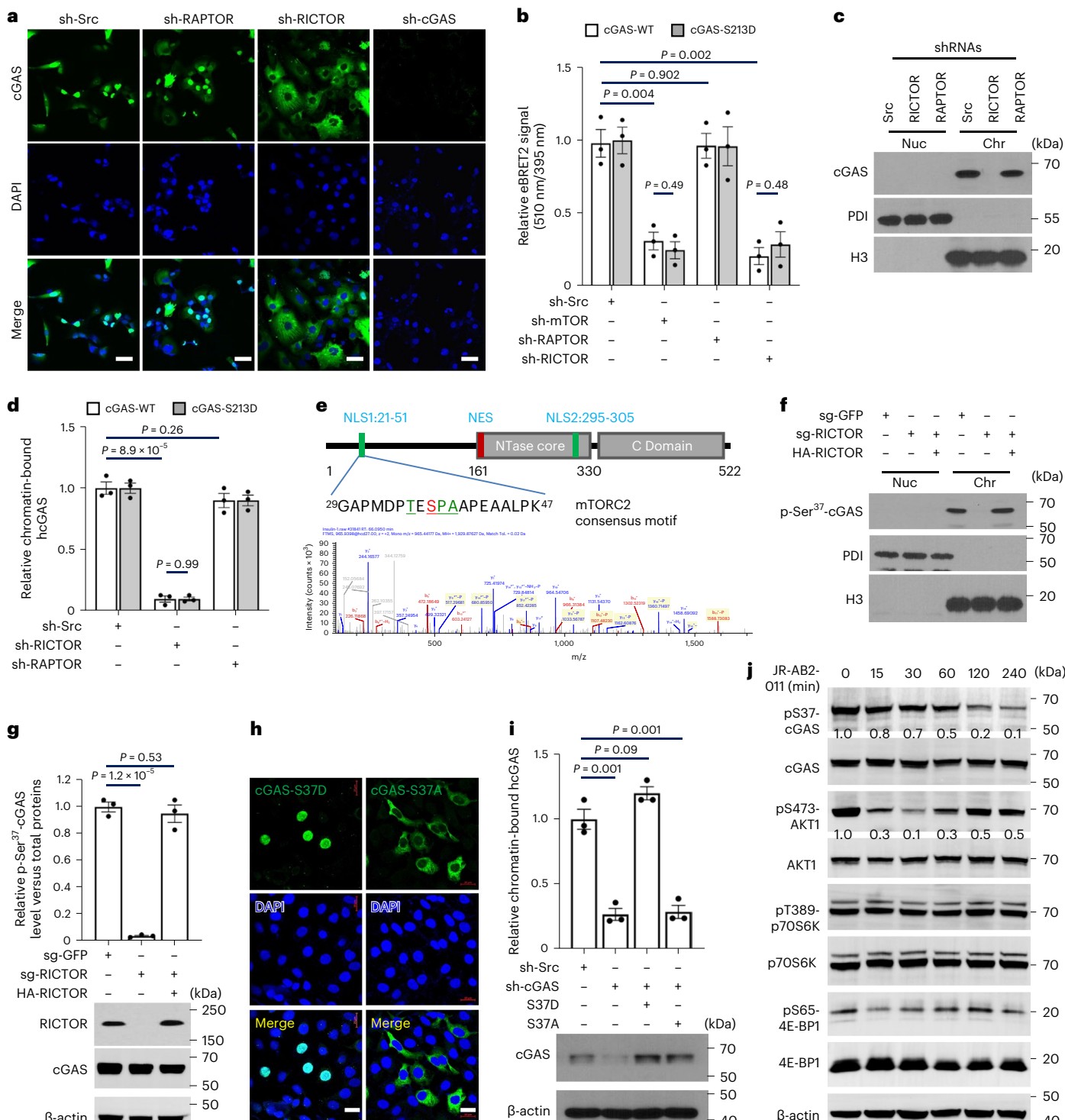

**Fig. 2 | mTORC2-mediated phosphorylation of cGAS at serine 37 promotes its chromatin localization. a**, Immunofluorescence analysis of HCT116 cells lentivirally infected with shRNA targeting the indicated genes or a scramble control (sh-Src). Nuclei were labelled with 4,6-diamidino-2-phenylindole (DAPI, blue). Scale bars, 20 μm. **b**, eBRET2 signal in HCT116 cells stably expressing the indicated shRNAs, reporting on cGAS–H2A interactions. **c**, Soluble nuclear (Nuc) and chromatin (Chr) fractions from HCT116 cells stably expressing the indicated shRNAs were collected for IB analysis with the antibodies shown. **d**, ELISA quantification of cGAS protein levels in chromatin fractions isolated using a Chromatin Extraction Kit from HCT116 cells stably expressing indicated shRNAs. hcGAS, human cGAS. **e**, In vitro LC–MS/MS phosphorylation assays identified recombinant cGAS serine 37 as a direct phosphorylation site of mTORC2.

**f**, RICTOR knockout HCT116 cells were fractionated and immunoblotted. **g**, ELISA quantification of pSer37-cGAS levels in RICTOR knockout HCT116 cells stably expressing HA-RICTOR. **h**, Localization of mutant cGAS was assessed by immunofluorescence in cGAS-knockout HCT116 cells. Scale bars, 20 μm. **i**, ELISA quantification of cGAS protein levels in chromatin fractions from cGAS-knockdown HCT116 cells expressing shRNA-resistant cGAS mutants. **j**, STING$^{-/-}$ HCT116 cells were treated with 10 μM JR-AB2-011 for the times indicated, then whole-cell lysates (WCLs) were analysed by IB with the antibodies shown. Data are shown as mean ± s.e.m. from three independent experiments (**b**, **d**, **g**, **i**). Unpaired two-tailed $t$-test. Experiments were repeated three times (**a**, **c**, **f**) with similar results. Numerical data and unprocessed blots are available as source data. WT, wild type.

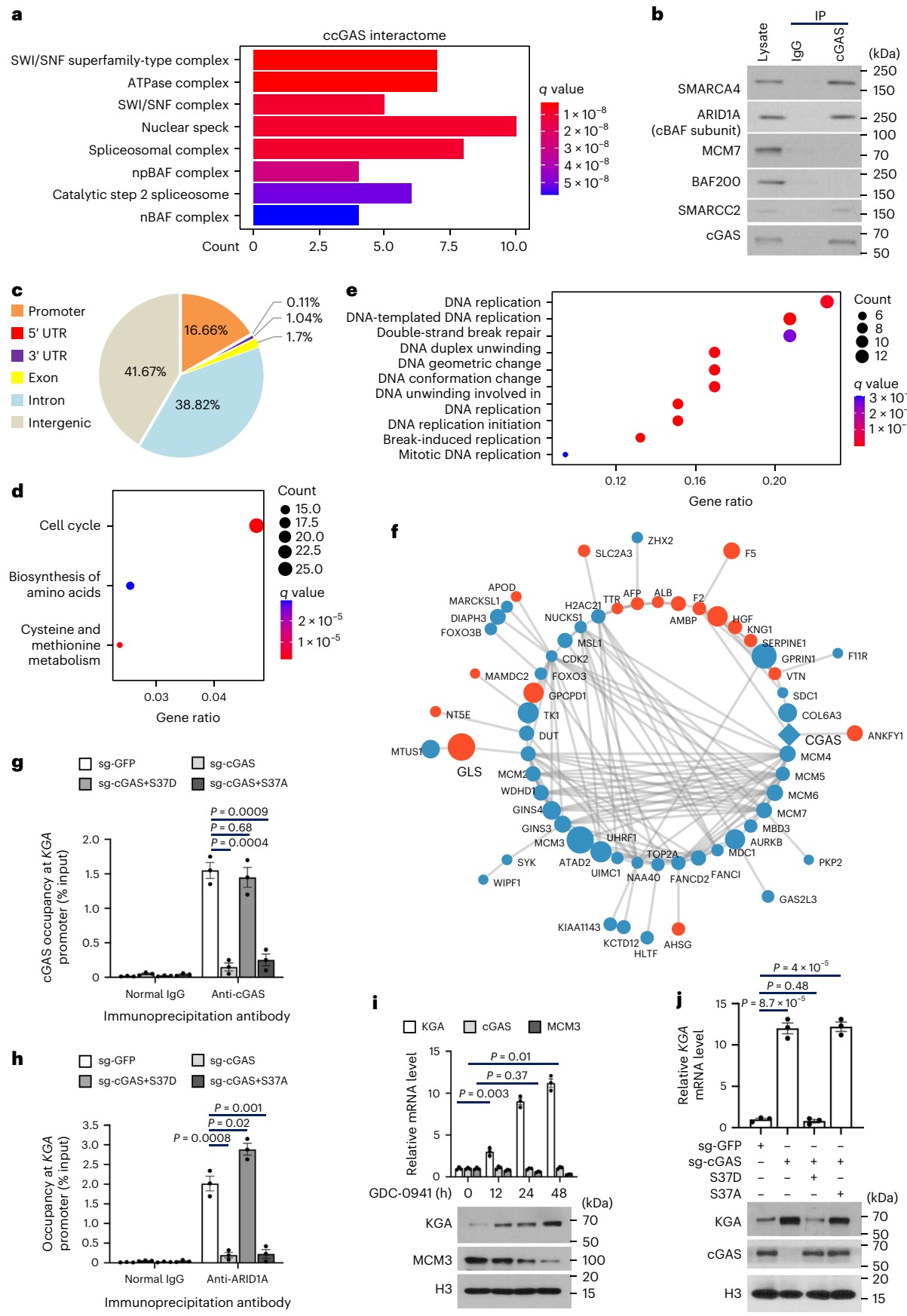

**Fig. 3 | ccGAS recruits SWI/SNF to regulate genes involved in DNA replication and glutaminolysis. a**, Functional enrichment (cell component) analysis of 33 unique cGAS-interacting intranuclear proteins. **b**, IB analysis of immunoprecipitation (IP) and WCLs derived from HCT116 cells. **c**, The cGAS CUT&Tag peaks were annotated by ChIPseeker, and 16.64% were found to be located on gene promoters. **d**, KEGG pathway analysis of promoter-localized cGAS peaks showed that these genes were closely related to cell cycle and amino acid metabolism. **e**, Functional enrichment analysis of 53 downregulated proteins under cGAS knockdown in HCT116 cells. **f**, Eighty-one proteins significantly altered in abundance under cGAS knockdown, of which 61 proteins, 18 upregulated (orange) and 43 downregulated (blue), were annotated to 142 reliable protein–protein interactions (grey line) based on databases such

as BioGrid and StringDB. Node size indicates the degree of altered abundance. **g**, ChIP–qPCR analysis of cGAS occupancy at the *KGA* promoter in HCT116 cells lentivirally infected with the indicated shRNAs. **h**, ChIP–qPCR analysis of ARID1A occupancy at the *KGA* promoter in cGAS-knockout HCT116 cells stably expressing control or cGAS mutants. **i**, HCT116 cells were treated with 2 μM GDC-0941 for the indicated times, mRNA levels of indicated genes were quantified by qPCR analysis and normalized to ACTB control and WCLs were analysed by IB. **j**, cGAS mutants were transfected into cGAS-knockout HCT116 cells and *KGA* mRNA levels and WCLs were analysed. Data are shown as mean ± s.e.m. from three independent experiments (**g**–**j**). Unpaired two-tailed *t*-test. Experiments were repeated three times (**b**) with similar results. Numerical data and unprocessed blots are available as source data.

## ccGAS negatively regulates KGA expression in colorectal cancer cells

In addition to characterizing proteins positively regulated by ccGAS, the proteomic analysis identified 28 proteins significantly upregulated (fold > 2) in cGAS-knockdown cells (Fig. 2f and Extended Data Fig. 7c). In addition to proteins negatively regulating proteolysis, such as SERPINE1, F5, RP5-1022P6.2, APOH, HGF, HMGA2 and F2 (Fig. 3f), these included KGA, an isoform of GLS1 (ref. 28), which increased over threefold (Fig. 3f and Extended Data Fig. 8c). Compared to controls, KGA messenger RNA levels and glutaminase activity were significantly increased in cGAS-knockdown colorectal cancer cells (Extended Data Fig. 8d–f). Re-expressing wild-type cGAS reduced these elevated levels and activity (Extended Data Fig. 8d–f). These results suggest that ccGAS negatively regulates KGA expression.

Consistent with CUT&Tag data showing ccGAS interaction with the *KGA* promoter (Extended Data Fig. 6i), chromatin immunoprecipitation (ChIP) assays found that ccGAS localized to the *KGA* promoter region in colorectal cancer cells (Fig. 3g). ccGAS localization decreased with mTOR or RICTOR knockdown but not RAPTOR knockdown (Fig. 3g). STING knockout or cGAS deactivation did not impact ccGAS localization (Extended Data Fig. 8g). Compared to wild-type cGAS, the S37D mutant, but not S37A, maintained *KGA* promoter localization (Fig. 3h and Extended Data Fig. 8h). These results validate the cGAS genomic binding data and suggest that *KGA* transcription is strictly regulated by ccGAS.

Inhibiting PI3K by GDC-0941 increased both KGA protein and mRNA levels in a time-dependent manner correlated with reduced ccGAS protein levels (Fig. 3i). KGA mRNA and protein levels also significantly elevated in cGAS-knockout cells compared with controls, but re-expressing wild-type or S213D mutant cGAS rescued this (Extended Data Fig. 8i), indicating cGAS enzymatic activity is dispensable for regulating KGA. Overexpressing cGAS mutant S37D, but not S37A, reduced elevated KGA mRNA and protein levels in cGAS-knockout cells (Fig. 3j). Additionally, Seahorse analysis showed that cGAS knockout increased glutamine-dependent oxygen consumption, an effect rescued by S37D but not S37A mutant expression (Extended Data Fig. 8j). Together, these results demonstrate that ccGAS depletion induces KGA expression and promotes glutaminolysis.

## ccGAS depletion induces a diapause-like state in colorectal cancer cells

Given cancer cells' addiction to glutaminolysis and DNA replication[35], we infer that ccGAS was hypothesized to be essential for cancer progression. cGAS knockdown induced G1/S-phase arrest in HCT116 cells after nocodazole synchronization (Extended Data Fig. 9a,b). Overexpressing shRNA-resistant S37D or S213D cGAS rescued this arrest, whereas S37A mutant or STING activator (ADU-S100) supplementation did not (Fig. 4a and Extended Data Fig. 9c). Plate cloning and cell counting assays confirmed that cGAS knockdown inhibited HCT116 cell growth, rescued by S37D but not S37A overexpression (Fig. 4b–d). Therefore, these results demonstrated that ccGAS depletion induces colorectal cancer cell cycle arrest.

Cells with chromatin-localized cGAS exhibited a small, fusiform 'epithelial-like' morphology (Fig. 4e and Extended Data Fig. 9d). In contrast, those with non-chromatin cGAS seemed large and round with a 'diapause-like' morphology characterized by cell cycle arrest and reduced proliferation[36–40] (Fig. 4e and Extended Data Fig. 9d). cGAS-knockout HCT116 cells expressing S37D-mutated cGAS displayed mesenchymal characteristics, whereas the S37A mutant adopted a diapause-like morphology (Extended Data Fig. 9e,f). Consistently, inhibiting PI3K–mTORC2 or RICTOR knockdown induced a diapause morphology, whereas inhibiting the AKT1–mTORC1–S6K axis or RAPTOR knockdown did not change morphology (Figs. 1f and 2a). Collectively, these results indicate that inhibiting the mTORC2–cGAS axis induces diapause-like plasticity in colorectal cancer cells.

## ccGAS depletion restricts tumour growth and induces chemoresistance

While diapause-like cancer cells exhibit reduced proliferation, they often acquire drug resistance properties[36–42]. We investigated whether ccGAS is required for chemosensitivity in colorectal cancer. Fluorouracil (5-FU) is a widely used chemotherapeutic for colorectal cancer[43,44]. cGAS knockdown inhibited HCT116 cell proliferation under normal growth conditions but desensitized cells to 5-FU-induced death (Fig. 4f). Re-expressing wild-type cGAS resensitized cells to 5-FU, an effect blocked by inhibiting mTORC2 with JR-AB2-011 (Fig. 4g). Expressing S37D or S213D mutants also resensitized cells to 5-FU, whereas

**Fig. 4 | ccGAS depletion induces a diapause-like state and chemoresistance in colorectal cancer cells. a**, Quantification data of cell cycle distribution for HCT116 cells are presented. shRNA-resistant cGAS mutants were transfected into HCT116 cells stably expressing cGAS shRNA. ADU-S100, an activator of STING, 10 μM. **b**, Representative images of colony formation assays showed the proliferative capacity of HCT116 cells stably expressing indicated cGAS mutants. **c**, Colony formation assays were quantified in HCT116 cells stably expressing indicated cGAS mutants. **d**, HCT116 cells stably expressing indicated cGAS mutants were measured for cell proliferation. **e**, The relative size of cGAS-knockout HCT116 cells with cGAS mutants restoration was calculated. *n* = 2 × 10⁷ cells per group. **f**, HCT116 cells stably expressing cGAS shRNA were treated with the indicated concentrations of 5-FU and cell viability was assessed after 24 h. **g**, cGAS-knockdown HCT116 cells stably transfected cGAS or pretreated with

10 μM JR-AB2-011 for 6 h were treated with the indicated concentrations of 5-FU, and cell viability was assessed after 24 h. JR, JR-AB2-011, a selective mTORC2 inhibitor. **h**, HCT116 cells stably expressing indicated cGAS mutants were treated with the indicated concentrations of 5-FU and cell viability was assessed after 24 h. **i**, Colony formation assays were performed in HCT116 cells in the presence of 2 μM 5-FU in DMSO. **j**, HCT116 cells stably expressing indicated shRNAs were treated with the indicated concentrations of 5-FU and cell viability was assessed after 24 h. **k**, Cell viability assays measuring idarubicin response of THP-1 cells stably expressing indicated cGAS mutants. Data are shown as mean ± s.e.m. from three independent experiments. Unpaired two-tailed *t*-test (**a**, **c**, **e**, **i**) and Kruskal–Wallis one-way analysis of variance (ANOVA) followed by Dunn's multiple comparison tests (**d**, **f**–**h**, **j**, **k**). Experiments were repeated four times (**b**) with similar results. Numerical source data are available.

S37A did not (Fig. 4h,i and Extended Data Fig. 9g). Unlike ccGAS, STING activation or knockdown did not affect 5-FU resistance in colorectal cancer cells (Fig. 4j and Extended Data Figs. 9g and 10a). Additionally, ccGAS regulation of chemoresistance was independent of MLH1 status (Extended Data Fig. 10b,c). This STING-independent chemoresistance mediated by ccGAS generalized across colorectal cancer cell lines with varying microsatellite instability (Fig. 4i–k and Extended Data Fig. 10d,e). Unlike mTORC1 inhibition[30], inhibiting mTORC2 via JR-AB2-011 or mTOR knockdown desensitized HCT116 cells to 5-FU (Fig. 5a and Extended Data Fig. 10b). Knocking down mTORC2-specific subunits RICTOR and SIN1 also decreased 5-FU sensitivity in HCT116 cells (Extended Data Fig. 10f). Notably, overexpression of S37D mutant,

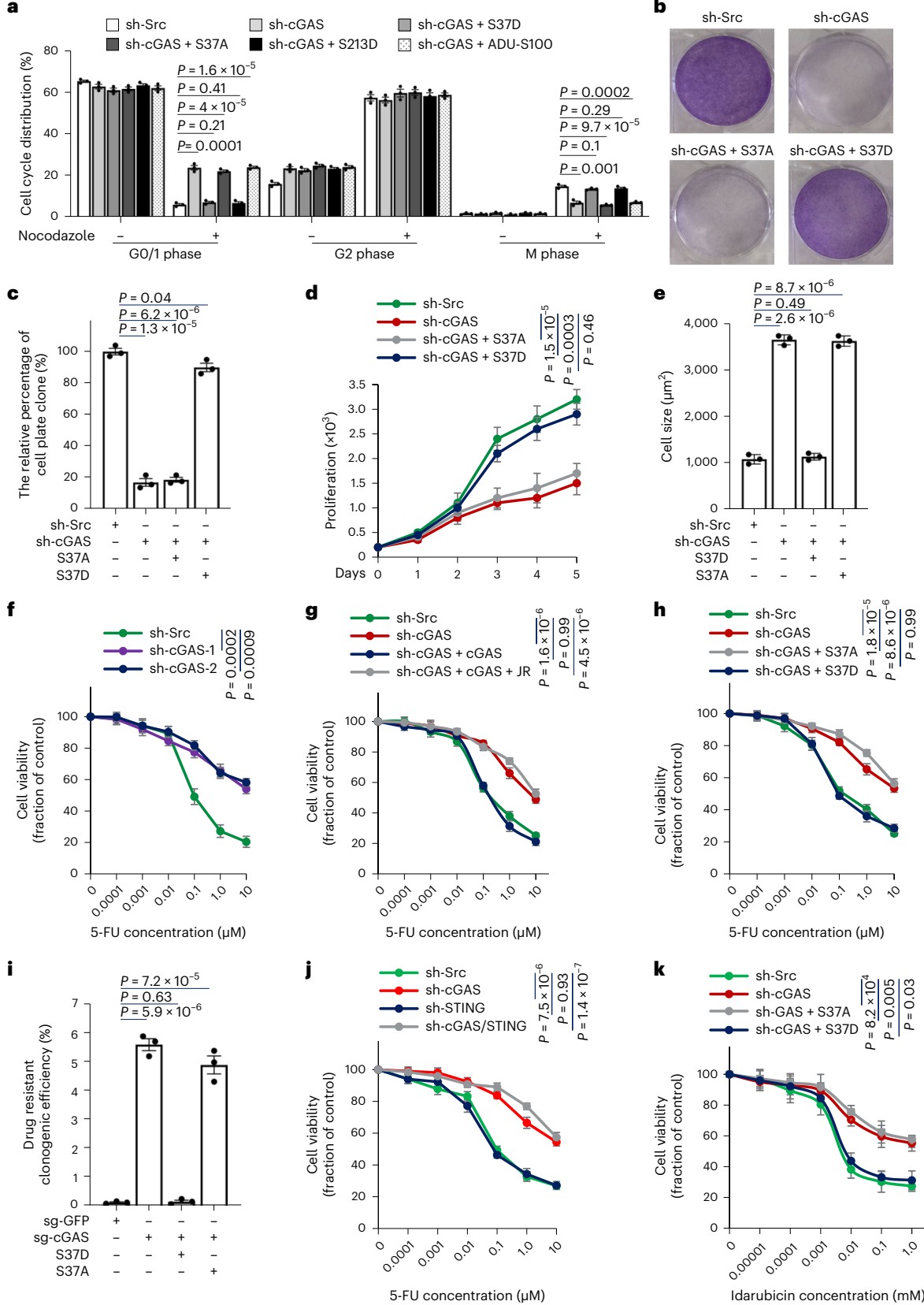

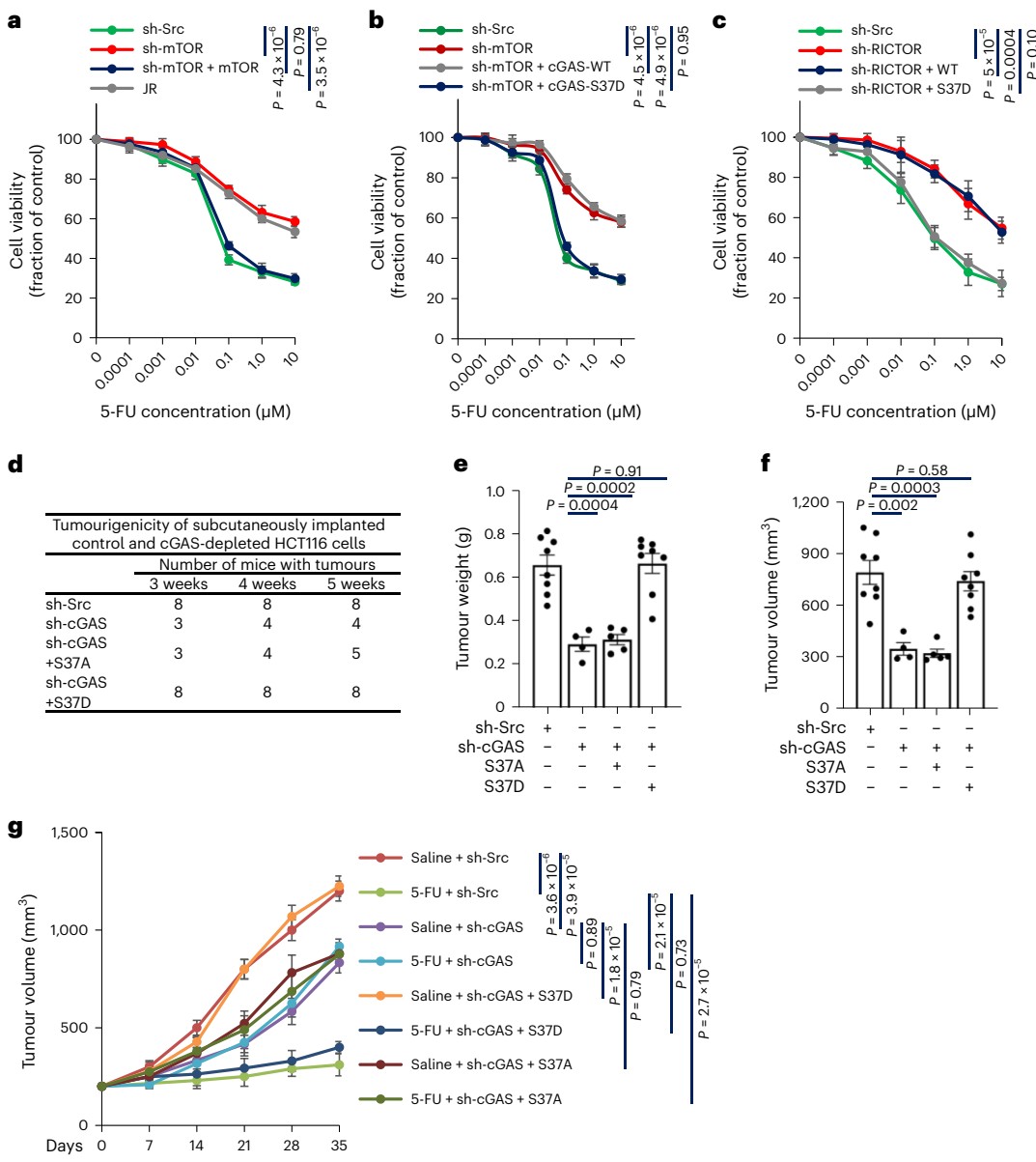

**Fig. 5 | mTORC2-driven ccGAS directs colorectal cancer plasticity and acquired chemoresistance in vivo. a**, mTOR-knockdown HCT116 cells stably transfected mTOR or pretreated with 10 μM JR-AB2-011 for 6 h were treated with the indicated concentrations of 5-FU, and cell viability was assessed after 24 h. **b**, mTOR-knockdown HCT116 cells stably expressing indicated cGAS mutants were treated with the indicated concentrations of 5-FU and cell viability was assessed after 24 h. **c**, RICTOR-knockdown HCT116 cells stably expressing indicated cGAS mutants were treated with the indicated concentrations of 5-FU and cell viability was assessed after 24 h. **d**, HCT116 cells lentivirally infected with indicated shRNAs or cGAS mutants were tested for tumour formation in nude mice ($2 \times 10^7$ cells per mouse). $n = 8$ mice per group. **e**, Mice were killed

after 5 weeks of xenotransplantation and tumour weight in each group was calculated. $n$ (left to right) = 8, 4, 5, 8 mice. **f**, Mice were killed after 5 weeks of xenotransplantation and the tumour volume in each group was calculated. $n$ (left to right) = 8, 4, 5, 8 mice. **g**, Mouse xenograft experiments using HCT116 cells stably expressing cGAS shRNA or cGAS mutants. When the tumour diameter reached 5 mm, 5-FU (23 mg kg$^{-1}$, twice a week) was intraperitoneally injected for five consecutive weeks. Tumour growth curves were calculated. $n = 5$ mice per group. Data are shown as means ± s.e.m. from three independent experiments (**a–c**). Unpaired two-tailed $t$-test (**e**, **f**) and Kruskal–Wallis one-way ANOVA followed by Dunn's multiple comparison tests (**a–c**, **g**). Numerical data are available as source data.

---

but not wild-type cGAS, resensitized mTOR or RICTOR-knockdown HCT116 cells to 5-FU (Fig. 5b,c), indicating that mTORC2-driven ccGAS regulates chemosensitivity.

Given that ccGAS depletion inhibits cancer cell proliferation under normal conditions but promotes chemoresistance under drug exposure, we examined whether ccGAS depletion inhibits cancer growth to confer chemoresistance in vivo. In a mouse xenograft model, cGAS-knockdown or control HCT116 cells were injected subcutaneously into nude mice. Three weeks later, tumour formation was significantly reduced in cGAS-knockdown mice (37.5%) versus controls

(100%) (Fig. 5d and Supplementary Table 5). At 5 weeks, tumour volume and weight were also significantly decreased in knockdowns (Fig. 5e,f). Overexpressing cGAS-S37D rescued tumour formation and growth in knockdowns, whereas cGAS-S37A did not (Fig. 5d,g). These results indicate that ccGAS depletion inhibits colorectal cancer tumorigenicity by suppressing proliferative capacity.

5-FU was administered intraperitoneally for 5 weeks when tumours reached 5 mm in diameter. Consistent with previous tumour growth observation, cGAS-knockdown tumours grew slower than controls under saline treatment (Fig. 6a). However, while controls responded

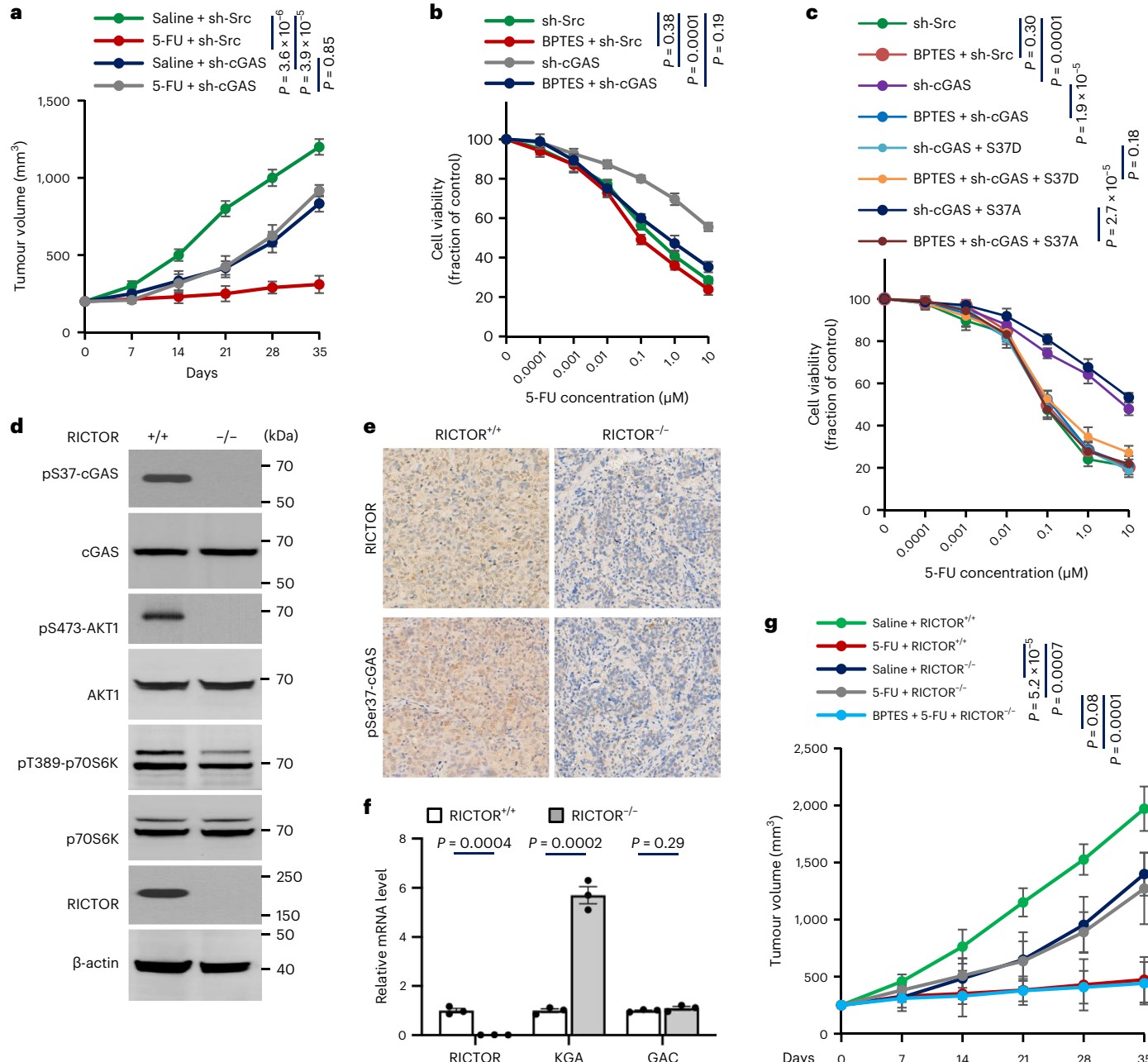

**Fig. 6 | KGA inhibition overcomes chemoresistance induced by disruption of mTORC2–ccGAS axis. a**, Mouse xenograft experiments were performed with HCT116 cells stably expressing cGAS shRNA. When the tumour diameter reached 5 mm, 5-FU (23 mg kg⁻¹, twice a week) was intraperitoneally injected for five consecutive weeks. Tumour growth curves were calculated. $n = 5$ mice per group. **b**, HCT116 cells stably expressing cGAS shRNA were treated with 10 μM BPTES and the indicated concentrations of 5-FU and cell viability was assessed after 24 h. $n = 3$ independent experiments per group. **c**, HCT116 cells stably expressing the indicated shRNA or cGAS mutants were treated with 10 μM BPTES and the indicated concentrations of 5-FU and cell viability was assessed after 24 h. $n = 3$ independent experiments per group. **d**, Immunoblot analysis of WCLs derived from PDX models (RICTOR⁺/⁺ and RICTOR⁻/⁻). **e**, Immunohistochemical staining of patient-derived xenograft (PDX) models for the indicated proteins. **f**, qPCR analysis of indicated gene expression in RICTOR homozygous deletion PDX model normalized to ACTB. **g**, The PDX model was established with RICTOR homozygous deletion colorectal cancer. When the tumour reached 5 mm in diameter, intraperitoneal injection was performed using 5-FU (23 mg kg⁻¹, twice per week) and BPTES (30 mg kg⁻¹, twice per week) for 5 weeks. Tumour growth curves were calculated. $n = 5$ mice per group. Data are shown as mean ± s.e.m. Kruskal–Wallis one-way ANOVA followed by Dunn's multiple comparison tests (**a**–**c**, **g**) and unpaired two-tailed $t$-test (**f**). Experiments were repeated three times (**d**, **e**) with similar results. Numerical data and unprocessed blots are available as source data.

sensitively to 5-FU, cGAS-knockdown tumours did not (Fig. 6a). Overexpressing S37D cGAS restored 5-FU response in knockdowns, whereas cGAS-S37A did not (Fig. 5g). These findings reveal that ccGAS depletion promotes chemoresistance even at the expense of reduced proliferation, suggesting that restoring ccGAS function overcomes chemoresistance.

## KGA inhibition overcomes ccGAS deficiency-induced chemoresistance in mice

Given that ccGAS depletion strongly induces KGA expression (Fig. 3) and GLS1 inhibition overcomes resistance to mTOR inhibitors in lung tumours and glioblastoma[28,29], we assessed whether inhibiting KGA with bis-2-(5-phenylacetamido-1,2,4-thiadiazol-2-yl)ethyl sulfide

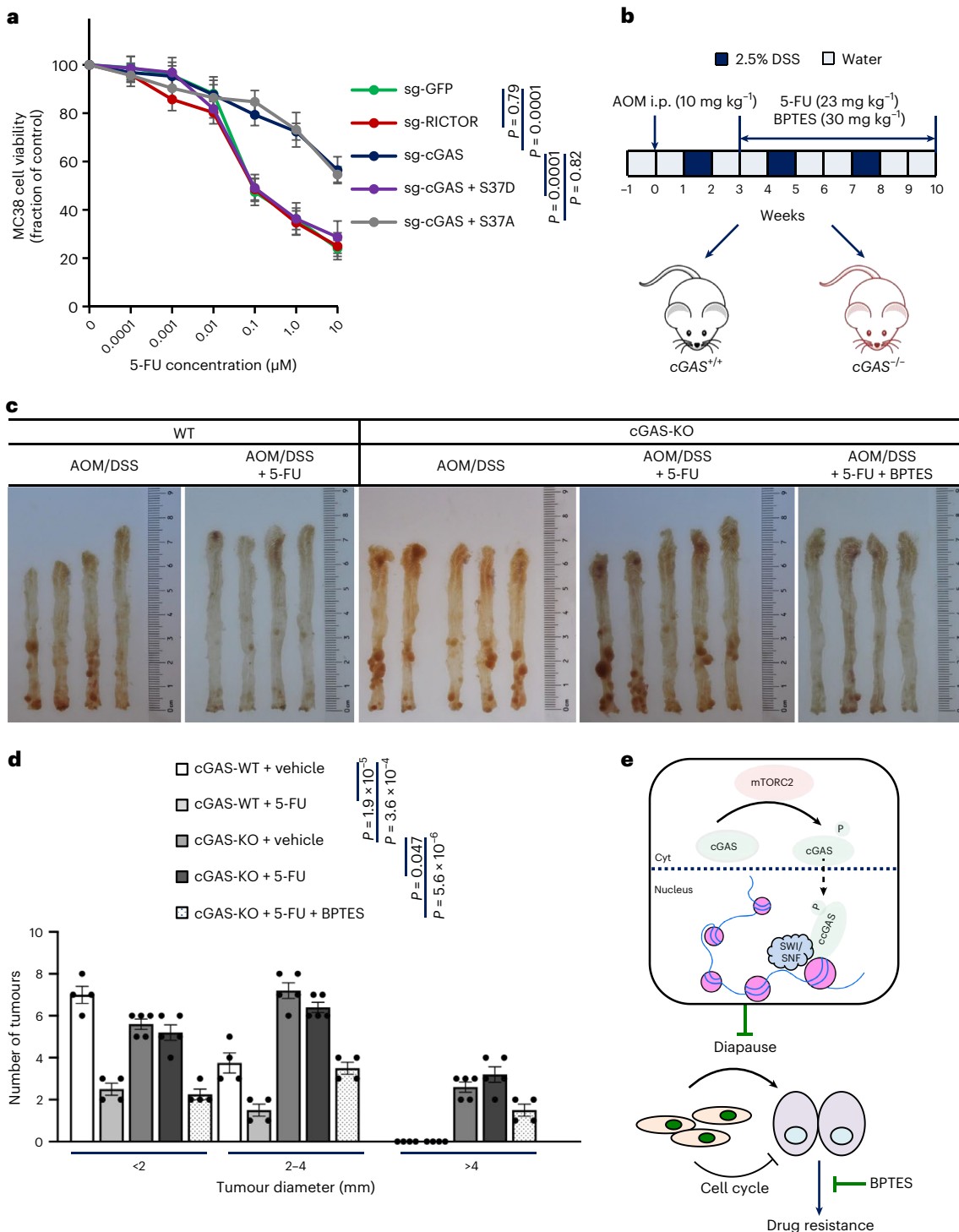

**Fig. 7 | Targeting KGA re-establishes chemotherapy sensitivity in tumours of cGAS-deficient mice. a**, MC38 cells stably expressing indicated hcGAS mutants were treated with the indicated concentrations of 5-FU and cell viability was assessed after 24 h. $n = 3$ independent experiments per group. **b**, The colitis-associated colorectal cancer model was established with AOM/DSS in cGAS-WT and knockout (KO) littermate mice. After 21 days of treatment, intraperitoneal injection (i.p.) was performed using 5-FU (23 mg kg$^{-1}$, twice per week) and BPTES (30 mg kg$^{-1}$, twice per week) for 7 weeks. **c**, Representative images of AOM/DSS-induced tumour in colon tissues demonstrating the number and location of colon tumours. **d**, The tumour number of AOM/DSS mice in each group was counted by macroscopic examination of colon tissue. $n$ (left to right) = 4, 4, 5, 5 and 4 mice. **e**, mTORC2 phosphorylates cGAS to promote its chromatin localization and SWI/SNF recruitment to regulate target gene expression, thereby mediating plasticity and chemoresistance in colorectal cancer. Data are shown as mean ± s.e.m. Kruskal–Wallis one-way ANOVA followed by Dunn's multiple comparison tests (**a**) and unpaired two-tailed $t$-test (**d**). Numerical data are available as source data.

(BPTES)[45] overcomes ccGAS-depletion-induced chemoresistance. BPTES sensitized cGAS-knockdown HCT116 cells to 5-FU-induced death without toxicity (Fig. 6b). While S37A overexpression did not improve 5-FU sensitivity in knockdowns, combining BPTES and 5-FU markedly enhanced the response (Fig. 6c). In contrast, BPTES provided no additional benefit to S37D overexpressing cGAS-knockdown cells treated with 5-FU (Fig. 6c). These results suggest that inhibiting KGA with BPTES may overcome ccGAS depletion-induced chemoresistance.

To validate these findings in vivo, a patient-derived xenograft model with RICTOR deletion, impairing mTORC2, was used (Fig. 6d–f and Extended Data Fig. 10g). RICTOR-null tumours grew significantly slower than controls (Fig. 6g). Mice bearing RICTOR-null tumours received 5-FU alone or with BPTES. While 5-FU strongly inhibited control tumour growth, it had no effect on RICTOR-null lesions (Fig. 6g). Notably, combining 5-FU and BPTES resensitized RICTOR-null tumours, reducing their growth comparably to 5-FU monotherapy of controls (Fig. 6g). These findings demonstrate that disruption of the mTORC2–ccGAS axis inhibits tumour proliferation but induces resistance, an effect overcome through dual KGA inhibition and chemotherapy.

While murine cGAS is predominantly nuclear, mTORC2 contributes minimally to its chromatin localization in mice (Extended Data Fig. 10h). This differs from humans likely due to lack of N-terminal NLS conservation between species. Notably, the fundamental ccGAS–KGA pathway regulating glutaminolysis and chemotherapy response remained functionally conserved in murine colorectal cancer cells (Fig. 7a and Extended Data Fig. 10i–k). An azoxymethane (AOM)/dextran sodium sulfate (DSS) model of colitis-associated cancer in cGAS-KO mice was used (Fig. 7b), mimicking human disease pathologically[46]. Colorectal tumours developed in both cGAS[+/+] and cGAS[−/−] mice after AOM/DSS treatment, predominantly in the distal colon (Fig. 7c). Of note, cGAS deficiency significantly increased tumour burden, with greater numbers and larger sizes of colorectal tumours in cGAS-KO mice compared with their wild-type littermates (Fig. 7c,d). This contrasted with ccGAS knockdown inhibiting cancer cell growth in vitro, suggesting cGAS deficiency may promote in vivo tumour growth by relieving anti-tumourigenic effects through the cGAS–STING pathway.

Notably, as observed in cell lines, cGAS deficiency also conferred resistance to 5-FU in the AOM/DSS tumour model. 5-FU treatment significantly reduced tumour numbers and sizes in wild-type littermate control mice but not cGAS-KO mice (Fig. 7c,d). However, combining 5-FU and BPTES markedly attenuated tumour burden in KOs (Fig. 7c,d). Together, these results indicate that KGA inhibition can overcome cGAS deficiency-induced chemoresistance and that targeting both ccGAS and KGA may represent a promising therapeutic strategy for colorectal cancer.

## Discussion

In this study, we uncovered a mechanism by which mTORC2-induced cGAS phosphorylation at serine 37 promotes its chromatin recruitment and functions. This post-translational modification represents a key regulatory node influencing cGAS-dependent processes in cancer cells. By modulating cGAS–chromatin localization, PI3K-mTORC2 signalling supports proliferation under normal conditions but its disruption provokes acquired resistance to chemotherapy (Fig. 7e). This elucidates the importance of the PI3K-mTORC2-ccGAS pathway in tumour growth and treatment response.

We found that inhibiting the mTORC2–ccGAS axis drives colorectal cancer cells into a diapause-like state of chemoresistance. This provides insights into how mTOR inhibition clinically elicits drug tolerance[47,48]. Emerging evidence associates such chemoresistance with cell plasticity[49–52], though mechanisms were unclear. Tracing resistance back to the mTORC2–ccGAS node, we discovered diapause-like plasticity underlies resistance upon ccGAS depletion. Adding ccGAS as a biomarker may help optimize strategies combining mTOR inhibitors with KGA blockade to eliminate persistent quiescent chemoresistance tumours.

Our study also revealed an epigenetic mechanism, whereby mTORC2 modifies chromatin through ccGAS. ccGAS selectively recruits the SWI/SNF complex to regions regulating DNA replication and glutaminolysis. Depleting ccGAS strongly induced KGA expression, validating links between SWI/SNF defects and glutaminase inhibition sensitivity[53]. Characterizing ccGAS cistromes and targets may elucidate plasticity governance across contexts. While targeting SWI/SNF

is challenging, selectively inhibiting downstream nodes such as KGA may improve precision oncology.

While our findings demonstrate mTORC2-mediated S37 phosphorylation promotes cGAS–chromatin localization, we cannot exclude contributions of additional factors downstream of PI3K–mTORC2 signalling. Our findings require validation across diverse models and clinical settings, as additional ccGAS regulators remain unknown. Validating findings using multi-omic patient data and longitudinal analyses strengthens translational relevance. In summary, we provide provisional evidence that the mTORC2–ccGAS–KGA axis mediates cell plasticity and acquired chemoresistance in cancer. Further exploring modulatory factors and pathways may optimize precision strategies against adaptive survival, pending validation. Continued investigation holds promise to refine the clinical impact of cGAS biology.

## Online content

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

## Methods

### Ethics statement

All animal experiments were conducted in accordance with the 'Guide for the Care and Use of Laboratory Animals' and 'Principles for the Utilization and Care of Vertebrate Animals' and approved by the Institutional Human Research Ethics and Animal Care and Use Committee at Peking University and Peking University Shenzhen Hospital. Mice were monitored daily and experiments were terminated if tumour diameters exceeded 15 mm or volumes exceeded 2,000 mm$^3$ to avoid unnecessary suffering, according to guidelines established by our Institutional Animal Care and Use Committee. Tumour specimens were collected from six male patients with colorectal cancer aged 40–60 years who underwent surgical resection at Peking University Shenzhen Hospital. All patients provided written informed consent and the study was approved by the Institutional Review Board of Peking University Shenzhen Hospital. No compensation was provided for the collection of tumour samples. Two PDX lines were established: RICTOR$^{-/-}$ from a 47-year-old man with AJCC stage IIIB colorectal cancer and RICTOR$^{+/+}$ from a 53-year-old man with AJCC stage IIIA colorectal cancer.

### Compounds, plasmids and antibodies

GDC-0941 (Selleck, no. S1065), BEZ-235 (Selleck, no. S1009) and MK-2206 (Selleck, no. S1078), rapamycin (Selleck, no. S1039), KU-0063794 (Selleck, no. S1226), PF-4708671 (Selleck, no. S2163), JR-AB2-011 (Selleck, no. E1151), Nocodazole (Selleck, no. S2775), 5-FU (Selleck, no. S1209), 2′,3′-cGAMP (Selleck, no. S7904), BPTES (Selleck, no. S7753) and Bioactive Compound Library (Selleck, no. L1700) were purchased from Selleck Chemicals. AOM (Sigma, A5486) was purchased from Sigma and ADU-S100 (MCE, HY-12885) was purchased from MedChemExpress.

Full-length cGAS, RICTOR, SIN1, mTOR and STING were cloned into the pFlag-CMV-2 vector. shRNA-resistant cGAS-WT, cGAS-S37D, cGAS-S37A, cGAS-S305D, cGAS-S305A and cGAS-S213D were cloned into the pCMV-HA-His vector.

Antibodies used in immunoblotting were anti-cGAS (CST, no. 15102, D1D3G, 1:1,000 dilution), anti-HELLS (CST, no. 7998, 1:1,000 dilution), anti-β-actin (CST, no. 4967, 1:1,000 dilution), anti-PDI (CST, no. 2446, 1:1,000 dilution), anti-HA (CST, no. 3724, C29F4, 1:1,000 dilution), anti-Flag (CST, no. 14793, D6W5B, 1:1,000 dilution), anti-histone H3 (CST, no. 9715, 1:1,000 dilution), anti-ARID1A (CST, no. 12354, D2A8U, 1:1,000 dilution), anti-MCM7 (CST, no. 3735, D10A11, 1:1,000 dilution), anti-SMARCC2 (CST, no. 12760, D8O9V, 1:1,000 dilution), anti-Akt (CST, no. 9272, 1:1,000 dilution), anti-pSer473-Akt (CST, no. 4060, D9E, 1:1,000 dilution), anti-MCM3 (CST, no. 4012, 1:1,000 dilution), anti-GLS1 (CST, no. 49363, E4T9Q, 1:1,000 dilution), anti-mTOR (CST, no. 2983, 7C10, 1:1,000 dilution), anti-RICTOR (CST, no. 2114, 53A2, 1:1,000 dilution), anti-RAPTOR (CST, no. 48648, E6O3A, 1:1,000 dilution), anti-SIN1 (CST, no. 12860, D7G1A, 1:1,000 dilution), anti-PAI-1 (CST, no. 11907, D9C4, 1:1,000 dilution), anti-BAF200 (CST, no. 82342, D8D8U, 1:1,000 dilution), anti-CDC45 (CST, no. 11881, D7G6, 1:1,000 dilution), anti-p70 S6K (CST, no. 9202, 1:1,000 dilution), anti-pThr389-p70 S6K (CST, no. 9209, 1:1,000 dilution), anti-4E-BP1 (CST, no. 9452, 1:1,000 dilution), anti-pSer65-4E-BP1 (CST, no. 9451, 1:1,000 dilution) and anti-GAPDH (CST, no. 5174, D16H11, 1:2,000 dilution) were purchased from Cell Signalling Technology. Anti-KGA (Proteintech, 20170-1-AP, 1:1,000 dilution) and anti-GAC (Invitrogen, PA5-40134, 1:1,000 dilution) were purchased from Thermo Fisher Scientific. Anti-SMARCA4 (Sigma, MABE121, 1:1,000 dilution) and anti-MCM5 (Sigma, SAB1406111, 1:1,000 dilution) were purchased from Sigma-Aldrich.

Antibodies used in immunoprecipitation and immunofluorescence were anti-cGAS (CST, no. 79978, E5V3W, IF, 1:200 dilution), anti-cGAS (CST, no. 31659, D3O8O, IP, 1:200 dilution), anti-HA (CST, no. 3724, C29F4, 1:200 dilution), anti-H2A (CST, no. 12349, D6O3A, 1:200 dilution) and Phalloidin (CST, no. 8953, 1:200 dilution) and were purchased from Cell Signalling Technology. The polyclonal anti-pSer37-cGAS antibodies generated by us were derived from rabbits. The antigen sequence used for immunization was cGAS aa$^{29–47}$ (GAPMDPTES*PAAPEAALPK), where S* indicates a phosphorylated serine residue in these synthetic peptides. The antibodies were affinity purified using the antigen peptide column, but they were not counter selected on unmodified antigen.

### Lentivirus infection

shRNA lentiviral vectors targeting human cGAS (TRCN0000146282 and TRCN0000149984), RAPTOR (TRCN0000039772), RICTOR (TRCN0000307122), mTOR (TRCN0000039785) and STING (TRCN0000135555); mouse cGAS (TRCN0000417906) were purchased from Sigma. Single-guide RNAs of human cGAS (AAGTGCGACTC-CGCGTTCAG), RICTOR (CCATCTGAATAACTTTACTA), STING (CGGGC-CGACCGCATTTGGG), mouse cGAS (GAGCGTGACGGGGACACCA) were cloned into lentiCRISPRv2 vectors. The HEK293T cell lines were transfected overnight at 37 °C using 15 μg cloned vector for each gene, 4.5 μg pMD2. G (Addgene, 12259) and 12 μg psPAX2 (Addgene, 12260). After 24 h of virus production, the medium containing virus was collected and cell debris was removed by centrifugation at 300$g$ for 10 min. The virus in the supernatant was added to infect HCT116 cells overnight in the medium with 8 μg ml$^{-1}$ polybrene. After 48 h infection, cells were then subjected to puromycin selection for 7 days. Clones with stable knockdown of cGAS, RAPTOR, RICTOR, mTOR or knockout of cGAS, RICTOR were identified and verified by IB.

### Cell transfection, immunoprecipitation and immunoblotting

HCT116, HT29, SW480, MC38, HEK293T, MDA-MB-231, 786-0 and HCC44 cells were purchased from ATCC, authenticated by STR profiling and tested for *Mycoplasma* contamination. Cells were transfected with various plasmids using Lipofectamine 2000 (Invitrogen) or peptides using the BioPORTER QuikEase Protein Delivery kit (Sigma) according to the manufacturer's protocol. For IP assays, cells were lysed with HEPES lysis buffer (20 mM HEPES, pH 7.2, 50 mM NaCl, 0.5% Triton X-100, 1 mM NaF and 1 mM dithiothreitol (DTT)) supplemented with protease inhibitor cocktail (Roche).

IP was performed using anti-HA agarose (Sigma, A2095) for HA-tagged proteins, anti-FLAG M2 magnetic beads (Sigma, M8823) for Flag-tagged proteins or the indicated primary antibody and protein A/G agarose beads (Sigma) at 4 °C. The immunocomplexes were then washed with HEPES lysis buffer four times. Both lysates and immunoprecipitates were examined using the indicated primary antibodies followed by detection with the related secondary antibody and the SuperSignal West Pico chemiluminescence substrate (Thermo).

### High-throughput screening based on eBRET2 technique

The pcDNA3.1(+) vector was used as a Backbone to clone cGAS construct fused to the C terminus of a GFP2 tag and to clone H2A construct fused to the C terminus of Rluc8. At 24 h after transfection the activity of the biosensor component GFP2 was verified by fluorescence microscopy with the Keyence BZ-8000 (Keyence), and transfected HCT116 cells with eBRET2 constructs were isolated by flow cytometer. Then, 185 μl of resuspended cells ($5 \times 10^4$ cells per well) were distributed in 96-well microplates (white Optiplate; Packard BioScience). Then, 5 μl each drug (the final concentration is about 10 μM) was added to cells in triplicate wells and microplates were shaken for 2 min. After 6 h, 10 μl of the Rluc8 substrate DeepBlueC was injected into each well to obtain a final concentration of 5 μM, and readings were then collected 2 s after each injection (Mithras LB 940 plate reader; Berthold Technologies). Signals at 395 (Rluc luminescence signal) and 510 nm (emission of light from excited GFP2) were measured sequentially and 510/395 ratios (eBRET2 signal) were calculated and expressed as eBRET2.

### In vitro eBRET2 measurements

The eBRET2 biosensors were constructed and 24 h after transfection, the lysates of HCT116 cells with eBRET2 constructs and controls were

pipetted in triplicates into a 96-well plate (COSTAR Lumiplates Flat White; Corning) with 20 µl per well. The luciferase substrate Coelenterazine 400a (CLZ400a; 0.5 mg ml$^{-1}$ in 100% ethanol, NanoLight) was diluted 1:100 in eBRET2 assay buffer (PBS supplemented with 1 g l$^{-1}$ D-glucose monohydrate (Roth), 0.1 g l$^{-1}$ calcium chloridedihydrate (Merck) and 0.1 g l$^{-1}$ magnesium chloride-hexahydrate (Merck)) and incubated for 20 min at room temperature under light protection to avoid light emission caused by substrate oxidation. Then, 100 µl of the substrate solution was added per well with the injector of the Tecan Infinite M1000Pro plate reader (Tecan). For eBRET2 measurements the dual colour luminescence mode of the Tecan plate reader was used with two filters in 395 nm (blue filter) and 510 nm (green filter). The measurements were carried out with an integration time of 1 s.

## Recombinant cGAS purification

The coding sequence of cGAS was cloned into the pET28a(+) vector with an N-terminal Flag tag. Plasmids were transformed into *Escherichia coli* BL21(DE3) cells. For cGAS expression, cells were induced by the addition of 0.5 mM isopropyl-b-D-thiogalactoside and incubated for 12 h at 20 °C. Cells were collected and disrupted by sonication in a buffer containing 50 mM Tris-HCl (pH 7.4), 150 mM NaCl and 1 mM PMSF. Flag-cGAS fusion proteins were purified by anti-FLAG M2 affinity resin. The fusion proteins were eluted with Flag peptides and further purified by size-exclusion chromatography using a Superdex 200 Increase 10/300 column (GE Healthcare) in 50 mM Tris-HCl, pH 7.4, 150 mM NaCl and 1 mM DTT.

BL21 (DE3) cells containing MBP-His$_8$-cGAS-His$_6$ were collected by centrifugation and lysed by cell disruption (Emulsiflex-C5, Avestin) in 20 mM imidazole (pH 8.0), 150 mM NaCl, 5 mM β-ME, 0.1% NP-40, 10% glycerol, 1 mM PMSF, 1 µg ml$^{-1}$ antipain, 1 µg ml$^{-1}$ pepstatin and 1 µg ml$^{-1}$ leupeptin. Centrifugation-cleared lysate was applied to Ni-NTA agarose (QIAGEN), washed with 10 mM imidazole (pH 8.0), 150 mM NaCl, 5 mM β-ME, 0.01% NP-40 and 10% glycerol, and eluted with the same buffer containing 500 mM imidazole (pH 8.0). The MBP tag and His$_6$ tag were removed using TEV protease treatment for 16 h at 4 °C. Cleaved protein was applied to a Source 15 Q anion exchange column and eluted with a gradient of 200 mM-300 mM NaCl in 20 mM HEPES (pH 7.0) and 2 mM DTT followed by size-exclusion chromatography using a Superdex 200 prepgrade column (GE Healthcare) in 25 mM HEPES (pH 7.5), 150 mM NaCl, 1 mM MgCl$_2$ and 1 mM DTT.

## In vitro phosphorylation assays

HCT293T cells transfected with HA-RICTOR were cultured under serum starvation (for 48 h) or insulin stimulation (100 nM insulin for 30 min) conditions. HA immunoprecipitation was then performed in CHAPS buffer (50 mM Tris-HCl (pH 7.5), 120 mM NaCl and 0.3% CHAPS). The immunoprecipitate (mTORC2) was washed four times in CHAPS buffer and supplied as the kinase sources for in vitro phosphorylation assay. The immunoprecipitate (from 4 mg total proteins) was incubated with 4 µg Flag-cGAS and 200 µM ATP in the kinase assay buffer (10 mM Tris-HCl (pH 7.5), 10 mM MgCl$_2$, 0.1 mM EDTA and 2 mM DTT) at 30 °C for 1 h. The reactions were gently tapped every 15 min to mix the reaction well, then subjected to IB or MS analysis.

HCT116 cells lysed in 50 mM Tris, 0.15 M NaCl, 0.1% SDS, 1% Triton X-100, 0.5% sodium deoxycholate, pH 7.5, with 2.5 U µl$^{-1}$ Benzonase (Merck) and cOmplete Protease Inhibitor Cocktail (Roche) were subjected to IP with 5 µg ml$^{-1}$ anti-RICTOR/RAPTOR antibodies and Dynabeads Protein G (Thermo Fisher). After 1 h, beads were washed three times in PBS. In vitro phosphorylation reactions were carried out in 50 mM Tris-HCl, 1 mM ATP, 10 mM MgCl$_2$, 1 mM EGTA, 10 mM NaF, 20 mM β-glycerophosphate, pH 7.5 with PhosSTOP phosphatase inhibitors (Merck) and approximately 100 nM of recombinant cGAS substrate and 10 nM of mTORC2 for 30 min at 37 °C. The reactions were subjected to IB or MS analysis.

## MS analysis for cGAS phosphorylation

In vitro cGAS phosphorylation reactions were stopped by the addition of 8 M urea and 5 mM DTT, incubated at 37 °C for 60 min and alkylated with 15 mM iodoacetamide for 30 min in the dark at room temperature. Then, 50 mM Tris-HCl (pH 7.8) and 1 mM CaCl$_2$ were added to make the final urea concentration 1 M, and then digested with trypsin (trypsin:protein, 1:50) at 37 °C overnight. After overnight digestion, the sample was acidified by adding formic acid to a final concentration of 2% to a pH of 2–3 to stop the enzymatic activity. Desalt peptides by using Pierce peptide desalting spin columns (Thermo, 89852). Phosphorylated peptides were enriched by High-Select Fe-NTA phosphopeptide enrichment kit (Thermo, A32992). The enriched products were analysed by high-precision MS and data retrieval was completed by professional proteome discoverer software.

For in vivo cGAS phosphorylation detection, proteins in different cell components were obtained through a Chromatin Extraction Kit (Abcam, ab117152) and ProteoExtract kit (Millipore, 539790). Anti-cGAS antibody-mediated IP was performed with whole-cell lysates (WCLs) derived from three 10-cm dishes of HCT116 cells. The IP proteins were resolved by SDS–PAGE and identified by Coomassie staining. The bands containing cGAS were reduced with 10 mM DTT for 30 min, alkylated with 55 mM iodoacetamide for 45 min and in-gel-digested with trypsin enzymes. The resulting peptides were extracted from the gel and analysed by microcapillary reversed-phase (C18) liquid chromatography (LC)–MS/MS using a high-resolution Orbitrap Elite (Thermo Fisher Scientific) in positive ion DDA mode (Top 6) via higher energy collisional dissociation coupled to a Proxeon EASY-nLc II nano-HPLC. MS/MS data were searched against the reviewed UniProt Human protein database (v.20230119 containing 20,308 entries) using Mascot v.2.5.1 (Matrix Science) and data analysis was performed using Scaffold v.4.4.8 software (Proteome Software). Peptides and modified peptides were accepted if they passed a 1% false discovery rate threshold.

## SILAC medium preparation and cell culture conditions

All standard stable isotope labelling with amino acids in cell culture (SILAC) medium preparation and labelling steps were followed as previously described. In brief, the base medium for DMEM (Macgene) was divided into two parts and to each added L-arginine (Arg$^0$) and L-lysine (Lys$^0$) (light) or $^{13}$C$_6$ $^{15}$N$_4$-L-arginine (Arg$^{10}$) and $^{13}$C$_6$ $^{15}$N$_2$-L-Lysine (Lys$^8$) (heavy) to generate the two SILAC-labelling mediums. Each medium with the full complement of amino acids at the standard concentration, was sterile filtered through a 0.22-µm filter (Milipore). Cells were grown in the corresponding labelling medium, prepared as described above, supplemented with 2 mM L-glutamine (Gibco) and 10% dialysed foetal bovine serum (Sigma) plus antibiotics (Gibco), in a humidified atmosphere with 5% CO$_2$ at 37 °C. Cells were cultured in labelling medium for at least six cell divisions.

## TMT labelling

Protein at 100 µg from each sample was reduced with 100 mM DTT at 56 °C for 30 min, followed by the addition of 20 mM iodoacetamide (Sigma) at room temperature while in the dark with alkylating solution for 45 min. Then, the protein sample was diluted by adding 100 mM triethylammonium bicarbonate (TEAB; Sigma) to a urea concentration below 2 M. Finally, each protein sample was digested with trypsin at a mass ratio of 1:50 (trypsin:protein) for the first digestion overnight for 16–18 h and at 1:100 (trypsin:protein) for the second 4 h digestion. After trypsin digestion, peptides were desalted with a Strata X C18 SPE column (Phenomenex) and vacuum dried. Subsequently, peptides were reconstituted in 0.5 M TEAB and processed according to the manufacturer's protocol for the six-plex TMT kit (Thermo Fisher Scientific). In brief, one unit of TMT reagent (defined as the amount of reagent required to label 100 µg of proteins) was equilibrated at room temperature; 100 µg of each sample was resuspended in 24 µl anhydrous acetonitrile and TMT reagent was added to the peptides dissolved in

0.5 M TEAB. After 2 h at room temperature, 8 µl 5% hydroxylamine (*w/v*) was added and incubated for 15 min. The samples were then combined, desalted and dried via vacuum centrifugation.

## LC–MS/MS analysis

The dried peptides were then fractionated using a high-pH reverse-phase HPLC system fitted with an Agilent 300Extend C18 column (5-µm particles, 4.6 mm internal diameter, 250 mm length). In brief, the peptides were first separated with a gradient of 2% to 60% acetonitrile in 10 mM ammonium bicarbonate (pH 10) over 80 min into 51 fractions. Then, the peptides were combined into 18 fractions and dried with vacuum centrifugation. The peptides were dissolved in 0.1% solvent A (formic acid), directly loaded onto a reversed-phase pre-column (Thermo Scientific, Acclaim PepMap 100). Peptide separation was performed using a reversed-phase analytical column (Thermo Scientific, Acclaim PepMap RSLC). The gradient was increased with solvent B (0.1% formic acid and 90% anhydrous acetonitrile) from 8% to 26% in 22 min, from 26% to 40% in 12 min and increased to 80% in 3 min to remain at 80% for the last 3 min. This was carried out at a constant flow rate of 400 nl min$^{-1}$ using an EASY-nLC 1000 UPLC system. The peptides were then analysed on a Q Exactive plus hybrid quadrupole-orbitrap mass spectrometer (Thermo Fisher Scientific). The peptides were analysed via the Q Exactive plus (Thermo Scientific) with a positive ion model and data-dependent acquisition. The resolution of the MS scan was 70,000, and the ion fragment was 17,500. Based on the MS scan, the top 20 precursor ions were selected to fragment with 30 s of dynamic exclusion. The electrospray voltage applied was 2.0 kV. The MS/MS spectra were generated using automatic gain control to prevent overfilling of the Orbitrap and accumulation of 5E4 ions. For the MS scans, the $m/z$ scan range was set from 350 to 1,800. The first fixed mass was set at 100 $m/z$.

## CUT&Tag

The Epicypher protocol for Cleavage under targets and tagmentation was used with slight modifications[34]. Concanavalin A (BioMag Plus, 86057) beads were activated with bead activation buffer and stored on ice until further use. Cells (100,000) were collected and washed with cold PBS. Cells were spun at 800*g* for 5 min at 4 °C and PBS supernatant was removed from the cell pellet. Nuclear extraction buffer was added to the tube and the pellet was gently resuspended by pipetting to lyse cells and extract nuclei. Activated Concanavalin A beads and nuclei were incubated were mixed and incubated at room temperature for 10 min. The nuclei-conjugated bead complexes were resuspended in antibody binding buffer and adding primary antibody rotating on a nutator overnight at 4 °C (2, 1 and 1 µg of IgG, cGAS and HA were added, respectively). The primary antibody mixture was removed and nuclei–bead complexes were incubated with 0.5 µg secondary antibody in digitonin 150 buffer for 1 h at room temperature on the nutator. After secondary antibody incubation, samples were washed with digitonin 150 buffer and resuspended in digitonin 300 buffer supplemented with 2 µl of CUTANA pAG-Tn5 (Epicycpher, 15-1117) added per sample. Samples were incubated with Tn5 for 1 h at room temperature on the nutator. Digitonin 300 buffer was added two times to remove excess enzyme from samples. Targeted chromatin tagmentation was completed following the Epicypher protocol. Libraries were amplified with 14 PCR cycles and purified by single sided 1.3× AMPure bead purification. The NextSeq500 and 35 base-pair paired-end sequencing parameters were used for library sequencing.

## FACS cell cycle analysis

HCT116 cells were synchronized in thymidine 2.5 mM for 24 h and released in nocodazole 100 nM for 16 h. Mitotic arrested cells were collected by shake off and plated in nocodazole for 56 h, then released and seeded for subsequent analysis.

Cells were incubated with 33 µM bromodeoxyuridine for 20 min, collected and centrifuged at 250*g* for 10 min. The pellet was resuspended in 750 µl PBS 1× and fixed by adding 2,250 µl of ice-cold (−20 °C) pure ethanol dropwise while vortexing. Samples were washed once in 1% BSA/PBS and resuspended in 1 ml 2 N HCl and incubated for 25 min at room temperature allowing DNA denaturation. Then 3 ml 0.1 M sodium borate (pH 8.5) was added to neutralize the acidic pH of the HCl solution and samples were incubated at room temperature for 2 min, centrifuged and washed twice in 1% BSA/PBS. Samples were then transferred to an Eppendorf tube and centrifuged at 800*g* for 5 min. Pellets were resuspended in 100 µl pure anti-bromodeoxyuridine antibody (Life Technologies) diluted 1:5 in 1% BSA/PBS and incubated for 1 h at room temperature in the dark. Samples were washed with 1% BSA/PBS and resuspended in 100 µl anti-mouse FITC (Life Technologies) diluted 1:50 in 1% BSA/PBS for 1 h at room temperature in the dark. After washing once with 1% BSA/PBS pellets were resuspended in 1 ml propidium iodide (2.5 µg ml$^{-1}$) and RNase (250 µg ml$^{-1}$) (ribonuclease A from bovine pancreas, Sigma) and incubated overnight at 4 °C. Acquisition was made with FACScalibur and data analysis was performed with FlowJo software.

## Cell viability measurements

Cell viability was typically assessed in 96-well format by Cell Counting Kit-8 (CCK-8; Dojindo) and alamarBlue Cell Viability Reagent Blue. In brief, cells were seeded onto 96-well plates at a density of $2 \times 10^4$ per well. Subsequently, cells exposed to 10 µl CCK-8 reagent (100 µl medium per well) for 1 h at 37 °C, 5% $CO_2$ in an incubator. The absorbance at a wavelength of 450 nm was determined using a FLUOstar Omega microplate reader (BMG Labtech). The alamarBlue (Invitrogen) fluorescence (ex/em 530/590) was measured on a Victor3 plate reader (PerkinElmer). In some experiments, Trypan blue dye exclusion counting was performed by using an automated cell counter (ViCell, Beckman-Coulter). Cell viability under test conditions is reported as a percentage relative to the negative control treatment.

## Fluorescence microscopy

For detection of subcellular localization by immunofluorescence, after being fixed with 4% paraformaldehyde and permeabilized in 0.2% Triton X-100 (PBS), cells were incubated with the indicated antibodies (dilution 1:50) for 8 h at 4 °C, followed by incubation with TRITC-conjugated or FITC-conjugated secondary antibody (dilution 1:200; Cwbio) for 1 h at 25 °C. F-actin was detected with Acti-stain 555 phalloidin (Cytoskeleton). The nuclei were stained with DAPI (Sigma) and images were visualized with a Zeiss LSM 510 Meta inverted confocal microscope.

## ChIP–qPCR

The EZ-Magna ChIP G kit (17-409; Sigma-Aldrich, Merck KGaA) was used to analyse the binding between cGAS and GLS1 promoter. In brief, $1 \times 10^7$ HCT116 cells were digested and collected for protein–DNA crosslinking in formaldehyde. The crosslinking was terminated by glycine. Then, the cells were lysed and ultrasonicated for DNA truncation. The lysates were probed with the cGAS antibody (1:100 dilution, 31659; CST) or rabbit IgG (1:100 dilution, ab171870; Abcam). Protein and DNA were de-crosslinked by proteinase K treatment. DNA was collected and purified, in which the enrichment of GLS1 promoter fragments was analysed by qPCR analysis. Primer sequences used for ChIPs are listed in Supplementary Table 6.

## RNA preparation and quantitative real-time PCR analysis

HCT116 cells were lysed in Trizol (Invitrogen). Total RNA was recovered from cells following the manufacturer's protocol. Two micrograms of purified RNA from each sample was reverse transcribed to single-stranded complementary DNA with an All-In-One RT MasterMix (ABM, G486). The newly synthesized cDNA was mixed with TransStart Top Green qPCRSuperMix (AQ131, Transgen Biotech) in a volume of 20 µl. For quantitative PCR, a Real Time PCR Detection system (ABI 7500) was using to detect each gene in triplicate. Fold changes were

analysed (quantified) relative to the internal control gene ACTB on the basis of the $2^{-\Delta\Delta CT}$ method. The qPCR primer pairs are listed in Supplementary Table 7.

## Dot immunoblot assays

Peptides were spotted onto nitrocellulose membranes, allowing the solution to penetrate (usually 3–4 mm diameter) by applying it slowly at a volume of 1 μl. The membrane was dried and blocked in non-specific sites by soaking in TBST buffer with 5% non-fat milk for IB analysis as described previously.

## Glutaminase activity measurement assay

Glutaminase activity was determined using a GLS Assay kit (Biomedical Research Service). In brief, 2 million cells were washed with ice-cold PBS and lysated by 100 μl 1× Cell Lysis buffer on ice for 5 min with gentle agitation. The supernatant was collected after centrifugation at maximum for 3 min. Followed by measuring the protein concentration, samples were diluted to 0.2–2 mg ml$^{-1}$ and 10 μl was used for GLS assay. Samples were combined with 40 μl fresh glutamine solution and incubated in a humidified 37 °C non-CO$_2$ incubator for at least 2 h. Followed by adding 50 μl TA Assay solution and incubating for another 2 h, the reaction was stopped by adding 50 μl 3% acetic acid. GLS activity was measured by absorbance at OD$_{492}$ using a plate reader (Versamax).

## Colony formation assays

Cells were seeded into 12-well plates (1,000 or 2,000 cells per well) and left for 8–12 days (37 °C and 5% CO$_2$) until the formation of visible colonies. Colonies were washed with PBS, fixed with 10% acetic acid/10% methanol for 20 min and then stained with 0.4% crystal violet in 20% ethanol for 20 min. After staining, the plates were washed and air-dried, and colony numbers were counted. Three independent experiments were performed to generate the standard error of the difference (SED).

## Soft-agar colony formation assay

In brief, the assays were performed using six-well plates where the solid medium consisted of two layers. A total of $1 \times 10^4$ or $3 \times 10^4$ cells were resuspended in DMEM containing 0.35% low-melting agarose (Sigma) and 10% FBS and seeded onto a coating of 0.7% low-melting agarose in DMEM containing 10% FBS. Plates were incubated at 37 °C and 5% CO$_2$. Then, 500 μl of complete DMEM was added every 7 days to keep the top layer moist and 4 weeks later, the cells were stained with iodonitrotetrazolium chloride (1 mg ml$^{-1}$) (Sigma, I10406) for colony visualization and counting. Colonies larger than 0.1 mm in diameter were scored as positive. Three independent experiments were performed to generate the SED.

## In vitro proliferation assay

Cells were assayed for MTS (3-(4,5-dimethylthiazol2-yl)-5-(3-carboxymethoxyphenyl)-2-(4-sulfophenyl)-2H-tetrazolium, inner salt Sigma) reduction. Cells were plated in 96-well plates (100 μl cell suspensions, $1 \times 10^4$ cells ml$^{-1}$). Twenty-four hours later, 0.05 mg ml$^{-1}$ MTS reagent (Promega) was added to each well and incubated at 37 °C for 4 h, followed by absorbance measurement at 490 nm. The values were standardized to wells containing medium alone.

## Tumour growth assay

BALB/c nude mice (6 weeks old, 18.0 ± 2.0 g) were randomly divided into the indicated groups; the mice in the groups were subcutaneously injected with the indicated cells stably expressing the indicated shRNAs or constructs ($2 \times 10^6$ cells in a volume of 100 μl PBS). Tumour growth was measured twice weekly with a caliper for the length ($L$, largest diameter) and perpendicular width ($W$), including the skin fold. The volume was calculated using the formula $V = W^2 \times L/2$. The tumour latency/lag phase was recorded and defined as the duration between tumour implantation and the first palpable tumour detection.

## PDX establishment

Tumour specimens were collected in serum-free DMEM and used within 24 h. Fragments (4–8 mm) or cell pellets were mixed with 10% Matrigel (Corning, 354234) at 4 °C and implanted subcutaneously into the flanks of 6–8-week-old male NOD-SCID mice (five mice per patient sample). Tumour growth was measured twice weekly with a caliper for the length ($L$, largest diameter) and perpendicular width ($W$), including the skin fold. The volume was calculated using the formula $V = W^2 \times L/2$.

The tumour latency/lag phase was recorded and defined as the duration between tumour implantation and the first palpable tumour detection. The initial implant of the patient tumour fragment into the mouse host was defined as passage 0 (P0), followed by serial propagation of tumour fragments in subsequent new hosts. Tumours were collected once they reached a humane end point size of 1.5 cm in largest diameter and were divided for further studies. Engraftment was successful if tumours were established in the initial tumour implant (P0) and as stable engrafters in at least another two successive passages.

## Mouse models of colitis-associated colorectal cancer

cGAS-KO mice were provided by F. You (Peking University Health Science Centre, Beijing) and used and genotyped according to the protocols provided by The Jackson Laboratory. To induce colitis-associated colorectal cancer, mice were intraperitoneally injected with 10 mg kg$^{-1}$ BW AOM (Sigma-Aldrich, A5486) and subsequently treated with three cycles of DSS (36,000–50,000 MW; MP Biomedicals, 0216011010) starting on day 7. Animals received 2.5% DSS in the drinking water for 7 days (first two cycles) or 5 days (third cycle) ad libitum followed by a recovery phase of 14 or 16 days, respectively.

## Statistics and reproducibility

Experimental data were analysed and processed in Excel (2016) and plotted using GraphPad Prism v.9.4.0 (GraphPad Software) using unpaired two-tailed $t$-tests and one-way ANOVA with Dunnett correction for multiple comparisons. All results are shown as the mean ± s.e.m. of multiple independent experiments, not technical replicates. The number of experiments, sample size and statistic tests are reported in the respective figure legends. No statistical method was used to predetermine sample size. Animals were randomly assigned to the experimental groups to prevent bias in group allocation. The order of experimental conditions or stimulus presentation was not randomized. No data were excluded from the analyses. The investigators were not blinded to allocation during experiments and outcome assessment.

## Reporting summary

Further information on research design is available in the Nature Portfolio Reporting Summary linked to this article.

## Data availability

All data supporting the findings of this study are available within the paper. MS data have been deposited in ProteomeXchange with primary accession code PXD053068. All sequencing datasets have been deposited at the Sequence Read Archive BioProject PRJNA1124425 and Biosample SAMN41852702, SAMN41852703 and SAMN41852704. All other data supporting the findings of this study are available from the corresponding authors on reasonable request. Source data are provided with this paper.

## Code availability

The codes for genomics and proteomics analysis are available at GitHub at https://github.com/Cuixiaojian21/cGAS_script.

## Acknowledgements

We thank M. Ding (UT Southwestern Medical Center), Z. Zi (UT Southwestern Medical Center), X. Wang (UT Southwestern Medical

Center) for providing technical support and helpful discussions. We thank Y. Yu (UT Southwestern Medical Center) and H. Yu (UT Southwestern Medical Center) for helpful discussions and suggestions. We thank B. Wang (University of Washington) and Y. He (Jingwan Biomedical Tech) for revising the paper. We also thank the Biomedical Core Facility of the Medicine School at the Chinese University of Hong Kong, Shenzhen for providing technical support with the MS tests. This work was supported by grants from the Guangdong Basic and Applied Basic Research Foundation (202(4A1515030060 to L.Y.), Chinese National Natural Science Foundation Projects (81602422 to L.Y. and 81903094 to X.K.) and Ningbo Natural Science Foundation (2023J030 to H.D.).

## Author contributions

L.Y. conceived the study, designed, performed and interpreted the experiments, and wrote the paper. Most of the experiments were performed by L.Y., L.L. and Q.W. Q.Z., X.L. and Q.Y. developed and performed bioinformatics and statistical analysis and revised the paper. X.K. and H.D. constructed all plasmids and cell lines. G.L. generated the mouse model. S.W., F.Y. and H.C. revised the paper.

## Competing interests

The authors declare no competing interests.

## Additional information

**Extended data** is available for this paper at https://doi.org/10.1038/s41556-024-01473-0.

**Correspondence and requests for materials** should be addressed to Song Wu or Lin Yuan.

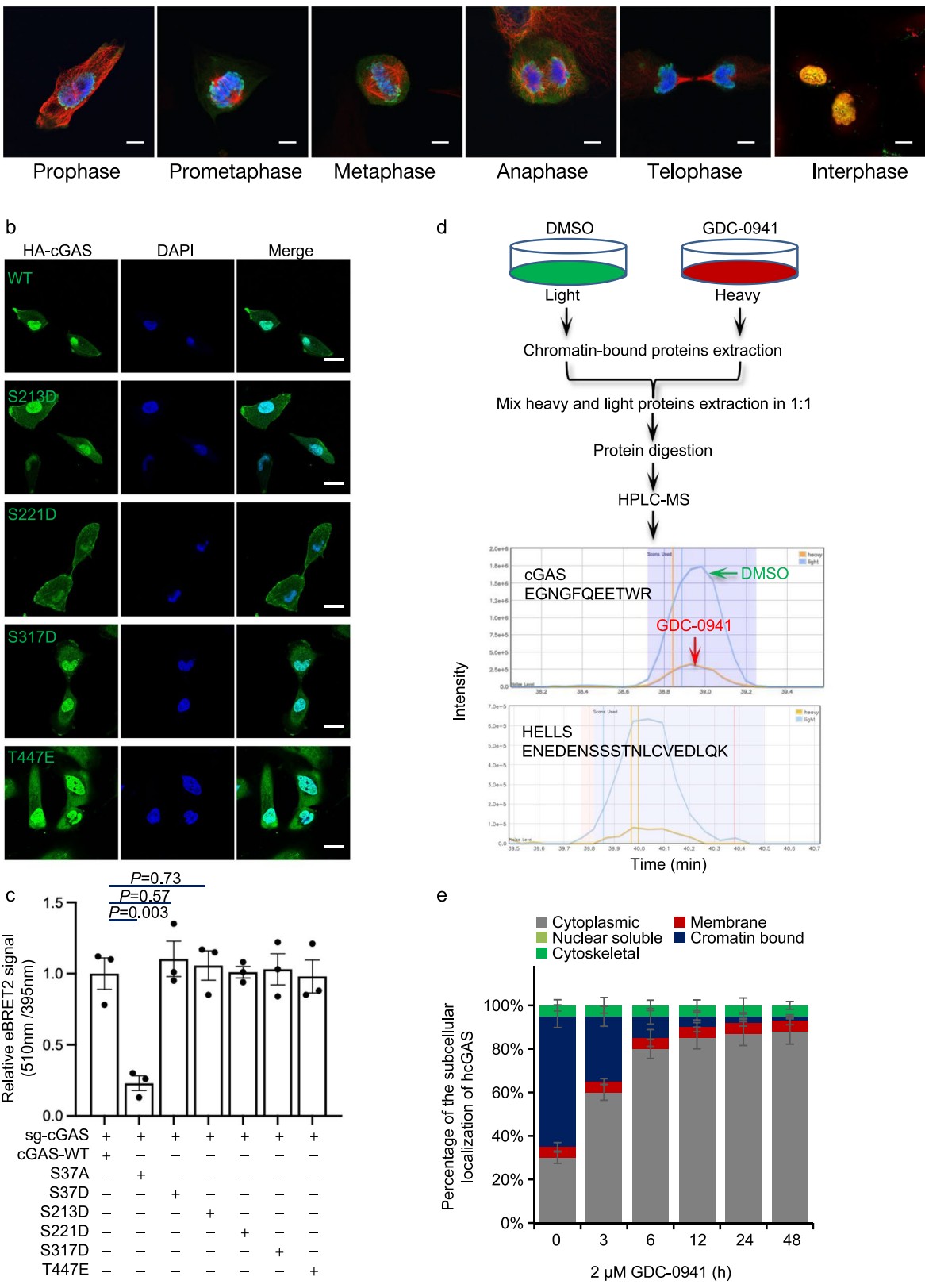

**Extended Data Fig. 1 | See next page for caption.**

Article

**Extended Data Fig. 1 | PI3K-mTOR pathway inhibition reduces cGAS chromatin localization. a**. Immunofluorescence analysis of cGAS localization (green) in HCT116 cells throughout the cell cycle as defined by DAPI staining (blue). Scale bar, 5 µm. **b**. Immunofluorescence analysis of subcellular localization of cGAS mutants versus wild-type in cGAS knockout HCT116 cells. Scale bar, 10 µm. **c**. eBRET2 signal measured the chromatin binding affinity of cGAS mutants versus wild-type, as indicated by the interaction between H2A-Rluc8 and cGAS-GFP following transfection into cGAS knockout HCT116 cells. **d**. SILAC proteomics identified cGAS and HELLS as proteins whose chromatin association is most sensitive to treatment with the PI3K inhibitor GDC-0941. The and Heavy (treated) and light (control) peptide signals are shown in yellow and blue, respectively. **e**. ELISA quantification of cGAS protein levels in different cell fractions from HCT116 cells treated with GDC-0941 for the indicated times. Fractionation was performed using ProteoExtract® Kit and Chromatin Extraction Kit. Data are means ± SEM of three independent experiments. Data are shown as means ± s.e.m. from three independent experiments (**c**, **e**). Unpaired two-tailed $t$-test. Experiments were repeated three times (**a**, **b**) with similar results. Numerical source data are available as source data.

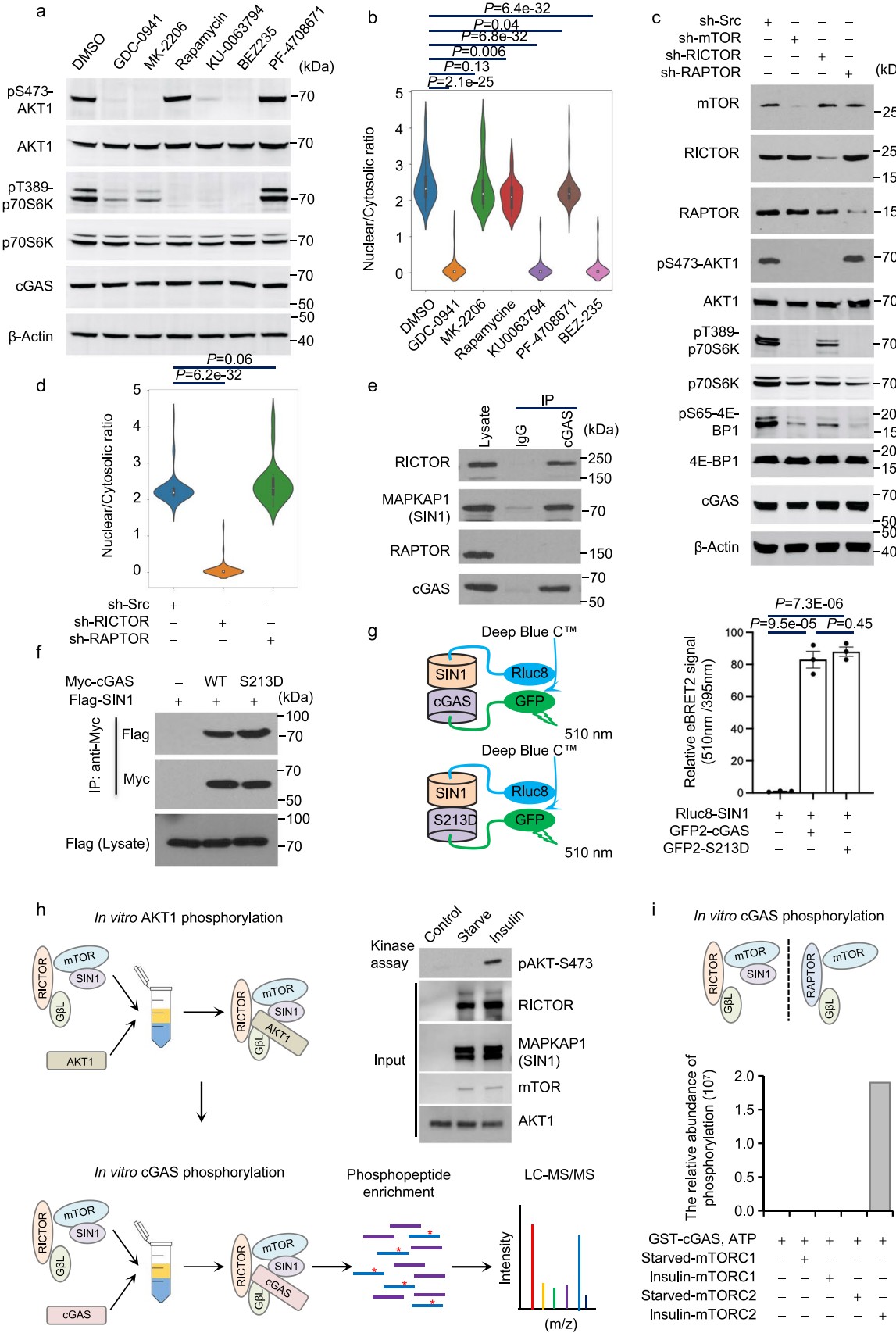

Extended Data Fig. 2 | See next page for caption.

**Extended Data Fig. 2 | mTORC2 induces cGAS chromatin localization.**
**a**. Immunoblot analysis of whole-cell lysates (WCLs) from HCT116 cells treated with PI3K-mTOR pathway inhibitors (2 μM, 12 hours). **b**. Automated quantification of cGAS intensities in individual cell nuclei and cytosol from (B) using CellProfiler. Nuclear/cytosolic ratios are plotted as violin plots with mean for each condition. *n* = 50 from 3 independent experiments. Unpaired two-tailed *t*-test. **c**. Immunoblot analysis of whole cell lysates from HCT116 cells lentivirally infected with scrambled (sh-Src) or shRNAs targeting mTOR components. **d**. Immunofluorescence analysis and quantification cGAS subcellular distribution in HCT116 cells lentivirally infected with the scramble control or gene-targeted shRNAs. Nuclear/cytosolic ratios are plotted as violin plots with means for each condition. *n* = 50 from 3 independent experiments. Unpaired two-tailed *t*-test. **e**. Co-immunoprecipitation and immunoblot analysis of cGAS

and mTORC2 component interaction in HCT116 cells. **f**. Co-immunoprecipitation of Myc-tagged cGAS mutants from HCT116 cells transfected with the indicated constructs. **g**. eBRET2 signal reporting the interaction between cGAS and SIN1 (mTORC2 component) in HCT116 cells expressing a GFP2-cGAS and Rluc8-SIN1 biosensor. Bars represent the means ± s.e.m. from three independent experiments. Unpaired two-tailed *t*-test. **h**. Schematic of the workflow establishing an *in vitro* cGAS phosphorylation system using purified mTORC2 complexes and AKT1 as a control substrate. **i**. Mass spectrometry analysis of *in vitro* cGAS phosphorylation by purified mTORC1 or mTORC2 complexes from starved (48 h) or insulin-stimulated (30 min with 100 nM insulin) HCT293T cells. Experiments were repeated three times (**a, c, e, f, h**) with similar results. Numerical source data and unprocessed blots are available as source data.

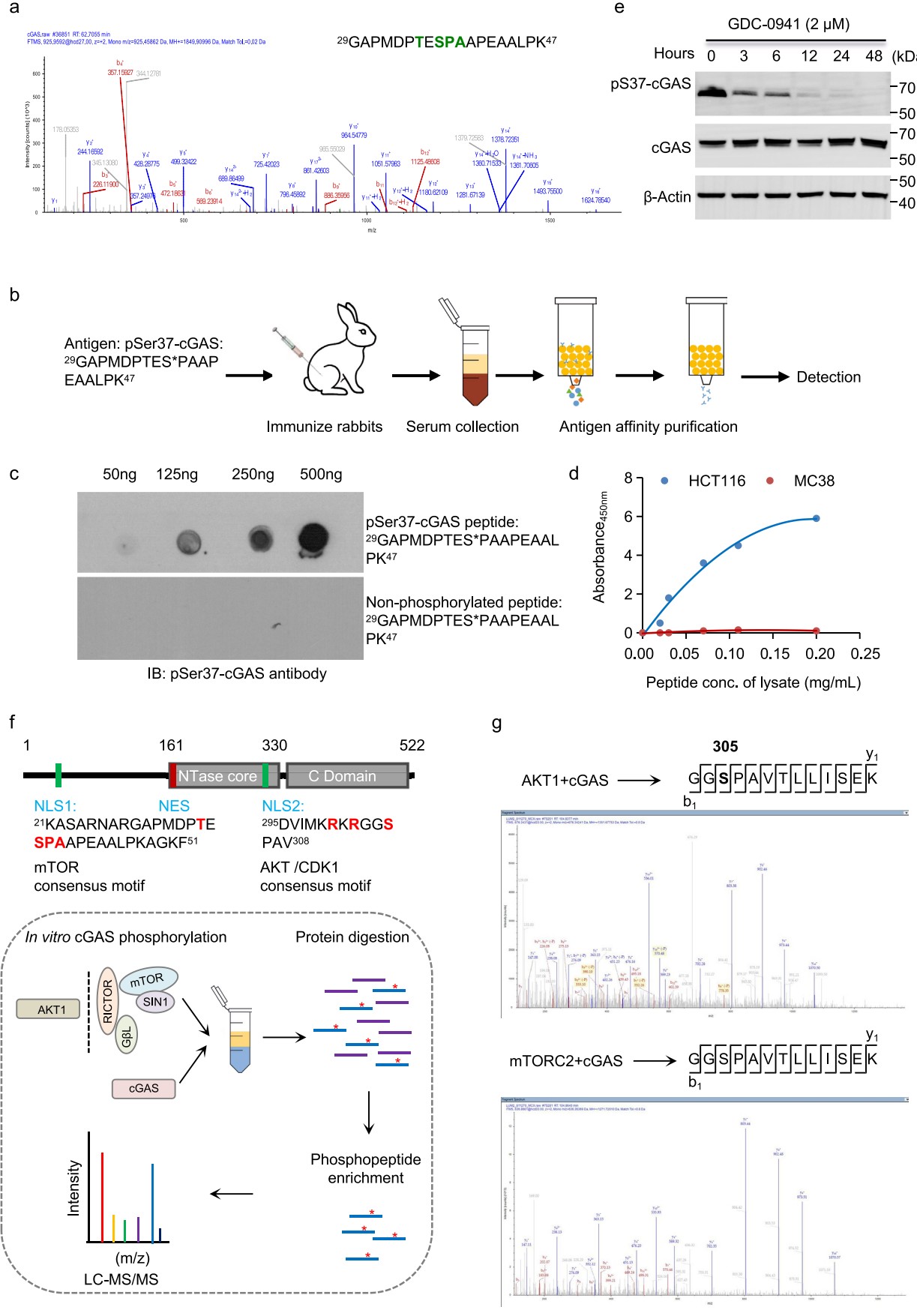

**Extended Data Fig. 3 | See next page for caption.**

**Extended Data Fig. 3 | mTORC2 induces cGAS phosphorylation at serine 37. a.** *In vitro* cGAS phosphorylation assays by means of LC–MS/MS showed that purified mTORC1 could not directly phosphorylate recombinant His-cGAS. **b**. Generation of a pSer37-cGAS polyclonal antibody by immunizing rabbits with a phosphorylated peptide antigen and affinity purifying the antibodies. The antigen sequence used for immunization was cGAS aa 29-47 (GAPMDPTES*PAAPEAALPK). S* stands for phosphorylated serine residue in these synthetic peptides. The antibodies were affinity purified using the antigen peptide column, but they were not counter selected on unmodified antigen. **c**. Dot blot analysis of antibody specificity using serial dilutions of phosphorylated and non-phosphorylated cGAS peptides. This experiment was performed three times with similar results. **d**. Correlation between cell lysate protein concentration and absorbance signal in a pSer37-cGAS ELISA using HCT116 and MC38 cell extracts, demonstrating cell type-specific expression. This experiment was performed three times with similar results. **e**. HCT116 cells were treated with GDC-0941 for times indicated, then WCLs were analysed by immunoblot (IB) with the antibodies shown. This experiment was performed three times with similar results. **f**. Purified mTORC2 or AKT1 was incubated with recombinant His-cGAS to establish *in vitro* cGAS phosphorylation system, and the phosphorylation sites were analysed by means of LC–MS/MS. **g**. *In vitro* cGAS phosphorylation assays by means of LC–MS/MS showed that purified AKT1, but not mTORC2, directly phosphorylated cGAS at serine 305 *in vitro*. Unprocessed blots are available as source data.

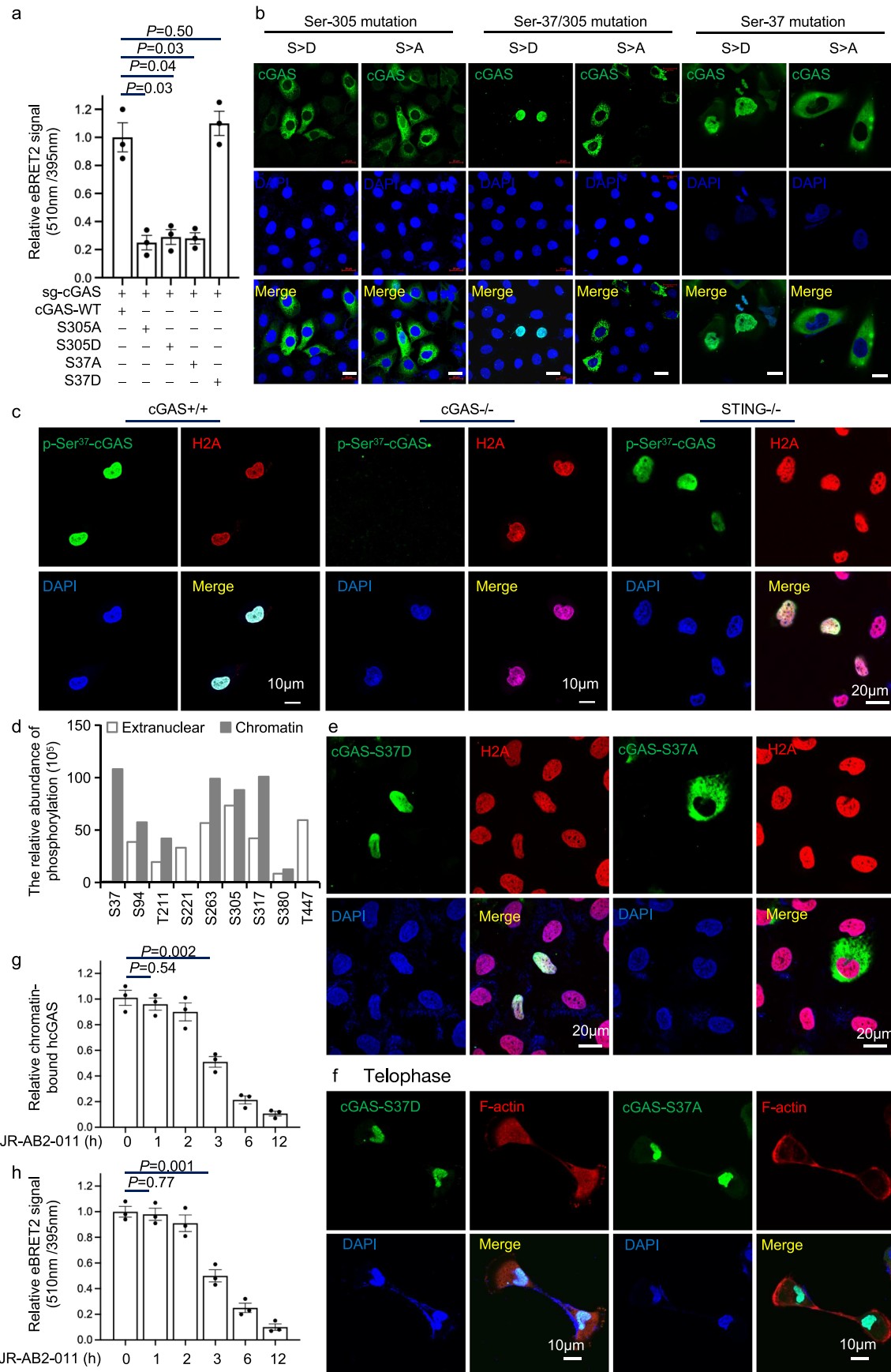

**Extended Data Fig. 4 | See next page for caption.**

**Extended Data Fig. 4 | Serine 37 phosphorylation promotes cGAS chromatin localization. a**. eBRET2 signal reporting on chromatin (H2A) binding of cGAS mutants following tansfection into cGAS knockout HCT116 cells. **b**. Immunofluorescence analysis of subcellular localization of cGAS S37A and S37D mutants re-expressed by lentiviral transduction in cGAS-knockout HCT116 cells. Scale bar, 20 μm. **c**. Immunofluorescence analysis of p-Ser[37]-cGAS (serine 37 phosphorylation) and H2A (chromatin marker) subcellular distribution in HCT116 cells. **d**. cGAS proteins were immunoprecipitation from extranuclear and intranuclear (chromatin-bound) fractions of HCT116 cells, and the phosphorylation sites were analysed by means of LC−MS/MS analysis. **e**. Immunofluorescence analysis of extranuclear versus intranuclear localization of cGAS S37D and S37A mutants re-expressed in cGAS-knockout HCT116 cells. **f**. Immunofluorescence analysis of cGAS S37D and S37A mutant localization in mitotic HCT116 cells, showing chromatin binding capacity. **g**. ELISA quantification of cGAS protein levels in chromatin fractions isolated using a Chromatin Extraction Kit from STING-/- HCT116 cells treated with 10 μM JR-AB2-011 for indicated times. **h**. eBRET2 signal measured the interaction between H2A-Rluc8 and cGAS-GFP in STING-/- HCT116 cells treated with 10 μM JR-AB2-011 for indicated times. Data are shown as means ± s.e.m. from three independent experiments (**a, g, h**). Unpaired two-tailed *t*-test. Experiments were repeated three times (**b, c, e, f**) with similar results. Numerical source data are available as source data.

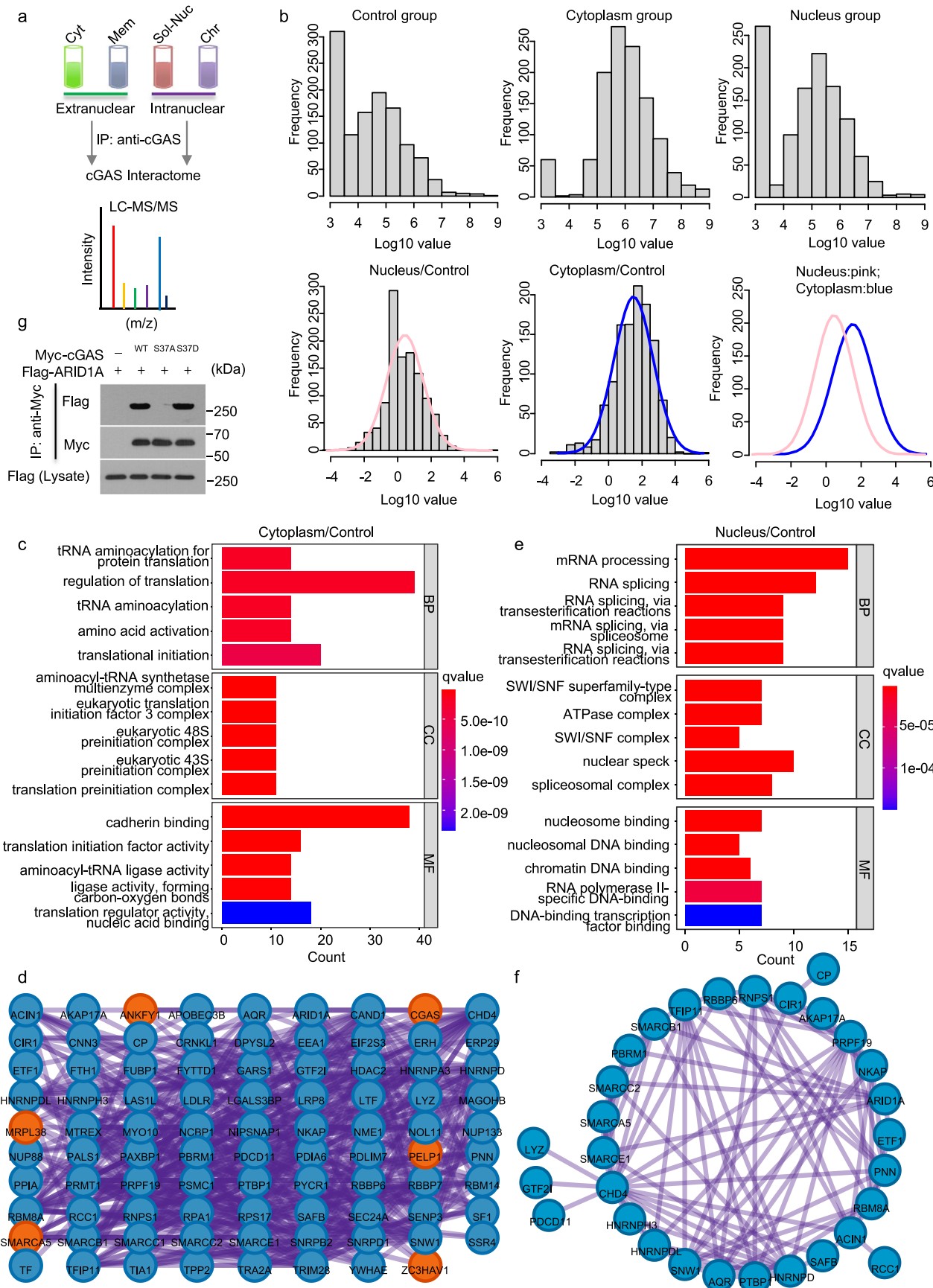

**Extended Data Fig. 5 | See next page for caption.**

**Extended Data Fig. 5 | Interactome analysis identifies SWI/SNF complex recruitment by ccGAS. a**. Schematic of experimental workflow for identification of cGAS interacting proteins. ProteoExtract®Kit and Chromatin Extraction Kit were used to obtain proteins of intracellular and extracellular cell fractions, and then the interaction proteins of cGAS in different cell fractions were obtained by immunoprecipitation. cGAS interactome was analysed by means of LC–MS/MS. **b**. LC–MS/MS analysis detected the binding proteins of cGAS in different cell fractions, and the frequency distribution histogram described the signal distribution. By comparing with the control group, proteins that significantly interact with cGAS in cytoplasm and nucleus could be screened out. **c**. Gene Ontology enrichment of cGAS interacting proteins in the cytoplasm. **d**. PPI network of 102 nuclear cGAS interactors including 89 nodes and 451 interactions annotated from public databases. among which 6 are cGAS and their Direct interactors are highlighted in orange. **e**. Gene Ontology enrichment of cGAS interacting proteins in the nucleus. **f**. PPI network of 33 unique intranuclear cGAS interactors including 30 nodes and 95 interactions, showing enrichment for SWI/SNF complex components. **g**. Co-immunoprecipitation and immunoblot validation of ARID1A binding to indicated cGAS mutants. This experiment was performed three times with similar results. Statistics source data and unprocessed blots are available as source data.

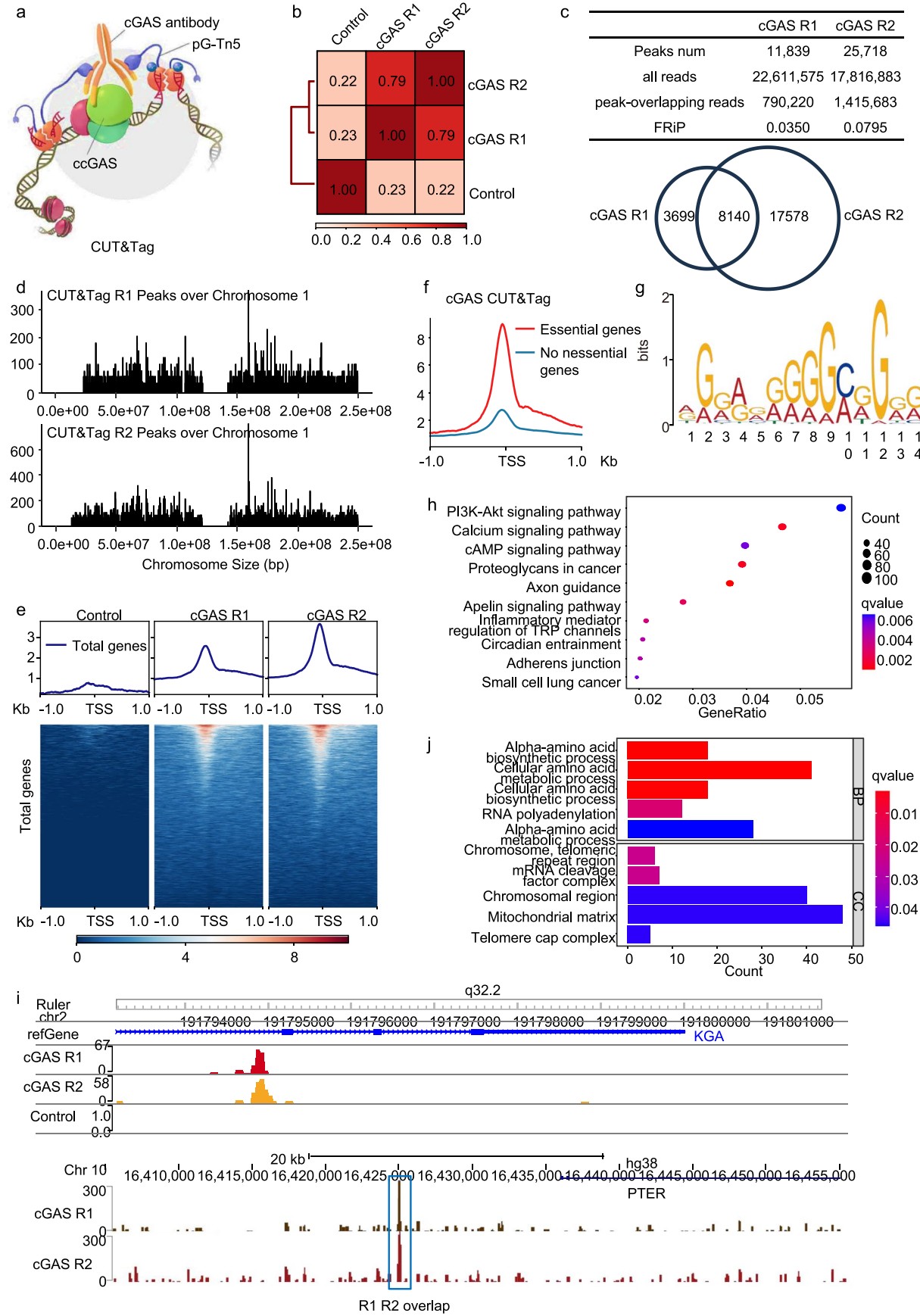

Extended Data Fig. 6 | See next page for caption.

**Extended Data Fig. 6 | CUT&Tag profiling reveals genes regulated by ccGAS associated with cell cycle and amino acid metabolism. a.** Schematic of CUT&Tag workflow to map genomic cGAS binding sites. **b.** Spearman correlation heatmap comparing biological replicates. **c.** Scatter plot and correlation of peak calls between technical replicates, demonstrating reproducibility. **d.** Visualization of cGAS CUT&Tag signals on Chromosome 1. **e.** Metagene analysis showing cGAS CUT&Tag enrichment at transcription start sites (TSS). **f.** Genomic distribution of cGAS peaks in relation to essential genes with necessary for cell survival. **g.** Motif enrichment analysis within top 500 cGAS-bound regions sorted by signal intensity. **h.** KEGG pathway analysis of all cGAS CUT&Tag peaks. **i.** Visualization of cGAS CUT&Tag signals at the promoter of *KGA* (chr2: 191,792,416–191,830,270) and *PTER* (chr10: 16,401,685–16,456,017). **j.** Gene Ontology enrichment of genes with cGAS CUT&Tag peaks in promoter regions. Statistics source data are available as source data.

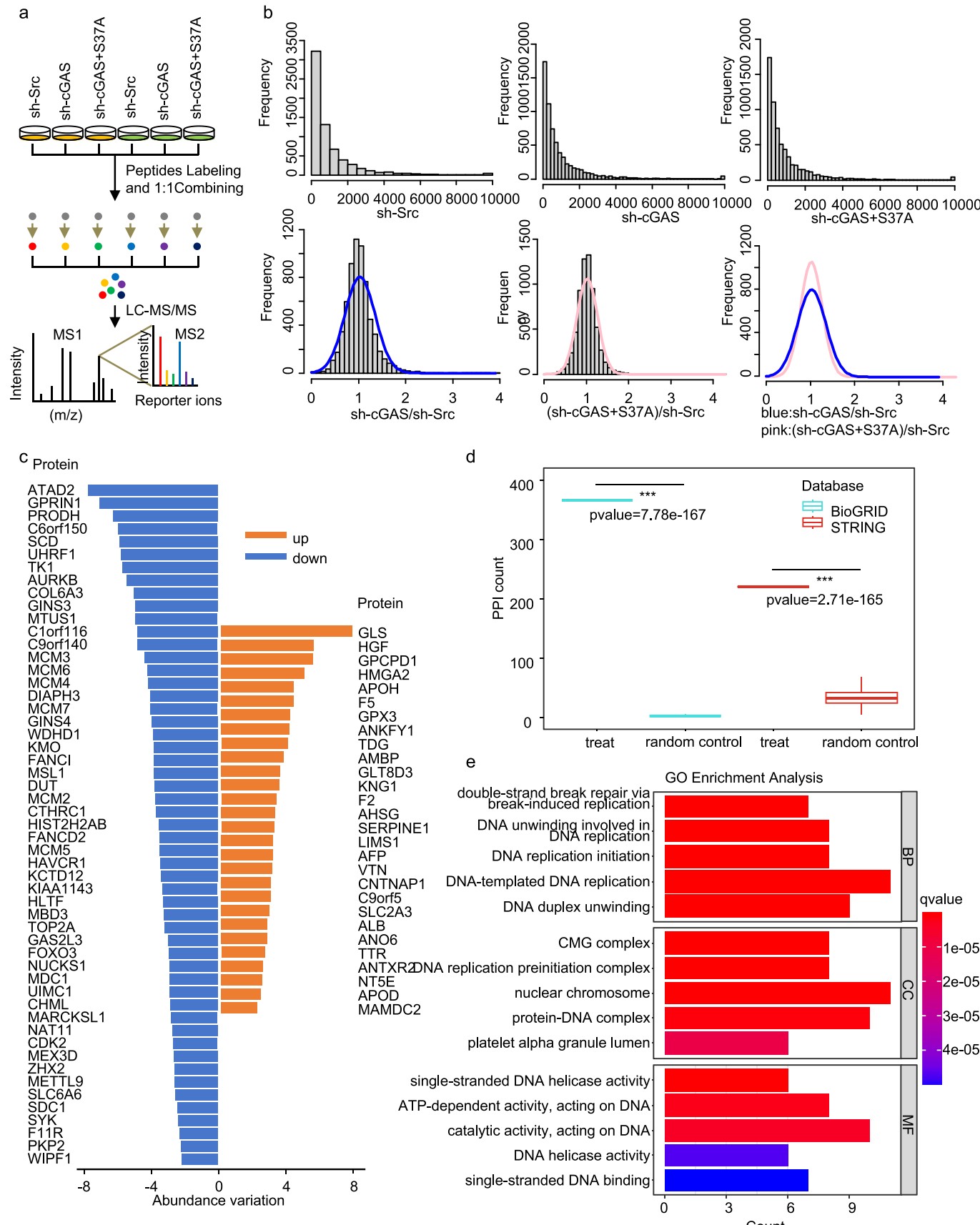

**Extended Data Fig. 7 | See next page for caption.**

**Extended Data Fig. 7 | TMT proteomics identifies DNA replication proteins and kidney-type glutaminase regulated downstream of ccGAS. a**. Schematic of experrimental workflow for identification of ccGAS downstream proteins. Proteins from HCT116 cells lentivirally infected with the indicated shRNAs or cGAS-S37A mutants were labelled with TMT reagents and analysed by LC–MS/MS. **b**. Protein signal distribution after cGAS knockdown determined by tandem mass tag (TMT). **c**. After cGAS knockdown, 81 proteins with significant changes in abundance were quantitatively screened from 6523 proteins, of which 28 were upregulated (orange) and 53 were downregulated (blue). **d**. The interaction frequency of 81 proteins with significant changes in abundance was significantly higher than that of randomized controls. **e**. Gene Ontology enrichment of 53 downregulated proteins, associated with DNA replication. Statistics source data are available as source data.

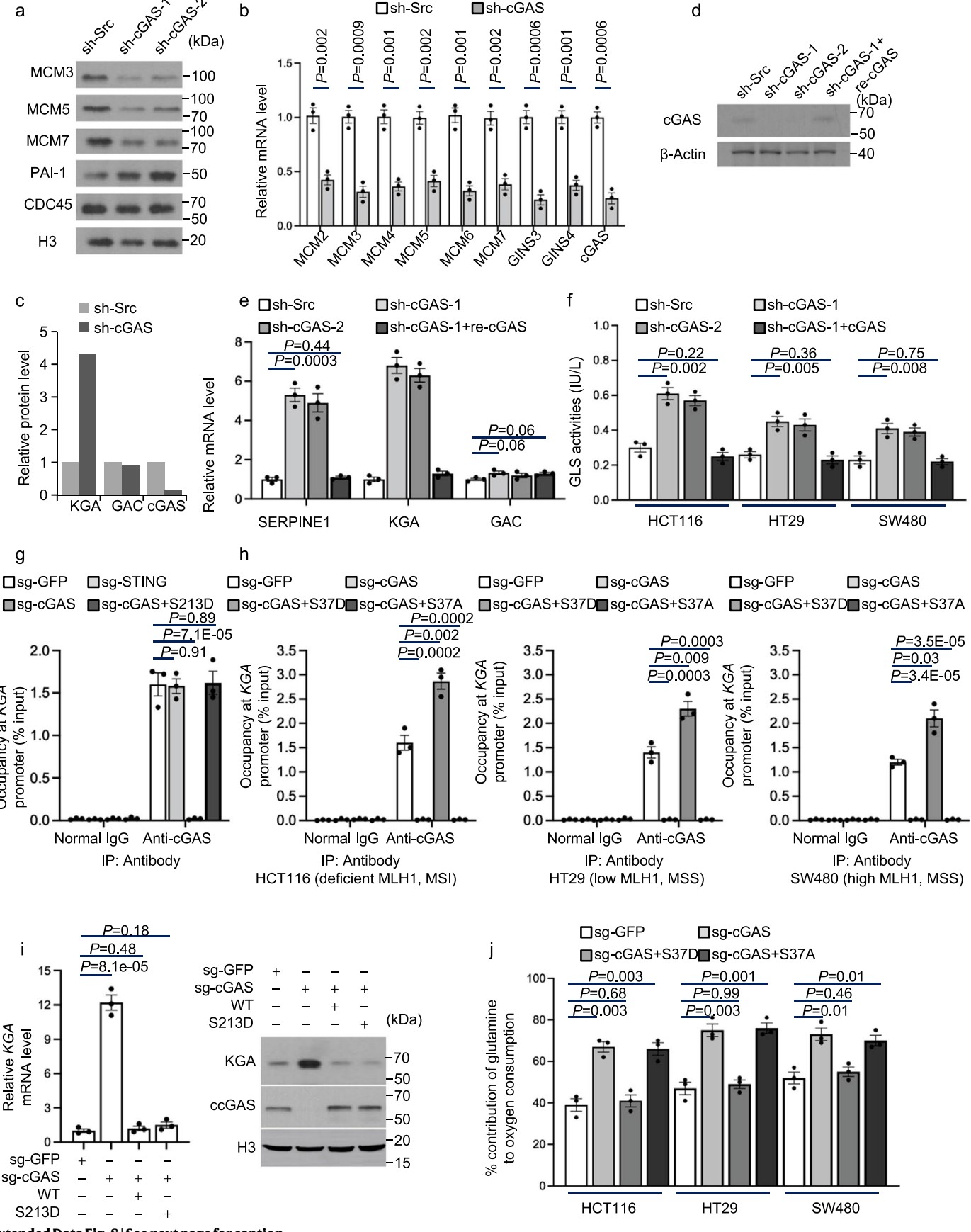

Extended Data Fig. 8 | See next page for caption.

**Extended Data Fig. 8 | ccGAS depletion regulates expression of the CDC45-MCM-GINS (CMG) complex and KGA. a**. Immunoblot analysis of whole cell lysates from HCT116 cells infected with cGAS shRNA or control lentiviruses. **b**. qPCR analysis of indicated gene expression normalized to ACTB in cGAS knockdown HCT116 cells. **c**. TMT proteomics quantification of KGA and GAC protein levels in cGAS knockdown HCT116 cells. **d**. Immunoblot analysis of WCL derived from cGAS-knockdown HCT116 cells infected with lentiviruses encoding shRNA-resistant cGAS. **e**. qPCR analysis of indicated gene expression in cGAS rescued knockdown HCT116 cells. **f**. Glutaminase activity assay of parental and cGAS knockout cells expressing cGAS mutants. **g**. ChIP–qPCR analysis of cGAS occupancy at *KGA* promoter in indicated HCT116 cells. **h**. ChIP–qPCR analysis of cGAS occupancy at the *KGA* promoter in cGAS knockout colorectal cancer cells stably expressing indicated cGAS mutants. **i**. Wild-type and S213D mutant cGAS were transfected into cGAS-knockout HCT116 cells, and *KGA* mRNA levels and chromatin-bound proteins were analysed. **j**. Contribution of glutamine to oxygen consumption in indicated cell lines analysed by Seahorse. Data are shown as means ± s.e.m. from three independent experiments (**b, e–j**). Unpaired two-tailed *t*-test. Experiments were repeated three times (**a, d**) with similar results. Numerical source data and unprocessed blots are available as source data.

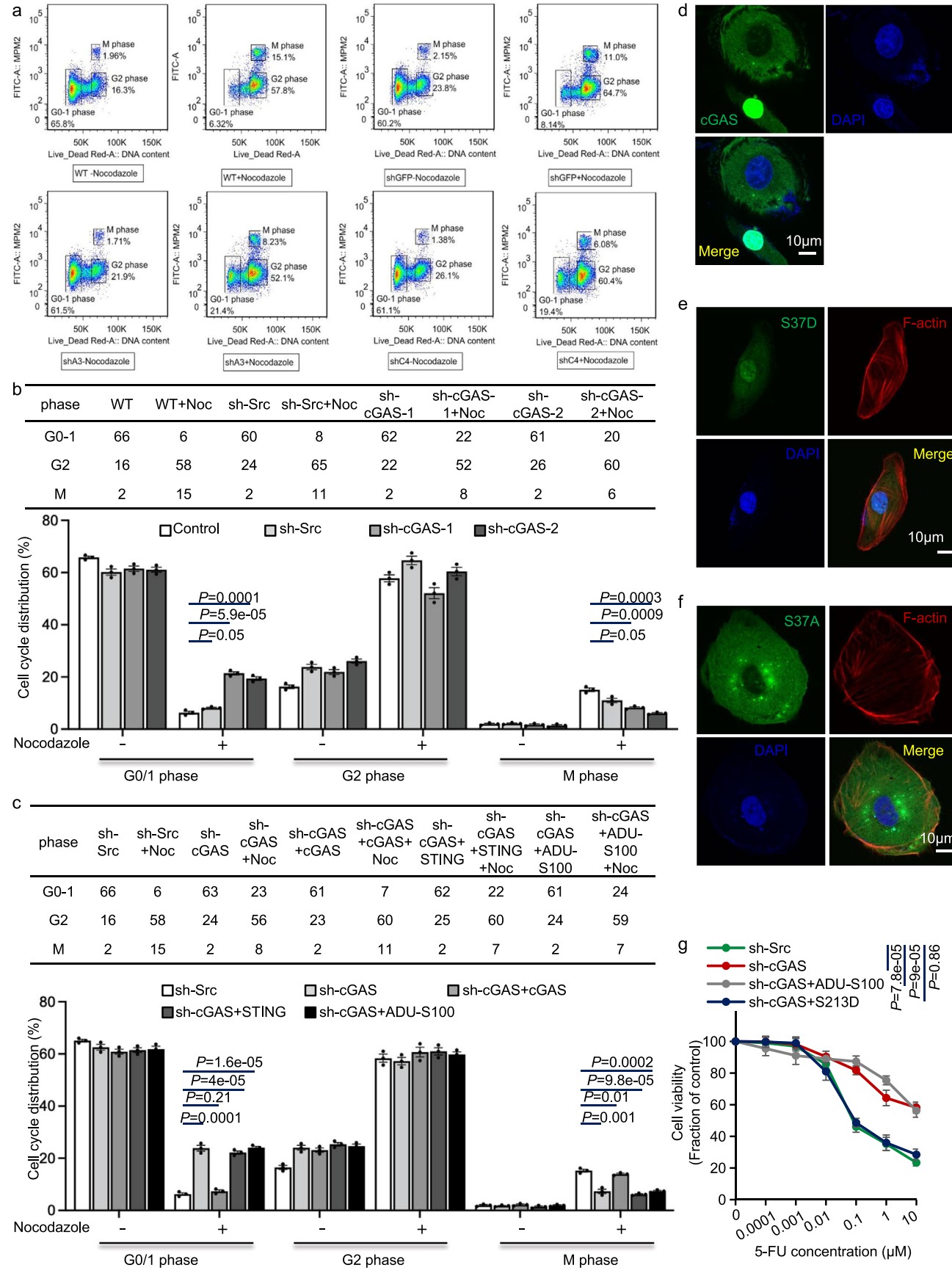

**Extended Data Fig. 9 | See next page for caption.**

**Extended Data Fig. 9 | ccGAS depletion induces diapause-like state and chemoresistance in colorectal cancer cells.** (**A**) Flow cytometric analysis of cell cycle profiles in HCT116 cells lentivirally infected with the indicated shRNAs. (**B**) Quantification of cell cycle distributions in HCT116 cells lentivirally infected with the indicated shRNAs. (**C**) cGAS or STING was transfected into HCT116 cells stably expressing cGAS shRNA. Quantification of cell cycle distributions in cGAS knockdown HCT116 cells stably expressing cGAS, STING, or treated with STING activator ADU-S100 (10 μM). (**D**) cGAS-S37D/A mutants were transfected into cGAS-knockout HCT116 cells, the subcellular distribution of cGAS mutants was analysed by immunofluorescence, and the morphology of the corresponding cells was observed. (**E**) Immunofluorescence analysis of cell morphology in cGAS-knockout HCT116 cells with cGAS-S37D mutants restoration. F-actin stain used to label cytoskeleton. (**F**) Immunofluorescence analysis of cell morphology in cGAS-knockout HCT116 cells with cGAS-S37A mutants restoration. (**G**) Cell viability assays measuring 5-FU response in cGAS knockdown HCT116 cells stably expressing shRNA-resistant cGAS-S213D mutants or pretreated with 10 μM ADU-S100 for 6 hours. Data are shown as means ± s.e.m. from three independent experiments (**b, c, g**). Unpaired two-tailed *t*-test (**b, c**) and Kruskal–Wallis one-way ANOVA followed by Dunn's multiple comparison tests (**g**). Experiments were repeated four times (**d–f**) with similar results. Numerical source data are available as source data.

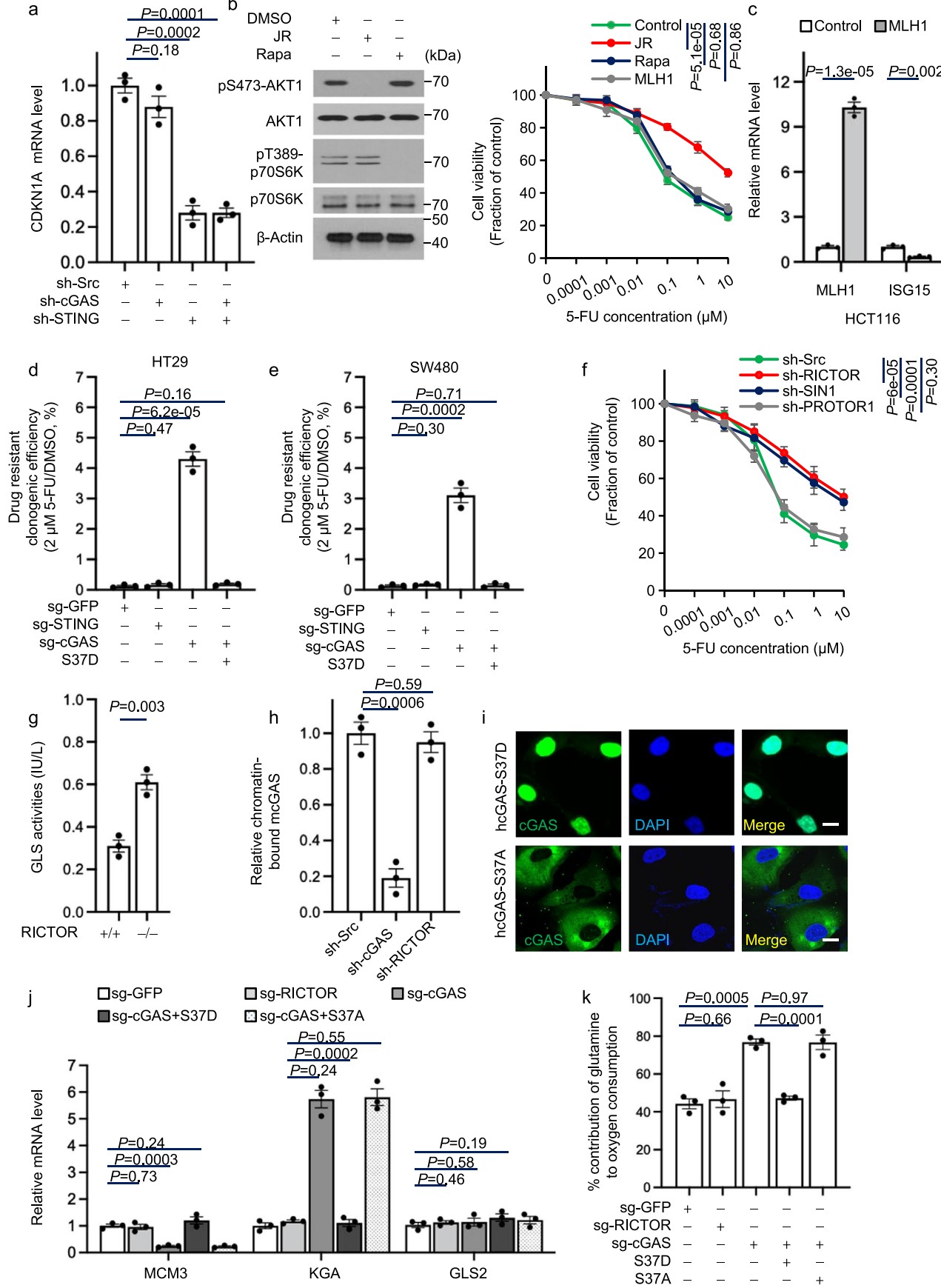

**Extended Data Fig. 10 | See next page for caption.**

**Extended Data Fig. 10 | The ccGAS-KGA signalling axis is conserved in murine colorectal cancer cells. a**. CDKN1A mRNA levels were quantified by qPCR analysis and normalized to ACTB control in HCT116 cells stably expressing indicated shRNAs. **b**. Cell viability assays measuring 5-FU response in HCT116 cells stably transfected MLH1 or pretreated with 10 μM JR-AB2-011 for 6 hours. JR (JR-AB2-011), a selective mTORC2 inhibitor. **c**. qPCR analysis of indicated gene expression in HCT116 cells stably transfected MLH1. **d**. Colony formation assays measuring drug response of HT29 cells. **e**. Colony formation assays measuring drug response of SW480 cells. **f**. HCT116 cells stably expressing indicated shRNAs were treated with the indicated concentrations of 5-FU and cell viability was assessed after 24 hours. **g**. Glutaminase activity assay of PDX models (RICTOR+/+ and RICTOR−/−). **h**. ELISA quantification of cGAS protein levels in chromatin fractions isolated using a Chromatin Extraction Kit from MC38 cells stably expressing indicated shRNAs. **i**. Immunofluorescence analysis of human cGAS S37A or S37D mutant localization in MC38 cells. Scale bar, 10 μm. **j**. qPCR analysis of gene expression in indicated MC38 cells normalized to ACTB. **k**. Contribution of glutamine to oxygen consumption in the indicated MC38 cells analysed by Seahorse. Data are shown as means ± s.e.m. from three independent experiments (**a**–**h**, **j**, **k**). Unpaired two-tailed *t*-test (**a**, **c**–**e**, **g**, **h**, **j**, **k**) and Kruskal–Wallis one-way ANOVA followed by Dunn's multiple comparison tests (**b**, **f**). Experiments were repeated four times (**i**) with similar results. Numerical source data and unprocessed blots are available as source data.

# Reporting Summary

## Statistics

For all statistical analyses, confirm that the following items are present in the figure legend, table legend, main text, or Methods section.

| n/a | Confirmed | |
|---|---|---|
| ☐ | ☒ | The exact sample size ($n$) for each experimental group/condition, given as a discrete number and unit of measurement |
| ☐ | ☒ | A statement on whether measurements were taken from distinct samples or whether the same sample was measured repeatedly |
| ☐ | ☒ | The statistical test(s) used AND whether they are one- or two-sided *Only common tests should be described solely by name; describe more complex techniques in the Methods section.* |
| ☐ | ☒ | A description of all covariates tested |
| ☐ | ☒ | A description of any assumptions or corrections, such as tests of normality and adjustment for multiple comparisons |
| ☐ | ☒ | A full description of the statistical parameters including central tendency (e.g. means) or other basic estimates (e.g. regression coefficient) AND variation (e.g. standard deviation) or associated estimates of uncertainty (e.g. confidence intervals) |
| ☐ | ☒ | For null hypothesis testing, the test statistic (e.g. $F$, $t$, $r$) with confidence intervals, effect sizes, degrees of freedom and $P$ value noted *Give P values as exact values whenever suitable.* |
| ☒ | ☐ | For Bayesian analysis, information on the choice of priors and Markov chain Monte Carlo settings |
| ☒ | ☐ | For hierarchical and complex designs, identification of the appropriate level for tests and full reporting of outcomes |
| ☒ | ☐ | Estimates of effect sizes (e.g. Cohen's $d$, Pearson's $r$), indicating how they were calculated |

*Our web collection on statistics for biologists contains articles on many of the points above.*

## Software and code

Policy information about availability of computer code

| Data collection | MS/MS data were collected by liquid chromatography-tandem mass spectrometry(LC-MS/MS); FACS data was collected by FACScalibur. |
|---|---|
| Data analysis | MS/MS data were searched against the reviewed UniProt Human protein database (version 20230119 containing 20,308 entries) using Mascot 2.5.1 (Matrix Science), and data analysis was performed using Scaffold 4.4.8 software (Proteome Software); FACS data analysis was performed with FlowJo software; All statistical analyses were performed with GraphPad Prism 9 and SPSS 19.0 software. |

For manuscripts utilizing custom algorithms or software that are central to the research but not yet described in published literature, software must be made available to editors and reviewers. We strongly encourage code deposition in a community repository (e.g. GitHub). See the Nature Portfolio guidelines for submitting code & software for further information.

## Data

Policy information about availability of data

All manuscripts must include a data availability statement. This statement should provide the following information, where applicable:
- Accession codes, unique identifiers, or web links for publicly available datasets
- A description of any restrictions on data availability
- For clinical datasets or third party data, please ensure that the statement adheres to our policy

> All data generated or analyzed during this study are included in this article and its Supplementary Information.

## Research involving human participants, their data, or biological material

Policy information about studies with human participants or human data. See also policy information about sex, gender (identity/presentation), and sexual orientation and race, ethnicity and racism.

| | |
|---|---|
| Reporting on sex and gender | Tumor specimens were collected from 6 colorectal cancer males patients (ages 40-60 years) undergoing surgical resection at the Peking University Shenzhen Hospital. |
| Reporting on race, ethnicity, or other socially relevant groupings | Asian |
| Population characteristics | All patients had previously received 5-FU chemotherapy. |
| Recruitment | Patients did not receive compensation for providing samples. |
| Ethics oversight | Related experiments were approved by the Institutional Human Research Ethics and Animal Care and Use Committee at Peking University and Peking University Shenzhen Hospital. |

Note that full information on the approval of the study protocol must also be provided in the manuscript.

# Field-specific reporting

Please select the one below that is the best fit for your research. If you are not sure, read the appropriate sections before making your selection.

☒ Life sciences ☐ Behavioural & social sciences ☐ Ecological, evolutionary & environmental sciences

For a reference copy of the document with all sections, see nature.com/documents/nr-reporting-summary-flat.pdf

# Life sciences study design

All studies must disclose on these points even when the disclosure is negative.

| | |
|---|---|
| Sample size | Data of most experiments were collected from at least 3 independent samples. |
| Data exclusions | No data were excluded. |
| Replication | All attempts at replication were successful. |
| Randomization | After the establishment of Cgas-KO mouse model, the animals were randomly grouped for in vivo experiments. |
| Blinding | The investigator was blinded for in vivo experiments of all animal experiments. |

# Reporting for specific materials, systems and methods

We require information from authors about some types of materials, experimental systems and methods used in many studies. Here, indicate whether each material, system or method listed is relevant to your study. If you are not sure if a list item applies to your research, read the appropriate section before selecting a response.

## Materials & experimental systems

| n/a | Involved in the study |
|-----|----------------------|
| ☐ | ☒ Antibodies |
| ☐ | ☒ Eukaryotic cell lines |
| ☒ | ☐ Palaeontology and archaeology |
| ☐ | ☒ Animals and other organisms |
| ☒ | ☐ Clinical data |
| ☒ | ☐ Dual use research of concern |
| ☒ | ☐ Plants |

## Methods

| n/a | Involved in the study |
|-----|----------------------|
| ☒ | ☐ ChIP-seq |
| ☐ | ☒ Flow cytometry |
| ☒ | ☐ MRI-based neuroimaging |

# Antibodies

| | |
|---|---|
| Antibodies used | Antibodies used in immunoblotting: Anti-cGAS (CST, #15102, D1D3G, 1:1,000 dilution), anti-HELLS (CST, #7998, 1:1,000), anti-β-Actin (CST, #4967, 1:1,000), anti-PDI (CST, #2446, 1:1,000), anti-HA (CST, #3724, C29F4, 1:1,000), anti-Flag (CST, #14793, D6W5B, 1:1,000), anti-Histone H3 (CST, #9715, 1:1000), anti-ARID1A (CST, #12354, D2A8U, 1:1000), anti-MCM7 (CST, #3735, D10A11, 1:1000), anti-SMARCC2 (CST, #12760, D8O9V, 1:1000), anti-Akt (CST, #9272, 1:1,000), anti-pSer473-Akt (CST, #4060, D9E, 1:1,000), anti-MCM3 (CST, #4012, 1:1,000), anti-GLS1 (CST, #49363, E4T9Q, 1:1,000), anti-mTOR (CST, #2983, 7C10, 1:1,000), anti-RICTOR (CST, #2114, 53A2, 1:1,000), anti-RAPTOR (CST, #48648, E6O3A, 1:1,000), anti-SIN1 (CST, #12860, D7G1A, 1:1,000), anti-PAI-1 (CST, #11907, D9C4, 1:1,000), anti-BAF200 (CST, #82342, D8D8U, 1:1,000), anti-CDC45(CST, #11881, D7G6, 1:1,000), anti-p70 S6K (CST, #9202, 1:1,000), anti-pThr389-p70 S6K (CST, #9209, 1:1,000), anti-4E-BP1 (CST, #9452, 1:1,000), anti-pSer65-4E-BP1 (CST, #9451, 1:1,000), and anti-GAPDH (CST, #5174, D16H11, 1:2,000) were purchased from Cell Signaling Technology. Anti-KGA (Proteintech, 20170-1-AP, 1:1,000) and anti-GAC (Invitrogen, PA5-40134, 1:1,000) were purchased from Thermo Fisher Scientific. Anti-SMARCA4 (Sigma, MABE121, 1:1000), anti-MCM5 (Sigma, SAB1406111, 1:1,000) were purchased from Sigma-Aldrich.<br><br>Antibodies used in immunoprecipitation and immunofluorescence: Anti-cGAS (CST, #79978, E5V3W, IF, 1:200), anti-cGAS (CST, #31659, D3O8O, IP, 1:200), anti-HA (CST, #3724, C29F4, 1:200), anti-H2A (CST, #12349, D6O3A, 1:200), and Phalloidin (CST, #8953, 1:200) were purchased from Cell Signaling Technology. The polyclonal anti-pSer37-cGAS antibodies generated by ourselves were derived from rabbits. The antigen sequence used for immunization was cGAS aa29-47 (GAPMDPTES*PAAPEAALPK). S* stands for phosphorylated serine residue in these synthetic peptides. The antibodies were affinity purified using the antigen peptide column, but they were not counter selected on unmodified antigen. |
| Validation | The purchased antibodies were all validated by the manufactures in their specific data sheets. |

# Eukaryotic cell lines

Policy information about cell lines and Sex and Gender in Research

| | |
|---|---|
| Cell line source(s) | HCT116, HT29, SW480, MC38, HEK293T, MDA-MB-231, 786-0, HCC44 cells were purchased from ATCC |
| Authentication | Cell lines were authenticated by STR profiling. |
| Mycoplasma contamination | Cell lines are regularly screened to ensure the absence of mycoplasma contamination using MycoAlert Mycoplasma detection kit (lonza). |
| Commonly misidentified lines (See ICLAC register) | No commonly misidentified cell lines were used. |

# Animals and other research organisms

Policy information about studies involving animals; ARRIVE guidelines recommended for reporting animal research, and Sex and Gender in Research

| | |
|---|---|
| Laboratory animals | 6 weeks old BALB/c nude mice were used in this study |
| Wild animals | Wild animals are not involved in this study. |
| Reporting on sex | Findings apply to both sex of animals, there is no sex distinction in this study. |
| Field-collected samples | None used |
| Ethics oversight | All animal experiments were conducted in accordance with the 'Guide for the Care and Use of Laboratory Animals' and 'Principles for the Utilization and Care of Vertebrate Animals', and were approved by the Institutional Human Research Ethics and Animal Care and Use Committee at Peking University and Peking University Shenzhen Hospital. Mice were monitored daily and experiments were terminated if tumor diameters exceeded 15 mm or volumes exceeded 2,000 mm3, in accordance with guidelines established by our Institutional Animal Care and Use Committee to prevent unnecessary suffering. |

Note that full information on the approval of the study protocol must also be provided in the manuscript.

# Plants

**Seed stocks**

Seed stocks are not involved in this study.

**Novel plant genotypes**

Novel plant genotypes are not involved in this study.

**Authentication**

not involved in this study.

# Flow Cytometry

## Plots

Confirm that:

☒ The axis labels state the marker and fluorochrome used (e.g. CD4-FITC).

☒ The axis scales are clearly visible. Include numbers along axes only for bottom left plot of group (a 'group' is an analysis of identical markers).

☒ All plots are contour plots with outliers or pseudocolor plots.

☒ A numerical value for number of cells or percentage (with statistics) is provided.

## Methodology

**Sample preparation**

Cells were incubated with 33 µM BrdU for 20 min, collected and centrifuged at 250G for 10 min. The pellet was re-suspended in 750µL of PBS 1X and fixed by adding 2250µL of ice-cold (−20°C) pure ethanol dropwise while vortexing. Samples were washed once in 1%BSA/PBS and re-suspended in 1 mL of 2N HCl and incubated for 25 min at room temperature allowing DNA denaturation. Then 3mL of 0.1M Sodium Borate (pH 8.5) was added to neutralize the acidic pH of the HCl solution and samples were incubated at room temperature for 2 min, centrifuged and washed twice in 1%BSA/PBS. Samples were then transferred in Eppendorf tube and centrifuged at 800G for 5 min. Pellets were re-suspended in 100µL of pure anti-BrdU antibody (life-technologies) diluted 1:5 in 1%BSA/PBS and incubated for 1 h at room temperature in the dark.    Samples were washed with 1%BSA/PBS and re-suspended in 100µL of anti-mouse FITC (life-technologies) diluted 1:50 in 1%BSA/PBS for 1 h at room temperature in the dark. After washing once with 1%BSA/PBS pellets were re-suspended in 1 mL of Propidium Iodate (PI) (2.5 µg/mL) and RNase (250 µg/mL) (RibonucleaseA from bovine pancreas, Sigma) and incubated overnight at 4°C.

**Instrument**

FACScalibur

**Software**

FlowJo software

**Cell population abundance**

Cell population abundance is not involved in this study

**Gating strategy**

The gating strategy for cell cycle analysis by flow cytometry:
1.Forward scatter area (FSC-A) vs Side scatter area (SSC-A) - Gate on the main cell population to exclude debris and clumps.
2.Doublet discrimination - Gate on single cells by plotting FSC-Width vs FSC-Area/Height. This will exclude doublets or clusters of cells.
3.DNA content - Plot the DNA-intercalating fluorescent dye (e.g. propidium iodide, DAPI) on the y-axis. This allows visualization of cells in G0/G1, S, and G2/M phases.
4.G0/G1 peak - Draw a gate around the first peak/largest population of cells with 2N DNA content (G0/G1 phase).
5.S phase - Identify cells between the G0/G1 and G2/M peaks as being in S phase of DNA replication.
6.G2/M peak - Draw a gate around the second peak of cells with 4N DNA content (G2/M phase).
7.Determination of percentages - Use gating and statistics tools to determine the percentage of cells in each phase of the cell cycle (G0/G1, S, G2/M).
The key steps are removing doublets and debris, then gating on the DNA content peaks to classify cells into their respective phases based on their DNA ploidy. Proper controls and compensation are also important for accurate cell cycle analysis.

☒ Tick this box to confirm that a figure exemplifying the gating strategy is provided in the Supplementary Information.

