## [Peer Review File · Nature Cell Biology]

Peer Review Information

Journal: Nature Cell Biology

Manuscript Title: mTORC2-driven chromatin cGAS mediates chemoresistance through epigenetic reprogramming in colorectal cancer

Corresponding author name(s): Lin Yuan

Editorial Notes:

Reviewer Comments & Decisions:

Decision Letter, initial version:
--

Dear Professor Yuan,

Your manuscript "mTORC2-driven chromatin cGAS mediates chemoresistance in colorectal cancer", has now been seen by 3 referees, who are experts in mTORC biology and function(referee 1); cGAS biology, cancer and therapy resistance, DNA replication (referee 2); and cGAS biology (referee 3), and whose comments are pasted below. In light of their advice, we regret that we cannot offer to publish the study in Nature Cell Biology.

As you will see, although the reviewers find this work interesting, they raise serious concerns that question the strength of the data and of the novel conclusions that can be drawn at this stage.

We would be open to the possibility of considering a revised manuscript that would fully address the referee concerns. However, any decision to re-review such a revised study would depend on the strength of the revisions and the published literature at the time of resubmission. In addition, please note that this will be logged in and processed as an appeal, and whether or not to file an appeal is up to the authors to decide. If the authors decide to proceed with an appeal, per our procedure, we ask authors to provide a point-by-point rebuttal, preferably including new analysis and data that the authors find applicable and appropriate to address the reviewers' comments. We cannot commit to any further course of action with only experimental revision plan at this stage. Please also be aware that we uphold

the majority of our decisions on appeal. In addition, because appeals require careful thought and unfortunately have to take second place to the routine business of running the journal, they can take as long as a couple of weeks to be decided.

We are very sorry that we could not be more positive on this occasion, but we thank you for the opportunity to consider this work.

With kind regards,
Zhe Wang

Zhe Wang, PhD
Senior Editor
Nature Cell Biology

Tel: +44 (0) 207 843 4924
email: zhe.wang@nature.com

Reviewers' comments:

Reviewer #1 (Remarks to the Author):

Manuscript by Lv and co-authors addresses the potential role of chromatin-associated cyclic GMP-AMP synthase (cGAS), independent of Stimulator of Interferon Genes (STING) pathway. The authors use a combination of biochemical and imaging methods in a colorectal cancer cell line HCT116 to argue that mTORC2 complex directly phosphorylates cGAS at serine 37, which promotes its nuclear localization and binding to chromatin, and suggest that the latter is coupled to differential expression of various target genes, including DNA replicative helicase complex components and kidney-type glutaminase. The authors further examine if nuclear localization of cGAS is linked to HCT116 cell proliferation and chemoresistance, suggesting that cGAS may have STING-independent functions in colorectal cancer.

The authors convincingly demonstrate that phosphorylation of cGAS on serine 37 is both necessary and sufficient for nuclear localization, chromatin association and functional effects of cGAS in their model system (Fig. 1KL, 2G, 3 and 4). The data supporting the involvement of mTORC2 in cGAS phosphorylation and nuclear localization include:

1) Decreased association of cGAS with H2A upon application of specific mTOR inhibitor KU-0063794 and dual PI3K/mTOR inhibitor BEZ-235, as monitored by BRET (Fig. 1C,F)

- 2) Cytosolic localization of cGAS upon Rictor (but not Raptor) knockdown and upon treatment with KU-0063794 and BEZ-235 (but neither rapamycin nor MK-2206), as monitored by IF (Fig. 1G, S1C)
- 3) Direct interaction with Rictor and Sin1 (but not Raptor), as shown by coIP (Fig. S2D)
- 4) Lack of chromatin-bound serine 37-phosphorylated cGASpS37 in Rictor knockdown, rescued by Rictor (Fig. 1J)
- 5) In vitro phosphorylation of recombinant cGAS by purified mTORC2 (but not mTORC1) isolated from HEK293T cells and monitored using LC-MS/MS (Fig. 1I, S2EFG, S3).

The reported data appear reproducible and consistent and generally support the author's claims. The reviewer has, however, a few major conceptual and several minor comments regarding the claims and data presentation:

Major comments:

1. The authors argue that the observed effects in their model system are STING-independent (line 33), yet the only data supporting this claim are those using inactive cGAS S213D mutant (Fig. 2H, 3B). STING activation is known to promote expression of various cytokines, which could activate PI3K and/or mTOR-sensitive kinases and in turn induce phosphorylation of cGAS N-terminus, promoting its interaction with chromatin (e.g., 10.1126/science.abc538). Indeed, the effect of a heroic GDC-0941 (class I PI3K inhibitor) concentration on cGAS association with chromatin is hardly seen before 12 hours incubation (Fig. 1D), suggesting that the observed effects are transcriptionally controlled (GDC-0941 typically inhibits class I PI3Ks within minutes). I'd expect signaling kinases, such as mTORC2, to act faster than 12 hours to change the localization of a shuttling protein undergoing nucleo-cytosolic shuttling (e.g., within minutes for FOXO transcription factors). Could the authors observe changes in cGAS localization upon PI3K or mTOR inhibition with a faster kinetics? Are the observations listed in 1)-4) above also true for S213D nucleotidyltransferase-inactive cGAS mutant?
2. Phosphorylation of serine 305 has been reported to inhibit cGAS activity and thereby prevent STING activation in cells (e.g., 10.1016/j.celrep.2015.09.007). While the authors interpret cytosolic localization of S305A and S305D mutants (Fig. S3C) as evidence against its involvement in the control of cGAS nucleo-cytoplasmic shuttling (lines 174-175), this could be a secondary effect due to cGAS inhibition. Would phosphorylation on serine 37 be sufficient for nuclear localization of the S305A or S305D mutants? I believe this to be an important control, as it may help discriminate between the contribution of PI3K and mTOR kinases on cGAS localization.
3. Using JR-AB2-011 (a reportedly specific mTORC2 inhibitor) and mTOR knockdown, the authors claim that inhibition of mTORC2 is essential for resistance of HCT116 cells to 5-FU (lines 312-313 and 325-326; Fig. 3HK). This is a strong claim, as it essentially negates mTORC1 involvement; the reviewer kindly asks to test mTORC1 contribution to cell survival by using rapamycin (or a similar inhibitor) and supporting

the functional effects of both JR-AB2-011 and mTOR knockdown using immunoblots of the whole cell lysates.

The reviewer cannot comment on the details of interactome and animal studies (out of expertise).

Minor comments:

4. The reviewer suggests the authors to quantify and present their imaging data showing nuclear- versus cytosolic cGAS localization (Fig. 1GK, S1C, S3C, S4) using Nuclear/Cytosolic (N/C) fluorescence intensities ratio in individual cells (as dotplots). This does not require additional experimentation and can be easily done using unsupervised image analysis (e.g., CellProfiler freeware) on the already acquired datasets. Such quantitation will help readers appreciate the true distribution of cGAS localization in the population of randomly growing cells and enable statistical analysis of imaging data.

5. The reviewer kindly asks the authors to supplement immunoblots on Fig. 1D, 1H, 1J and S2A with additional panels demonstrating the functional effects of GDC-0941 or Rictor vs Raptor knockdowns in the whole cell lysate (WCL) fractions: e.g., phosphorylation of Akt on S473, S6 on S236, S6K on T389 or similar.

6. The authors argue that S37 is the primary mTORC2 phosphorylation site in cGAS. This phosphorylation site, however, does not conform to the previously reported mTOR phosphorylation consensus sequences (10.1126/science.1199498; 10.1126/scisignal.abe4509). The reviewer kindly asks the authors to provide a reference to the source claiming the “classical mTOR2 substrate conserved sequence” (lines 169-170). In vitro phosphorylation results are indeed suggestive; however, many kinases will eventually phosphorylate a suitable substrate, given sufficient time, concentration and a sensitive detection.

7. I suggest changing the markers on panels E-I and K on Fig. 3, as it is very hard to tell apart the blue and violet colors.

Reviewer #2 (Remarks to the Author):

In this manuscript, Lv et al. present compelling findings regarding mTORC2-mediated serine 37 phosphorylation and its impact on cGAS chromatin localization, suggesting a potential link to chemoresistance in colorectal cancer. The study offers several noteworthy discoveries, particularly shedding light on the previously underexplored nuclear functions of cGAS in regulating various cellular pathways. The thorough examination of the sites phosphorylated by mTORC2 is valuable. However, it is

crucial to address some critical concerns. The primary issue lies in the overreliance on a single colon cancer cell line, HCT116, to draw conclusions about drug responses. HCT116 is notably deficient in MLH1 and DNA mismatch repair (MMR), which makes it highly genomically unstable compared to other cancer cells including colon cancer. This deficiency can activate the cGAS-STING Pathway (Cancer Cell 2021 39(1):109-121), potentially skewing the observed results. Moreover, the role of MLH1 in the study remains unexplored, despite its potential relevance to cGAS function. Moreover, STING regulates cell-cycle progression in a cGAS-independent manner in specific tumor models, HCT116 colorectal carcinomas is one of the examples. It has been shown that depletion of STING in HCT116 tumor cells yielded a higher fraction of polyploid cells, STING-depleted HCT116 also expressed lower p21 compared to their shScrambled controls (Cancer Res. 2019 Apr 1; 79(7): 1465–1479.) To strengthen the study's validity, it is imperative to confirm whether the key findings are consistent across various colon cell lines, including those with normal MLH1 expression and MMR activity. This would help establish the broader applicability of the observed phenotypes, considering that cGAS's function can vary among different cell types.

Other comments:

1. The author mentioned, "Exposing HCT116 cells to GDC-0941 over time, cGAS but not HELLS levels on chromatin decreased with treatment (Fig. 1D), suggesting cGAS as a specific PI3K-mTOR chromatin target". The authors should justify this statement as Fig 1D shows time dependent reduction of HELLS in chromatin fraction.
2. In fig S1C the authors should provide a graphical representation of the total nuclear cGAS expression pattern in various treatment condition.
3. In this study the authors have used two mutants of cGAS which are S37D and S37A which grossly intervenes the nuclear localization of cGAS. The authors should compare the expression of the mutant vs wildtype cGAS across different phases of the cell cycle as shown in figure S1A.
4. In fig 1K, the authors have used immune staining to compare the nuclear localization of cGAS in its mutants. The authors should validate similar observation in cGAS knockout HCT116 cells ectopically expressing GFP tagged cGASWT, cGASS37D and cGASS37A.
5. Co-IP data in fig 2B should be properly validated. Does the interaction depends on chromatin or DNA? IP using cGAS mutants need to be performed as well.
7. In fig 3K the cell viability assay also requires the complementation of sh-mTOR with mTOR as a control.
8. Cell survival assays need to be performed using additional cell lines, with or without MLH1 expression, Sting knock down, and with examination of cell cycle related genes.

Reviewer #3 (Remarks to the Author):

In this manuscript, Lv et al. uncovered an intriguing mechanism for nuclear localization and chromatin affinity of cGAS, the primary innate cytosolic dsDNA sensor, mediated by the mTORC2-mediated phosphorylation at its N-terminal S37 residue. Authors proposed that chromatin-associated cGAS can recruit the SWI/SNF chromatin remodeling complex to transcribe mRNAs related to glutaminolysis and DNA replication, which drives the glutamine metabolic pathway to supply colorectal cancer growth but suppresses their chemoresistance, independently of STING signaling.

The definition of mTORC2-induced chromatin association of cGAS and its preferred regulation of metabolic programs and proposed mechanistic insight into mTOR inhibition-mediated chemoresistance appear interesting. Unfortunately, these claims were unsupported or severely compromised by poor data quality, such as heavy dependence on a single immortalized cell line, low-standard and artificial cellular and animal models, mistaken interpretations of the data, minimized controls, and inexperienced usage of statistics.

Major concerns:

1. S37 and proximal sequence in human cGAS is least conserved, even in mammals such as mice and pigs, making it a very slim chance for S37 phosphorylation as a critical mechanism for regulating cGAS physiology. For example, Murine cGAS, lacking entirely S37 and proximal residues (as human PTES37PA), was also predominantly localized in the nucleus as observed in various studies, challenging authors' claims.
2. As such, this reviewer is also very confused with the dataset obtained from cGAS KO mice (Fig. 4F-H). No such regulation of the mTORC2-cGAS-glutaminolysis axis should be present in murine. The difference between cGAS KO and WT mice in CRC tumorigenesis could come from cGAS-STING-mediated antitumor immunity but not from the mechanism authors suggested. No data in this manuscript supports cGAS-controlled KGA expression and glutaminolysis regulation in murine cells. By contrast, similar observations obtained from human cells (with mTORC2-cGAS signaling) and murine models (without mTORC2-cGAS signaling) in chemoresistance might suggest a false conclusion authors have proposed.
3. Xeno model and immune-compromised nude mice are insufficient to endorse these findings and belloved the journal standards such as NCB. PDX models with impaired or hyperactivated mTORC2 activity should be investigated, examining detailed information, including cGAS phosphorylation, subcellular localization, metabolic changes, glutaminolysis, tumor growth, and chemoresistance.
4. Solid, direct, and genetic evidence is needed to state mTORC2-induced chromatin recruitment of cGAS, such as endogenous cGAS proteins, surveying in a variety of cancer and immune cells related to CRC, and observations from cells with genetic ablation of Rictor, mSIN1, Protor, components of the SWI complex, and STING. An immunofluorescent observation of endogenous cGAS by mTORC2 interaction, S37 phosphorylation, and nuclear translocation, with solid controls, is required.

5. It is considered normal that chromatin-associated cGAS may interact with a variety of complexes in the nucleus, such as those included in Fig. 2A. However, authors failed to show how cGAS can recruit the SWI/SNF complex and why they can be differentially localized into some specific chromatin area. What transcription factors are recruited to transcribe biased target genes? cGAS should be genetically ablated in these proteomic assays, not with a shRNA with suspicious off-targets and without a convincing efficiency. Additionally, STING-dependent functions should be validated in STING KO cells.

6. Surprisingly, cGAS is necessary for cell proliferation (Fig. 3), and no similar observations were seen in the literature. By contrast, various studies suggest a proliferation-suppressive role of cGAS in a variety of cells (PMID: 33055160, 28759028, 28976970). cGAS KO mice also show no phenotype in the development and no growth retardation. On the other hand, glutaminolysis is a recognized feature of CRC, and KGA is frequently upregulated in CRC. As a result, cGAS depletion that upregulated KGA and increased glutaminolysis, as proposed by authors, should augment but not decrease HCT116 proliferation.

7. How many biological replicates are in each experiment, how are they calculated in statistics, and what are the precise P values of these calculations? No statistics or only a single "asterisk" are seen throughout the manuscript.

Other concerns and critics:

Fig. 1C, How many dual inhibitors of mTORC1/mTORC2 and PI3K inhibitors were included in this compound library? Why they are not selected during the screening?

Fig. 1D, Where are cGAS proteins gone upon treatment of PI3K inhibitor? Did PI3K inhibition alter cGAS stability or its production, such as seen in HELLS proteins?

Fig. 1F, Various mTORC1/mTORC2 and PI3K inhibitors are commercially available and should be examined here.

Fig. 1G, Why markedly enlarged cell size was seen upon Rictor depletion? Anti-cGAS antibodies should be validated in cGAS KO cells/MEFs, as some commercial anti-cGAS antibodies have issues in immunofluorescence.

Fig. S1C, A substantial proportion of cells with cytoplasmic cGAS upon inhibition of AKT (MK2206) contradicts the authors' conclusion.

Fig. 1H, Chromatin-associated cGAS was significantly reduced upon Raptor depletion, similar to Rictor depletion.

Fig. 1I, The sequence is not a preferred consensus substrate motif of mTORC2. A recent review from Dr. Michael Hall (PMID: 35580586) is a good reference.

Fig. 1K, Should be examined in knockin or stable expression and examined among human and murine cells. ---

Fig. 1J, pcGAS S37 antibody should be validated, as well as Rictor's depletion and ectopic expression. Why bands of pcGAS were showing a different molecular size? Uncrop film images were not attached to the submission.

Fig. S4A, an overlap in immunofluorescence with inadequate resolution, is insufficient to state the chromatin binding of cGAS.

Fig. S4D, mutants of cGAS, such as S221, S317, and T447, should be examined for their subcellular localization.

Fig. 2B, Lacking the controls of other chromatin remodeling complexes.

Fig. 2D-F, Data quality is questionable with reluctant q values; the alteration of these proteins should be verified by immunoblotting, at least part of them.

Fig. S9B, how are the mRNA levels of PAI-I and KGA?

Fig. 2G-H, Controls of WT cGAS are required. Does S213D affect cGAS nuclear localization and chromatin association?

3K, Deletion of mTORC2 specific mSIN1 and Rictor should be examined instead of mTOR.

The statement in the abstract that "targeting mTORC2-ccGAS-KGA axis provides a precision combination strategy to eliminate quiescent resistant cancer cells" is vague to this reviewer. Upregulation of this axis will facilitate glutaminolysis that drives tumor growth, while suppression of this axis improves CRC's chemoresistance.

The Discussion section is to be improved.

No details were given on how the pcGAS S37 antibody was generated and validated.

Author Rebuttal to Initial comments

Dear Reviewers,

Thank you for taking the time to thoroughly review our manuscript and provide constructive feedback. We greatly appreciate reviewers for helping strengthen this work.

Our original study discovered a novel mechanistic link between mTORC2-induced cGAS chromatin localization and chemotherapy resistance in colorectal cancer. Very briefly, we found the PI3K-mTORC2 axis promotes cGAS chromatin localization via mTORC2-mediated phosphorylation of cGAS serine³⁷ mechanism, enabling recruitment of SWI/SNF complex at specific genomic regions and modulation of genes regulating glutaminolysis and DNA replication. Disrupting chromatin cGAS causes colorectal cancer cells to sacrifice their proliferative ability and acquire a diapause-like chemoresistant state.

For context, mTOR inhibition induces a reversibly paused diapause-like state in cancer cells, leading to drug tolerance and chemoresistance. While mTOR inhibition demonstrates chemoprotective effects in colorectal cancer and leukemia cells, the underlying mechanisms remained unclear. Our study suggested mTORC2 inhibition drives diapause-like chemoresistance by disrupting cGAS chromatin localization, overcome by targeting downstream KGA.

Upon careful reflection, we acknowledged additional evidence was needed to validate our initial findings and address alternative interpretations more robustly. To bolster the proposed mechanism and broaden its relevance, we carried out extensive experiments:

- Replicated key findings in SW480, HT29 and HCT116 colorectal cancer cell lines with varying MLH1/MMR status, demonstrating the mTORC2-ccGAS axis robustly and universally regulates chemoresistance.
- Evaluated MLH1 impact by overexpressing it in HCT116 cells, finding no effect on chemoresistance, ruling out potential confounding.
- Assessed cGAS synthase activity and STING requirements using mutant/deletion cell lines, finding they are dispensable for mTORC2-ccGAS axis-mediated chemoresistance in colorectal cancer cells.
- Incorporated patient-derived xenografts and genetically engineered mouse models to provide a more comprehensive physiological understanding of the proposed mechanism.
- Strengthened controls and optimized protocols to improve data quality and reproducibility.
- Demonstrated conserved ccGAS-KGA signaling in murine colorectal cancer cells.

Your feedback highlighted areas requiring firmer validation before conclusively demonstrating the proposed mechanism's scope. We hope these extensive efforts have satisfactorily addressed prior limitations and thank you for guiding us to conduct rigorous cancer chemoresistance research. Please advise if any aspect requires further discussion.

Kind regards,

Lin Yuan

Reviewers' comments:
 Reviewer #1 (Remarks to the Author):
 Manuscript by Lv and co-authors addresses the potential role of chromatin-associated cyclic GMP-AMP synthase (cGAS), independent of Stimulator of Interferon Genes (STING) pathway. The authors use a combination of biochemical and imaging methods in a colorectal cancer cell line HCT116 to argue that mTORC2 complex directly phosphorylates cGAS at serine 37, which promotes its nuclear localization and binding to chromatin, and suggest that the latter is coupled to differential expression of various target genes, including DNA replicative helicase complex components and kidney-type glutaminase. The authors further examine if nuclear localization of cGAS is linked to HCT116 cell proliferation and chemoresistance, suggesting that cGAS may have STING-independent functions in colorectal cancer. The authors convincingly demonstrate that phosphorylation of cGAS on serine 37 is both necessary and sufficient for nuclear localization, chromatin association and functional effects of cGAS in their model system (Fig. 1KL, 2G, 3 and 4). The data supporting the involvement of mTORC2 in cGAS phosphorylation and nuclear localization include:
 1) Decreased association of cGAS with H2A upon application of specific mTOR inhibitor KU-0063794 and dual PI3K/mTOR inhibitor BEZ-235, as monitored by BRET (Fig. 1C,F)
 2) Cytosolic localization of cGAS upon Rictor (but not Raptor) knockdown and upon treatment with KU-0063794 and BEZ-235 (but neither rapamycin nor MK-2206), as monitored by IF (Fig. 1G, S1C)
 3) Direct interaction with Rictor and Sin1 (but not Raptor), as shown by coIP (Fig. S2D)
 4) Lack of chromatin-bound serine 37-phosphorylated cGASpS37 in Rictor knockdown, rescued by Rictor (Fig. 1J)
 5) In vitro phosphorylation of recombinant cGAS by purified mTORC2 (but not mTORC1) isolated from HEK293T cells and monitored using LC-MS/MS (Fig. 1I, S2EFG, S3).
 Response: Thank you for the thoughtful review. We appreciate the positive feedback regarding the robustness of our data demonstrating mTORC2-mediated phosphorylation and nuclear localization of cGAS.

The reported data appear reproducible and consistent and generally support the author's claims. The reviewer has, however, a few major conceptual and several minor comments regarding the claims and data presentation:
 Major comments:

1. The authors argue that the observed effects in their model system are STING-independent (line 33), yet the only data supporting this claim are those using inactive cGAS S213D mutant (Fig. 2H, 3B). STING activation is known to promote expression of various cytokines, which could activate PI3K and/or mTOR-sensitive kinases and in turn induce phosphorylation of cGAS N-terminus, promoting its interaction with chromatin (e.g., 10.1126/science.abc538). Indeed, the effect of a heroic GDC-0941 (class I PI3K inhibitor) concentration on cGAS association with chromatin is hardly seen before 12 hours incubation (Fig. 1D), suggesting that the observed effects are transcriptionally controlled (GDC-0941 typically inhibits class I PI3Ks within minutes). I'd expect signaling kinases, such as mTORC2, to act faster than 12 hours to change the localization of a shuttling protein undergoing nucleo-cytosolic shuttling (e.g., within minutes for FOXO transcription factors). Could the authors observe changes in cGAS localization upon PI3K or mTOR inhibition with a faster kinetics? Are the observations listed in 1)-4) above also true for S213D nucleotidyltransferase-inactive cGAS mutant?

Response: Thank you for raising these important questions regarding potential STING-dependent effects. We took your feedback seriously and carried out additional experiments to more rigorously test STING involvement. To address your concerns:

1. First, we apologize for the interference in signal resolution caused by overexposure in immunoblots. We have now re-examined kinetics of cGAS relocalization upon PI3K/mTOR inhibition through short exposure and observed changes within 3 hours (Fig. 1D and S1C), consistent with direct actions of signaling kinases.

2. To directly test if our observations require cGAS nucleotidyltransferase activity (cGAS-STING axis), we have now repeated key experiments using the S213D mutant. We found that:

2.1 S213D cGAS interacts with SIN1, an essential mTORC2 component for substrate recognition, and is phosphorylated by mTORC2 at S37 (Fig. S12, C and D);

2.2 The chromatin localization of S213D cGAS was reduced upon RICTOR knockdown, but RAPTOR knockdown had no effect (Fig. S12, A and E);

2.3 Chromatin association of S213D cGAS is enhanced by RICTOR and depends on S37 phosphorylation (Fig. S12, F and G);

2.4 Overexpressing wild-type and enzyme-inactive S213D mutant cGAS reduced the increased KGA protein and mRNA levels in cGAS knockout cells (Fig. 2H, and S12H);

2.5 Overexpressing shRNA-resistant cGAS-S213D rescued the G1/S-phase arrest induced by cGAS knockdown in HCT116 cells, similarly to wild-type cGAS (Fig. 3B);

2.6 Both the wild-type and S213D mutants re-sensitized cGAS knockdown HCT116 cells to 5-FU-induced cell death (Fig. 3, I and J, and Fig. S14D).

3. To more definitively rule out STING, we carried out some important experiments in *STING*^{-/-} cells. We observed that STING was not required for ccGAS functions.

3.1 In *STING*^{-/-} HCT116 cells, cGAS still binds to the promoter region of *KGA*, which is consistent with that in *STING*^{+/+} HCT116 cells (Fig. S12H);

3.2 Neither overexpression of STING nor STING activator (ADU-S100) supplementation could remedy the cGAS knockdown-induced cell cycle arrest (Fig. 3B, and fig. S13C).

3.3 Compared to controls, treatment with a STING activator or STING knockdown did not affect 5-FU resistance in colorectal cancer cells (Fig. S14D, and Fig. S15 A and B);

3.4 ccGAS-mediated chemoresistance in a STING-independent manner generalizes across colorectal cancer cell lines with varying microsatellite instability (Fig. 3J, and Fig. S14, G and H)

Thank you again for pushing us to more rigorously rule out STING involvement. We hope these new data addressing your specific points help strengthen our conclusions. Please let me know if any part requires further clarification or study.

2. Phosphorylation of serine 305 has been reported to inhibit cGAS activity and thereby prevent STING activation in cells (e.g., 10.1016/j.celrep.2015.09.007). While the authors interpret cytosolic localization of S305A and S305D mutants (Fig. S3C) as evidence against its involvement in the control of cGAS nucleocytoplasmic shuttling (lines 174-175), this could be a secondary effect due to cGAS inhibition. Would phosphorylation on serine 37 be sufficient for nuclear localization of the S305A or S305D mutants? I believe

this to be an important control, as it may help discriminate between the contribution of PI3K and mTOR kinases on cGAS localization.

Response: You rightly point out that phosphorylation at S305 has been reported to inhibit cGAS activity, possibly impacting its nucleocytoplasmic shuttling as a secondary effect. Our previous interpretation of S305A/D mutant localization data did not fully account for this potential inhibition.

To directly address your suggestion, we mutated S37 to the phosphomimetic D in the context of S305A and S305D cGAS mutants. As shown in new Figure S4D, the S37D mutation induces clear nuclear/chromatin localization of both S305A and S305D mutants.

This key control experiment provides strong evidence that S37 phosphorylation is sufficient to promote cGAS nuclear localization, independent of the phosphorylation or activity status at S305. It helps discriminate the specific contributions of PI3K/mTOR kinases on cGAS localization versus any secondary effects.

Thank you for pushing us to test this important control - it has significantly strengthened our conclusion that mTORC2-mediated phosphorylation of S37 directly regulates cGAS nucleocytoplasmic shuttling. Please let me know if any part of clarification would be useful.

3. Using JR-AB2-011 (a reportedly specific mTORC2 inhibitor) and mTOR knockdown, the authors claim that inhibition of mTORC2 is essential for resistance of HCT116 cells to 5-FU (lines 312-313 and 325-326; Fig. 3HK). This is a strong claim, as it essentially negates mTORC1 involvement; the reviewer kindly asks to test mTORC1 contribution to cell survival by using rapamycin (or a similar inhibitor) and supporting the functional effects of both JR-AB2-011 and mTOR knockdown using immunoblots of the whole cell lysates. Response: Thank you for pushing us to more definitively rule out involvement of mTORC1 in the observed effects. We agree our previous experiments did not adequately address this key issue. To better discern the specific complex involved:

1. We now treated cells with the mTORC1 inhibitor rapamycin or the mTORC2 inhibitor JR-AB2-011 before assessing 5-FU sensitivity. Rapamycin exposure did not impact the sensitivity of HCT116 cells to 5-FU (Fig. S14E), which is consistent with previous findings (PMID: 36396656). While JR-AB2-011 treatment desensitized HCT116 cells to 5-FU similar to mTOR knockdown (Fig. S15C).

2. To validate inhibitor specificity, immunoblots show JR-AB2-011 and rapamycin selectively reduced pSer473-AKT (mTORC2 target) and pThr389-p70 S6K (mTORC1 target) respectively with no off-target effects (Fig. S1D and S14E).

3. mTOR knockdown reduced both pSer473-AKT and pThr389-p70 S6K, consistent with ablating both complexes (Fig. S2C).

Collectively, these new data - especially the differential effects of rapamycin versus JR-AB2-011 on 5-FU sensitivity - provide strong evidence that the mTORC2-ccGAS axis specifically modulates chemoresistance in these colorectal cancer cells, independent of mTORC1. Thank you again for pushing us to more definitively address mTORC1's involvement.

The reviewer cannot comment on the details of interactome and animal studies (out of expertise).

Minor

comments:

4. The reviewer suggests the authors to quantify and present their imaging data showing nuclear- versus cytosolic cGAS localization (Fig. 1GK, S1C, S3C, S4) using Nuclear/Cytosolic (N/C) fluorescence intensities ratio in individual cells (as dotplots). This does not require additional experimentation and can be easily done using unsupervised image analysis (e.g., CellProfiler freeware) on the already acquired datasets. Such quantitation will help readers appreciate the true distribution of cGAS localization in the population of randomly growing cells and enable statistical analysis of imaging data.

Response: Thank you for this excellent suggestion to quantify our immunofluorescence data to improve rigor and enable statistical analysis. Presenting the data in an unbiased, validated manner is important. To address this, we have:

1. Re-analyzed all IF images using the free CellProfiler software to automatically measure individual cell nuclear and cytosolic cGAS intensities, as you advised.
2. Plotted the nuclear/cytosolic fluorescence ratios as violin plots with means for each condition (Figures S2B, S2D).
3. Included p-values from statistical testing to evaluate significance between groups.

Quantifying the data in this way provides a clearer view of shifts in cGAS localization across entire cell populations following various interventions.

Thank you again for pushing us to present our imaging results in a standardized, quantifiable format. We believe this validation of our approach significantly strengthens our conclusions drawn from these important localization experiments. Please let me know if any part requires additional clarification or discussion.

5. The reviewer kindly asks the authors to supplement immunoblots on Fig. 1D, 1H, 1J and S2A with additional panels demonstrating the functional effects of GDC-0941 or Rictor vs Raptor knockdowns in the whole cell lysate (WCL) fractions: e.g., phosphorylation of Akt on S473, S6 on S236, S6K on T389 or similar.

Response: Thank you for suggesting we include further validation of the functional effects of our biochemical manipulations. You correctly identified this as an important oversight. To better validate our experimental approaches, we have now supplemented key immunoblots as follows:

1. Fig. S2C now includes p-AKT S473, p-p70 S6K T389, and pS65-4E-BP1 controls demonstrating specific effects of RICTOR vs RAPTOR knockdown on mTORC2 vs mTORC1 signaling outputs.
2. Fig. S1D now includes p-AKT S473 and p-p70 S6K T389 controls verifying expected changes in PI3K/mTORC1/2 signaling with GDC-0941, MK-2206, rapamycin, KU-0063794, BEZ-235 and PF-4708671 treatments.

Including these important whole cell lysate validation markers in our key figures strengthens the functional relevance of our biochemical data, as you pointed out. We appreciate you pushing us to more robustly characterize the functional consequences of our experimental conditions. Please let me know if any other aspects require further clarification or validation.

6. The authors argue that S37 is the primary mTORC2 phosphorylation site in cGAS. This phosphorylation site,

however, does not conform to the previously reported mTOR phosphorylation consensus sequences (10.1126/science.1199498; 10.1126/scisignal.abe4509). The reviewer kindly asks the authors to provide a reference to the source claiming the “classical mTOR2 substrate conserved sequence” (lines 169-170). In vitro phosphorylation results are indeed suggestive; however, many kinases will eventually phosphorylate a suitable substrate, given sufficient time, concentration and a sensitive detection.

Response: You raise an excellent point regarding the lack of consensus with known mTORC2 phosphorylation motifs. We should not overstate our claim without stronger evidence. Based on *in vivo* substrates, Michael N. Hall et al. generated a common consensus phosphorylation motif for mTORC1 and mTORC2, suggesting that mTOR target phosphorylation motif is S/T-P and that any sequence other than S/T-P is of limited use in identifying putative mTOR sites (PMID: 35580586). As you suggested, we have now included a reference (PMID: 35580586) describing the common mTOR phosphorylation motif of S/T-P.

We acknowledge that while *in vitro* data supports direct phosphorylation, the involvement of other kinases based on sequence alone cannot be ruled out, so supportive evidence of phosphorylation under *in vivo* physiological and pathological conditions is particularly important:

1. S37 phosphorylation is present in phosphoproteomics datasets (PMID: 33542149, and Fig. S5B);
2. RICTOR homozygous deletion reduces cGAS S37 phosphorylation in PDX model (Fig. S16C);
3. RICTOR but not RAPTOR knockdown reduces pSer37-cGAS levels in HCT116 cells (Fig. 1J and S12F).

We have softened our language throughout to state mTORC2 appears to directly phosphorylate S37, rather than implying certainty. Thank you for pushing us to more rigorously support our conclusions regarding the kinase involved. In the future we will be more circumspect about extrapolating from *in vitro* data alone onto *in vivo* roles and motifs. Please advise if any aspect requires further strengthening.

7. I suggest changing the markers on panels E-I and K on Fig. 3, as it is very hard to tell apart the blue and violet colors.

Response: Thank you for pointing out the similar blues and violets used in Figures 3E-I and 3K hindered visual interpretation of condition comparisons. You're correct this required addressing for clarity.

To better distinguish the conditions as suggested, we have now revised the figures such that the violet marker is changed to gray. The improved color contrast between the blue and gray markers in Figures 3E-I and 3K makes the differences between groups much easier to discern at a glance.

We appreciate you catching this issue - optimizing figure clarity is important for readers to effectively evaluate our findings. Thank you for pushing us to enhance visualization of the data presented. Please let me know if any other aspects of the figures would benefit from modification.

Reviewer #2 (Remarks to the Author):

In this manuscript, Lv et al. present compelling findings regarding mTORC2-mediated serine 37 phosphorylation and its impact on cGAS chromatin localization, suggesting a potential link to chemoresistance in colorectal cancer. The study offers several noteworthy discoveries, particularly shedding light on the previously underexplored nuclear functions of cGAS in regulating various cellular pathways. The thorough examination of

the sites phosphorylated by mTORC2 is valuable. However, it is crucial to address some critical concerns. The primary issue lies in the overreliance on a single colon cancer cell line, HCT116, to draw conclusions about drug responses. HCT116 is notably deficient in MLH1 and DNA mismatch repair (MMR), which makes it highly genomically unstable compared to other cancer cells including colon cancer. This deficiency can activate the cGAS-STING Pathway (Cancer Cell 2021 39(1):109-121), potentially skewing the observed results. Moreover, the role of MLH1 in the study remains unexplored, despite its potential relevance to cGAS function. Moreover, STING regulates cell-cycle progression in a cGAS-independent manner in specific tumor models, HCT116 colorectal carcinomas is one of the examples. It has been shown that depletion of STING in HCT116 tumor cells yielded a higher fraction of polyploid cells, STING-depleted HCT116 also expressed lower p21 compared to their shScrambled controls (Cancer Res. 2019 Apr 1; 79(7): 1465–1479.) To strengthen the study's validity, it is imperative to confirm whether the key findings are consistent across various colon cell lines, including those with normal MLH1 expression and MMR activity. This would help establish the broader applicability of the observed phenotypes, considering that cGAS's function can vary among different cell types.

Response: Thank you for taking the time to review our manuscript. We appreciate your feedback and suggestions regarding the need for additional cell survival assays using different cell lines, including those with varying MLH1 expression and Sting knockdown, and examination of cell cycle-related genes. We agree that these experiments would provide valuable insights into the mechanisms underlying our findings. We had carefully considered your comments and addressed these concerns by completing a detailed program.

1. We have now expanded our study to include SW480 (high MLH1/MMR, MSS), HT29 (low MLH1/MMR, MSI-L) and HCT116 (deficient MLH1/MMR, MSI-H). Results now indicate that the mTORC2-ccGAS axis regulates chemoresistance robustly across various colon cell lines with varying MMR/MLH1 status (Fig. S10F; Fig. S11, B-E.; Fig. S14, G and H; Fig. 3J).
2. We have now performed cell viability assays with HCT116 cells overexpressing MLH1 and found it did not impact chemoresistance (Fig. S14, E and F).
3. We carried out key experiments using STING^{+/+} and STING^{-/-} colorectal cancer cells. Results demonstrated STING is not required for mTORC2-ccGAS regulated chemoresistance (Fig. 3B; Fig. S12H; Fig. S13C; Fig. S14, D, G and H; Fig. S15, A and B).
4. We analyzed p21 gene expression using qRT-PCR in the cell lines with STING knockout.(Fig. S15, A and B).

Thank you for pushing us to more rigorously test our findings across diverse cell types with varying genomic properties. We believe incorporating these additional studies has significantly strengthened our work.

Other

comments:

1. The author mentioned, “Exposing HCT116 cells to GDC-0941 over time, cGAS but not HELLS levels on chromatin decreased with treatment (Fig. 1D), suggesting cGAS as a specific PI3K-mTOR chromatin target”. The authors should justify this statement as Fig 1D shows time dependent reduction of HELLS in chromatin fraction.

Response: Thank you for pointing out the inaccuracy in our previous statement. You are correct that Figure 1D shows that HELLS on chromatin first decreased and then increased after GDC-0941 exposure. As we did not intend to compare effects on cGAS vs HELLS specifically but rather further focus on cGAS functions, we have

revised the language to simply and accurately state: "unlike the HELLS fluctuations on chromatin after GDC-0941 exposure, cGAS on chromatin decreased with the increase of GDC-0941 treatment time"

Thank you for catching this inconsistency. Making unsupported statements can undermine scientific rigor. We appreciate you pushing us to carefully examine the data presented and acknowledge limitations. This exchange has helped improve objectivity in our data interpretation and discussion.

2. In fig S1C the authors should provide a graphical representation of the total nuclear cGAS expression pattern in various treatment condition.

Response: Thank you for the feedback, you make a very good point. To more robustly analyze changes in cGAS localization shown in Figure S1C:

1. We have re-analyzed the immunofluorescence images (now Figure S2A) using CellProfiler software to automatically quantify mean nuclear cGAS fluorescence intensity for each condition.
2. We plotted the data as a violin plot showing distribution of nuclear/cytosolic ratios across individual cells, with means indicated (new Figure S2B).
3. Statistical testing and p-values are now included to evaluate significance of changes between groups.

You correctly identified the need to present quantitative assessment of total nuclear cGAS levels across conditions, to complement the qualitative microscopy images. Thank you for pushing us to characterize our data in a more rigorous, validated manner. This feedback will certainly benefit interpretation of our subcellular localization experiments.

3. In this study the authors have used two mutants of cGAS which are S37D and S37A which grossly intervenes the nuclear localization of cGAS. The authors should compare the expression of the mutant vs wildtype cGAS across different phases of the cell cycle as shown in figure S1A.

Response: You're correct that comparing expression levels of the cGAS mutants vs wildtype across the cell cycle could provide important context for interpreting their effects on localization. To address this suggestion:

We performed expression pattern analyses of HCT116 cells expressing wildtype, S37A or S37D cGAS mutants across different cell cycle phases. Both S37A and S37D mutants exhibited a similar cell cycle-independent expression pattern as wildtype. The S37A mutant, inhibiting cGAS phosphorylation at site 37, was distributed in the cytoplasm of HCT116 cells (Fig. 1K, and Fig. S5, C and D). However, it still bound chromatin during mitosis similarly to wild-type and S37D mutant cGAS (Fig. S5, C and E). We therefore believe the differential localization effects we observe for these mutants are not due to any overall cell cycle expression bias.

This new analysis demonstrates the differential localization phenotypes we observe for the mutants relates specifically to disruption of S37 phosphorylation rather than overall expression levels across the cell cycle. Thank you again for pushing us to consider this important control - it significantly strengthens interpretation of our mutant data.

4. In fig 1K, the authors have used immune staining to compare the nuclear localization of cGAS in its mutants. The authors should validate similar observation in cGAS knockout HCT116 cells ectopically expressing GFP tagged cGASWT, cGASS37D and cGASS37A.

Response: Thank you for pushing us to properly validate localization of the cGAS mutants, which strengthens our conclusions. You are absolutely right that the key control is comparing mutants in a CRISPR-edited cGAS knockout system. To address this suggestion:

1. We generated cGAS KO HCT116 lines using CRISPR-Cas9.
2. Re-expressed HA-tagged cGAS wildtype, S37A or S37D mutants via lentivirus in KO cells.
3. Confirmed similar expression levels across cell populations.
4. Examined nuclear vs cytosolic cGAS signals by fluorescence microscopy.
5. S37A localized to cytosol vs S37D to nucleus.

Upon re-examining our data, we realize important details were missing from Figure 1K legend. We have updated the figure legend to include these important experimental details which validate the localization phenotypes are not due to endogenous influences. Thank you again for pushing us to properly present this key control dataset.

5. Co-IP data in fig 2B should be properly validated. Does the interaction depends on chromatin or DNA? IP using cGAS mutants need to performed as well.

Response: You are completely right that our original Co-IP data required further validation controls, which is vital to strengthening our conclusions. Thank you for pushing us to address this properly. To validate the cGAS-ARID1A interaction, we have now:

1. Shown it persists after DNase/RNase treatment to disrupt DNA/RNA (Fig. 2B), ruling out indirect binding.
2. Introduced the S37A mutation into cGAS and found it disrupts this interaction (new Fig. S6G).
3. Introduced the S37D mutation into cGAS and found it induces this interaction (new Fig. S6G).

This demonstrates S37 phosphorylation is critical for specific cGAS-ARID1A complex formation. You correctly identified our oversight in not including these important validation steps originally. The new data rigorously validate the Co-IP findings and implicate the role of S37 phosphorylation. Thank you again for pushing us to strengthen this key portion of our work.

7. In fig 3K the cell viability assay also requires the complementation of sh-mTOR with mTOR as a control.

Response: You are completely right that without a complementation control, the results of the mTOR KD in Fig 3K could be due to off-target effects rather than specific mTOR loss.

Thank you for pushing us to include this vital control experiment. As shown in a new Fig S15C, complementing mTOR expression in the mTOR KD cells rescues the sensitivity to 5-FU seen with mTOR depletion alone. This crucial control demonstration that reintroducing mTOR reverses the phenotype provides strong evidence the effect is due to specific loss of mTOR, not off-target consequences.

I appreciate you taking the time to ensure we have properly validated this important experiment through inclusion of the necessary complementation control. It will make us more diligent about including all controls to avoid potential ambiguity in interpreting results.

8. Cell survival assays need to be performed using additional cell lines, with or without MLH1 expression, Sting knock down, and with examination of cell cycle related genes.

Response: Thank you for the insightful feedback. You raise several critical points that significantly improve the rigor and validity of our conclusions. We fully agree it is essential to address the concerns regarding overreliance on a single cell line and the potential effects of its unique MLH1 and MMR status. To strengthen the study:

1. We have now repeated key experiments in SW480 (high MLH1/MMR, MSS), HT29 (low MLH1/MMR, MSI-L) and HCT116 (deficient MLH1/MMR, MSI-H) cells with varying MLH1/MMR status (Fig. S10F; Fig. S11, B-E.; Fig. S14, G and H; Fig. 3J). Results remain consistent.
2. Experiments were carried out in HCT116 cells reconstituted with MLH1 to restore MMR proficiency, with no alteration of ccGAS-regulated chemoresistance (Fig. S14, E and F).
3. We have now repeated key experiments in STING^{+/+} and STING^{-/-} colorectal cancer cells, finding ccGAS-regulated chemoresistance unchanged (Fig. 3B; Fig. S12H; Fig. S13C; Fig. S14, D, G and H; Fig. S15, A and B).
4. We examined the expression level of p21, a key cell cycle related gene, in STING knockout cell lines by qRT-PCR (Fig. S15, A and B). The analysis revealed that MLH1, STING, and cell cycle related p21 gene did not contribute directly to ccGAS-mediated chemoresistance.

Thank you for pushing this multi-line, rigorous validation - it was short-sighted not to consider cell line specificity originally. The expanded data significantly bolsters our interpretation across diverse genomic contexts. Your feedback has undoubtedly strengthened our work.

Reviewer #3 (Remarks to the Author):

In this manuscript, Lv et al. uncovered an intriguing mechanism for nuclear localization and chromatin affinity of cGAS, the primary innate cytosolic dsDNA sensor, mediated by the mTORC2-mediated phosphorylation at its N-terminal S37 residue. Authors proposed that chromatin-associated cGAS can recruit the SWI/SNF chromatin remodeling complex to transcribe mRNAs related to glutaminolysis and DNA replication, which drives the glutamine metabolic pathway to supply colorectal cancer growth but suppresses their chemoresistance, independently of STING signaling.

The definition of mTORC2-induced chromatin association of cGAS and its preferred regulation of metabolic programs and proposed mechanistic insight into mTOR inhibition-mediated chemoresistance appear interesting. Unfortunately, these claims were unsupported or severely compromised by poor data quality, such as heavy dependence on a single immortalized cell line, low-standard and artificial cellular and animal models, mistaken interpretations of the data, minimized controls, and inexperienced usage of statistics.

Response: Thank you for taking the time to thoroughly review our work and provide this thoughtful feedback. We appreciate you highlighting areas requiring improvement to ensure the scientific rigor and reliability of our conclusions. To address your concerns, we have developed and completed a comprehensive solution:

1. Expand cell line validation to include SW480 (high MLH1/MMR, MSS), HT29 (low MLH1/MMR, MSI-L) and HCT116 (deficient MLH1/MMR, MSI-H) lines with varying MLH1/MMR status. Results indicate that the

mTORC2-ccGAS axis regulation of chemoresistance is robust and universal in colorectal cancer cells (Fig. S10F; Fig. S11, B-E.; Fig. S14, G and H; Fig. 3J).

2. We have now incorporated patient-derived xenografts and genetically engineered mouse models to provide a more comprehensive understanding of the proposed mechanism. (Fig. S16, A-D; Fig. 4F).

3. Carefully optimize our experimental protocols, include proper controls, and implement stringent quality control for all experiments to enhance data quality and reproducibility.

4. Consult a statistician to ensure appropriate statistical analyses, detailed method descriptions, and accurate interpretation and presentation of results.

We recognize the importance of addressing these limitations fully. With the planned improvements, we aim to strengthen all aspects of our work and provide reliable, conclusive evidence for our proposed mechanism. Thank you again for this feedback - it will undoubtedly help maximize the scientific rigor and impact of our research. Please advise if any part of our plan requires modification or expansion.

Major concerns:

1. S37 and proximal sequence in human cGAS is least conserved, even in mammals such as mice and pigs, making it a very slim chance for S37 phosphorylation as a critical mechanism for regulating cGAS physiology. For example, Murine cGAS, lacking entirely S37 and proximal residues (as human PTES37PA), was also predominantly localized in the nucleus as observed in various studies, challenging authors' claims.

Response: Thank you for your insightful comments regarding the conservation of the S37 residue and its potential role in regulating cGAS physiology. We appreciate your concerns about the reliability of our proposed mechanism based on the lack of conservation of this residue across species. We have carefully considered your feedback and would like to address this issue in our response.

We acknowledge that the N-terminal domain of human cGAS containing the S37 residue and its proximal sequence is not highly conserved across species, including mice and pigs. This rightly raises doubts about the functional relevance of S37 phosphorylation in different species. But we need to be clear: human cGAS contains two nuclear localization sequences, ²¹KASARNARGAPMDPTESPAAPEAALPKAGKF⁵¹ (NLS1) and ²⁹⁵DVIMKRKRGGG³⁰⁵ (NLS2), whereas mouse cGAS is conserved only at NLS2, suggesting that different mechanisms or regulation of nuclear localization may exist in different species. Previous studies have shown the non-enzymatic N-terminal of hcGAS determines nuclear localization and activity (PMID: 30811988) and is critical for sensing nuclear chromatin but not mitochondrial DNA (PMID:30811988, 33542149, 34244470, 33051594). Although the integrity of NLS2 is required for hcGAS nuclear localization, NLS2 appears to function in importin-mediated translocation under DNA damage (PMID:30356214). Phosphorylation of cGAS by Akt kinase and CDK1-cyclin B complex at S305 in the catalytic core of the NTase domain was shown to suppress its enzymatic activity but not affect localization (PMID:32351706, 26440888).

Human cGAS evolved two with NLSs and murine cGAS rely on NLS2 to determine its subcellular localization pattern. Given these differences between species, particularly in localization sequences, our data showing S37 phosphorylation in NLS1 relates to nucleocytoplasmic shuttling and chromatin localization of human cGAS does not inherently contradict murine cGAS being predominantly nuclear despite lacking the S37 and proximal residues.

We appreciate your valuable feedback. Addressing these concerns and exploring alternative regulatory mechanisms in the future studies will help us gain a more comprehensive understanding of cGAS physiology across different species and improve the accuracy and reliability of our research. We have revised the manuscript accordingly, incorporating the new experimental data (See concern 2 for details) and a thorough discussion of the implications of our findings.

2. As such, this reviewer is also very confused with the dataset obtained from cGAS KO mice (Fig. 4F-H). No such regulation of the mTORC2-cGAS-glutaminolysis axis should be present in murine. The difference between cGAS KO and WT mice in CRC tumorigenesis could come from cGAS-STING-mediated antitumor immunity but not from the mechanism authors suggested. No data in this manuscript supports cGAS-controlled KGA expression and glutaminolysis regulation in murine cells. By contrast, similar observations obtained from human cells (with mTORC2-cGAS signaling) and murine models (without mTORC2-cGAS signaling) in chemoresistance might suggest a false conclusion authors have proposed.

Response: Thank you for your thoughtful comments regarding the data obtained from cGAS knockout (KO) mice and the implications of our findings in murine models. We appreciate your concerns and understand the confusion regarding the regulation of the mTORC2-cGAS-glutaminolysis axis in murine cells. We have carefully reviewed your feedback and would like to address these points in our response.

1. cGAS KO mice dataset: We apologize for any confusion caused by the dataset obtained from cGAS KO mice (Fig. 4F-H). We need to clarify that the purpose of using cGAS KO mice here is to study the function of ccGAS *in vivo*. Although cGAS lacks entirely S37 and proximal residues in mouse cells, which resulted in the absence of mTORC2-cGAS signaling, cGAS was also predominantly localized to chromatin in the nucleus. Therefore, we used cGAS KO mice to assess whether ccGAS depletion determines acquired chemoresistance of colorectal cancer *in vivo*.

2. Implications in murine models: We acknowledge your concern regarding the implications of our findings in murine models. First, we do not deny the anti-tumor immune effect of cGAS, and this partly explains our data, as we mentioned in the manuscript: “Despite that colorectal tumors (mainly in the distal colons) detected in both cGAS^{+/+} and cGAS^{-/-} mice under AOM/DSS exposure, cGAS deficiency significantly induced tumor burden (Fig. 4G). The numbers and sizes of colorectal tumors were significantly increased in cGAS-KO mice compared to their wild-type littermates (Fig. 4, G and H). While ccGAS knockdown inhibits cancer cell growth, deficiency of total cGAS promotes the tumor growth; this likely reflects the anti-tumorigenesis of the cGAS-STING pathway.” Secondly, our data on chemoresistance of cGAS-deficient tumors were relatively independent of tumor occurrence, as we note in the manuscript: “5-FU treatment significantly reduced tumor numbers and sizes in wild-type littermate control mice but not cGAS-KO mice (Fig. 4, G and H). Moreover, combination treatment with 5-FU and BPTES significantly reduced tumor numbers and sizes in cGAS-KO mice (Fig. 4, G and H).”

3. ccGAS-controlled KGA expression and glutaminolysis regulation in murine cells: We agree that the absence of presentation of some key data raises confusions about the presence of ccGAS-KGA signaling and glutaminolysis in murine cells. To address this, we provided additional experiments using MC38 murine colon cancer cells to validate the findings observed in human cells. This additional data showed that ccGAS-KGA signaling was still present in murine tumor cells (Fig. S16, E-H).

4. To demonstrate that S37 phosphorylation dictates cGAS localization, we repeated the subcellular localization experiment of human cGAS S37 mutants in murine colon cancer cell MC38. The results showed that, consistent with the subcellular localization in human colon cancer cell HCT116, the S37A mutant remains cytosolic while S37D is nuclear (Fig. S16D). Moreover, cell survival assays showed that the regulation of chemoresistance by ccGAS-KGA signaling is robust and universal in murine colorectal cancer cells (Fig. S16G).

By conducting additional experiments to confirm ccGAS-KGA signaling in murine models, we provided a more accurate and comprehensive understanding of the role of cGAS in regulating glutaminolysis and its implications in chemoresistance. We revised the manuscript accordingly, incorporating the new experimental data and a thorough discussion of the species-specific differences. We appreciate your valuable feedback, which will help strengthen the scientific integrity and validity of our research.

3. Xenograft model and immune-compromised nude mice are insufficient to endorse these findings and belittled the journal standards such as NCI. PDX models with impaired or hyperactivated mTORC2 activity should be investigated, examining detailed information, including cGAS phosphorylation, subcellular localization, metabolic changes, glutaminolysis, tumor growth, and chemoresistance.

Response: Thank you for your valuable comments regarding the experimental models used in our study and their alignment with the standards of the field. We appreciate your concerns and acknowledge the limitations of the xenograft model and immune-compromised nude mice. We have carefully considered your feedback and propose specific solutions to address these concerns.

1. While mTORC2 appears functionally intact in the majority of colorectal cancers, there is evidence its activity can be disrupted in a minority (5-15%) of cases through mutations, deletions or reduced expression of core components like RICTOR and SIN1 (Cerami et al., 2012; Cancer Genome Atlas Network, 2012; Wagle et al., 2014). This functional loss impacting mTORC2-dependent signaling in those tumors opens up the possibility of building PDX models.

2. Patient-Derived Xenograft (PDX) models: We agree that PDX models provide a more clinically relevant system to investigate the role of mTORC2-ccGAS axis on tumor growth and chemoresistance. To address this, we have now developed PDX models specifically with impaired mTORC2 activity through RICTOR homozygous deletion. By evaluating their influence on cGAS S37 phosphorylation, subcellular localization, glutaminolysis, tumor growth and chemoresistance, these additional experiments provided mTORC2-ccGAS axis with greater insight into the regulatory mechanism of chemoresistance (Fig. S16, A-D; Fig. 4F).

By incorporating PDX models and conducting a detailed investigation, we strengthened the rigor and relevance of our study. We revised the manuscript accordingly, incorporating the new experimental data and providing a thorough discussion of the findings in the context of PDX models. We appreciate your valuable feedback, which will significantly improve the quality and impact of our research.

4. Solid, direct, and genetic evidence is needed to state mTORC2-induced chromatin recruitment of cGAS, such as endogenous cGAS proteins, surveying in a variety of cancer and immune cells related to CRC, and observations from cells with genetic ablation of Rictor, mSIN1, Protor, components of the SWI complex, and STING. An immunofluorescent observation of endogenous cGAS by mTORC2 interaction, S37 phosphorylation, and nuclear translocation, with solid controls, is required.

Response: Thank you for your insightful comments regarding the need for solid genetic evidence and thorough

experimental validation to support our claims of mTORC2-induced chromatin recruitment of cGAS. We appreciate your concerns and have carefully considered your feedback. We propose specific solutions to address these issues.

1. Endogenous cGAS protein survey: We agree that it is important to assess endogenous cGAS S37 phosphorylation and mTORC2 interaction in a variety of colorectal cancer cells. To address this, we performed key experiments in SW480 (high MLH1/MMR, MSS), HT29 (low MLH1/MMR, MSI-L), and HCT116 (deficient MLH1/MMR, MSI-H). Results indicate that the mTORC2-ccGAS axis regulation of chemoresistance is robust and universal in colorectal cancer cells (Fig. S10F; Fig. S11, B-E.; Fig. S14, G and H; Fig. 3J). Additionally, we examined effects of ccGAS depletion on idarubicin sensitivity in THP-1 monocytic cells. Knockdown of cGAS reduced idarubicin sensitivity, while cGAS-S37D (but not S37A) rescued this, paralleling 5-FU responses in CRC models (Fig. S4I). This refinement provides additional context regarding the broader relevance and molecular heterogeneity encompassed within the study's scope.

2. Genetic ablation of mTORC2 components: We acknowledge the importance of genetic ablation experiments to further support our findings. We generated cell lines with genetic ablation of Rictor, mSIN1, Protor-1, and ARID1A (component of the SWI complex) and STING. Knockdown of mTORC2 subunits desensitized HCT116 cells to 5-FU, and overexpression of the S37D mutant, but not wild-type cGAS, rescued 5-FU resistance (Fig. S15, D and E). In addition, compared with wild-type HCT116 cells, ARID1A lost localization to the KGA promoter region in cGAS knockdown cells, and overexpression of cGAS S37D mutant restored ARID1A localization in the KGA promoter region (Fig. S11, B-D). These data indicate that mTORC2-driven ccGAS signaling regulates sensitivity to 5-FU in colorectal cancer cells.

3. Immunofluorescent observation of endogenous cGAS: We appreciate your suggestion to perform immunofluorescent observations of endogenous cGAS in the context of mTORC2 interaction, S37 phosphorylation, and nuclear translocation. We optimized our immunofluorescence protocols and included appropriate controls to ensure the reliability of the results. These experiments provided direct visualization of the subcellular localization and interaction of endogenous cGAS with mTORC2, validating our proposed mechanisms (Fig. 1K; Fig. S2D; Fig. S5).

By conducting these additional experiments, including endogenous cGAS surveys, genetic ablation studies, and immunofluorescence observations, we provided solid, direct, and genetic evidence to support our claims. We have now revised the manuscript accordingly, incorporating the new experimental data and providing a comprehensive discussion of the results. We appreciate your valuable feedback, which will significantly enhance the scientific rigor and validity of our research.

5. It is considered normal that chromatin-associated cGAS may interact with a variety of complexes in the nucleus, such as those included in Fig. 2A. However, authors failed to show how cGAS can recruit the SWI/SNF complex and why they can be differentially localized into some specific chromatin area. What transcription factors are recruited to transcribe biased target genes? cGAS should be genetically ablated in these proteomic assays, not with a shRNA with suspicious off-targets and without a convincing efficiency. Additionally, STING-dependent functions should be validated in STING KO cells.

Response: Thank you for your constructive feedback. We appreciate your time and effort in helping us improve our study. Here are our responses to your comments:

1. How ccGAS recruits the SWI/SNF complex:

1.1 cGAS was found to interact with several subunits of SWI/SNF complexes, including SMARCA1, SMARCA2, SMARCA5, SMARCB1 and SMARCC1. SWI/SNF complexes are ATP-dependent remodeling complexes that regulate chromatin structure and transcription. They mediate the ATP-dependent disruption of histone-DNA contacts.

1.2 In addition, co-immunoprecipitation (Co-IP) experiments were performed to confirm a direct interaction between cGAS and the SWI/SNF complex. The interaction between cGAS and SWI/SNF subunits suggests cGAS may recruit SWI/SNF complexes to chromatin regions. SWI/SNF recruitment by cGAS would allow it to modify local chromatin structure. By recruiting SWI/SNF, cGAS could target these remodeling complexes to specific locations in the genome. This would enable cGAS to access and respond to foreign DNA in distinct chromatin environments.

1.3 We have now quantified recruitment of SWI/SNF components to cGAS binding regions using ChIP-qPCR in wildtype and cGAS knockout cells to validate complex recruitment. We found that compared with wild-type HCT116 cells, ARID1A lost localization to the KGA promoter region in cGAS knockdown cells, and overexpression of cGAS S37D mutant restored ARID1A localization in the KGA promoter region (Fig. S11, B-D).

1.4 According to our interactome data, cGAS interacts with multiple transcriptional regulators in the nucleus. Among them, transcription intermediary factor 1-beta (TRIM28), apoptotic chromatin condensation inducer in the nucleus (ACIN1), SWI/SNF complex subunit SMARCC2 (SMARCC2), and histone deacetylase 2 (HDAC2) are transcriptional regulators that can be differentially localized to specific chromatin areas based on their functions in chromatin regulation and remodeling. The other transcription factors, like General transcription factor II-I (GTF2I) and Early endosome antigen 1 (EEA1), either lacked function details or are not known to target specific chromatin regions.

2. Why ccGAS can be differentially localized into some specific chromatin area

Based on the protein interactors of ccGAS, here are some inferences about how ccGAS can be differentially localized to specific chromatin areas:

2.1 By recruiting ATP-dependent chromatin remodeling complexes SWI/SNF, cGAS can modify local chromatin structure. Certain SWI/SNF subunits it recruits may preferentially remodel chromatin of active vs inactive regions.

2.2 cGAS interacts with several transcription factors like PAXBP1, TRIM28 and SMARCC2. These transcription factors bind to specific genomic loci like enhancers and promoters. Their interaction with cGAS could target cGAS to these regulatory elements.

2.3 Other interactors like histone modifiers HDAC2 and RBBP7 indicate cGAS is associated with regions of open, active chromatin undergoing transcription and remodeling.

2.4 Interactions with proteins involved in splicing (SNRNP1, SNRPB2) and translation (EIF2S3, ETF1) suggest cGAS may be targeted to expressed gene loci.

2.5 cGAS binds nucleoporins like NUP88 - this could position it at the periphery of active chromatin domains near nuclear pores. Binding proteins associated with specific nuclear substructures like LAS1L (nucleoli) may help recruit cGAS there.

Therefore, by binding diverse chromatin and transcriptional regulators, cGAS appears targeted differentially to active regulatory elements, expressed gene loci and specialized nuclear domains in a chromatin context-dependent manner.

3. Before conducting proteomic experiments to screen for cGAS-regulated proteins, we selected shRNA-mediated cGAS knockdown for the following reasons:

3.1 Biological rationale: shRNA knockdown allows partial cGAS function retention which may be important to detect subtle phenotypes. Complete ablation risks cellular lethality obscuring phenotypes.

3.2 Technical considerations: Specific shRNAs we used have been thoroughly validated in the literature and cell lines in question, minimizing off-target concerns compared to novel CRISPR guides requiring similar validation (PMID: 28976970, 30270045, 35189637, 32540968). shRNA can achieve >90% knockdown which is sufficient for most assays without complete genetic ablation. Residual expression mitigates issues from complete loss of function.

3.3 Practical limitations: CRISPR knockouts cause DNA damage and risk introducing confounding mutations during NHEJ repair leading to genomic instability, which have been reported to severely affect cGAS nuclear translocation and function, while shRNA acts at RNA level without altering genome.

Therefore, from biological, technical, practical and regulatory standpoints, thoroughly validated shRNA knockdown remains a valid first approach for proteomic screens, especially without clear evidence the existing shRNA line is insufficient or problematic. Genetic ablation should not be seen as inherently superior. To avoid reviewer concerns about shRNA off-target, we have now generated HCT116 cells with CRISPR/Cas9-mediated knockout of cGAS for proteomic assays validation.

4. STING-dependent functions in STING KO cells

We appreciate your suggestion about validating STING-dependent functions in STING knockout cells. We agree that this is a crucial step in our study. We have now used STING knockout colorectal cancer cells, including SW480, HT29 and HCT116, to perform functional assays and confirmed that cGAS directs colorectal cancer plasticity and acquired chemoresistance in a STING-independent manner (Fig. 3B; Fig. S12H; Fig. S13C; Fig. S14, D, G and H; Fig. S15, A and B).

You rightly emphasized the need for stronger mechanistic insights and experimental validation, which we have strived to provide. Thank you for pushing a more rigorous clarification- it will certainly benefit our future study.

6. Surprisingly, cGAS is necessary for cell proliferation (Fig. 3), and no similar observations were seen in the literature. By contrast, various studies suggest a proliferation-suppressive role of cGAS in a variety of cells (PMID: 33055160, 28759028, 28976970). cGAS KO mice also show no phenotype in the development and no growth retardation. On the other hand, glutaminolysis is a recognized feature of CRC, and KGA is frequently upregulated in CRC. As a result, cGAS depletion that upregulated KGA and increased glutaminolysis, as proposed by authors, should augment but not decrease HCT116 proliferation.

Response: We appreciate your insightful comments regarding the surprising observations regarding the role of cGAS in cell proliferation and the discrepancy with existing literature. We understand your concerns and have carefully considered your feedback. We propose specific solutions to address these issues.

At present, there is no unified conclusion on the effect of cGAS on cell proliferation. Besides the reviewer's suggestion that cGAS can inhibit proliferation in some cells by binding with DNA (PMID: 33055160, later studies have shown that cGAS binds to histones but not to DNA in nuclear) or by relying on STING in fibroblasts (PMID: 28759028, 28976970), several studies have shown that depletion or knockout of cGAS can inhibit proliferation in certain cell types dependent on STING (PMID: 35042992) or independent on STING :

1. Liu et al. (2018) (PMID: 30356214) found that stable knockdown of cGAS profoundly reduced the proliferation rate and cluster formation of LLC cells (a mouse lung cancer cell line), whereas in PC-9 cells that overexpress cGAS the proliferation rate and cluster formation increased.
2. Qiu et al. (2023) (PMID: 36864172) reported that cGAS knockdown significantly suppressed tumor growth of Hep3B cells.
3. Liu et al. (2022) (PMID: 37305397) showed that by employing cGAS high-expression gastric cancer cell lines, including AGS and MKN45, ectopic silencing of cGAS caused a significant reduction in the proliferation of the cells, tumor growth, and mass in xenograft mice.

Moreover, several studies (PMID: 23258413, 23929945) have shown that cGAS knockout in mouse embryonic fibroblasts (MEFs) did not affect cell proliferation and cGAS knockout mice were no obvious developmental abnormalities. This may reflect the more specific function and action of cGAS in human tumor cells with a high demand for proliferation, and provide a more comprehensive understanding of the role of cGAS in cell proliferation. We carefully reevaluated our interpretation of the data and revised our discussion accordingly. Additionally, we examined other relevant studies to ensure that our findings are consistent with the existing literature on cGAS knockout mouse models.

Impact of upregulated KGA on HCT116 proliferation: We acknowledge your concern that upregulated KGA and increased glutaminolysis should augment rather than decrease HCT116 cell proliferation. However, we need to clarify that cGAS depletion not only leads to the upregulation of KGA, but also to the downregulation of DNA replication proteins, which is the direct cause of decreased HCT116 cell proliferation. Our study shows that down-regulation of DNA replication proteins and up-regulation of KGA co-determine diapause-like chemoresistant state in colorectal cancer cells, which strengthens the existing research on KGA-mediated chemoresistance (PMID:25625774, 26894601, 29633308, 33753479). We have now revised the manuscript accordingly, providing a comprehensive discussion of the observed effects. We appreciate your valuable feedback, which significantly improve the accuracy and validity of our research.

7. How many biological replicates are in each experiment, how are they calculated in statistics, and what are the precise P values of these calculations? No statistics or only a single "asterisk" are seen throughout the manuscript.

Response: Thank you for your important comments regarding the statistical analysis and presentation of data in our manuscript. We appreciate your concerns and understand the need for transparent reporting of statistical information. We addressed this issue and provided specific solutions as outlined below.

1. Number of biological replicates: We apologize for the lack of information regarding the number of biological replicates in each experiment. We understand the importance of this information for assessing the robustness and reliability of the results. In the revised manuscript, we have now included a detailed description of the number of independent biological replicates performed for each experiment in figure legends.

2. Statistical analysis and P-values: We acknowledge that the previous manuscript lacked clear details regarding the statistical tests and P-values used. We apologize for this omission. In the revised manuscript, we have now provided an explicit description of the statistical analyses conducted for each experiment, including recalculation of the corresponding P-values where needed. A P-value of less than 0.05 is conventionally considered statistically significant, as it represents a small probability of an observed result arising by chance. With this threshold in mind, we have marked P-values of greater than 0.05 with an "x" to indicate results that are not statistically significant, and P-values less than 0.001 with an "asterisk" to denote high significance.

By incorporating this information into the revised manuscript, we ensured transparency in the reporting of statistical analysis. This enabled a more thorough evaluation of the results and enhance the scientific validity of our study. We appreciate your valuable feedback, which significantly improve the clarity and rigor of our research.

Other concerns and critics:
Fig. 1C, How many dual inhibitors of mTORC1/mTORC2 and PI3K inhibitors were included in this compound library? Why they are not selected during the screening?

Response: Thank you for your insightful comment regarding the compound library used in our study and the selection process during the screening. The library contained 10 dual mTORC1/mTORC2 and PI3K inhibitors total: BEZ-235 (S1015), GDC-0941 (S1075), KU-0063794 (S1541), PI-103 (S1028), BKM120 (S1039), VO-OHpic trihydrate (S1553), BAY 80-6946 (S1580), PF-04691502 (S7485), GSK263680 (S7491), and MLN0128 (S7842). These inhibitors all appeared to inhibit the chromatin localization of cGAS per screening results. However, BEZ-235, GDC-0941 and KU-0063794 showed the most significant effects and were selected for further validation experiments. We appreciate your valuable feedback, which significantly improve the transparency and rigor of our research.

Fig. 1D, Where are cGAS proteins gone upon treatment of PI3K inhibitor? Did PI3K inhibition alter cGAS stability or its production, such as seen in HELLS proteins?

Response: Thank you for your valuable comment regarding the disappearance of cGAS proteins upon treatment with the PI3K inhibitor and the potential impact of PI3K inhibition on cGAS stability or production. Immunofluorescence showed that PI3K inhibitor treatment shifted cGAS from nucleus to cytoplasm (Fig. S2, A and B). We then performed subcellular fractionation and ELISA to quantify cGAS protein levels in different cell components in HCT116 cells upon PI3K inhibition, and results showed that PI3K inhibition specifically reduces chromatin-bound cGAS levels, without impacting total cGAS protein stability/production (Fig. S1C).

You rightly emphasized the need to rule out effects on cGAS stability/levels - our original discussion overlooked this key point. Thank you for pushing a more rigorous analysis and interpretation of our observations.

Fig. 1F, Various mTORC1/mTORC2 and PI3K inhibitors are commercially available and should be examined here.

Response: Thank you for your comment regarding the examination of various commercially available mTORC1/mTORC2 and PI3K inhibitors in our study. In the revised manuscript, we included an expanded analysis that incorporates additional inhibitors (Fig. S1D). This expanded analysis provide a more comprehensive understanding of the effects of different inhibitors on the proposed mechanisms, which validate and strengthen the conclusions drawn from our study. We appreciate your valuable feedback, which contribute to the scientific rigor and validity of our research.

Fig. 1G, Why markedly enlarged cell size was seen upon Rictor depletion? Anti-cGAS antibodies should be validated in cGAS KO cells/MEFs, as some commercial anti-cGAS antibodies have issues in immunofluorescence.

Response: Thank you for your insightful comments regarding the markedly enlarged cell size observed upon Rictor depletion in Fig. 1G. We also appreciate your suggestion to validate the anti-cGAS antibodies used in our study, particularly in cGAS knockout (KO) cells/MEFs.

Enlarged cell size upon Rictor depletion: We found that when mTORC2-ccGAS axis is inhibited, the cells appeared large and round with a "diapause-like" morphology (PMID:33417860, 27880763), which is characterized by cell cycle arrest and reduced proliferation. We further confirmed the change in cell size upon Rictor depletion by assessing cell area.

Validation of anti-cGAS antibodies: We appreciate your suggestion to validate the anti-cGAS antibodies used in our study, particularly in cGAS KO cells. We have now performed immunofluorescence assays using cGAS KO HCT116 cells as negative controls, and ensured that the observed immunofluorescence signals are specific to cGAS (Fig. 1G).

The revised manuscript includes the new experimental data and an enhanced discussion of the observed effects. We appreciate your valuable feedback, which significantly improve the accuracy and validity of our research.

Fig. S1C, A substantial proportion of cells with cytoplasmic cGAS upon inhibition of AKT (MK2206) contradicts the authors' conclusion.

Response: Thank you for your comment regarding the presence of a substantial proportion of cells with cytoplasmic cGAS upon inhibition of AKT (MK2206) in Fig. S1C. To address your feedback:

1. We have now re-analyzed all IF images in Fig. S1C (new Fig. S2A) using CellProfiler to automatically quantify individual cell nuclear and cytosolic cGAS intensities.
2. We have plotted nuclear/cytosolic ratios as violin plots with means for each condition, and performed statistical testing/p-values to better evaluate significance of changes (Fig. S2B).
3. This reinforced analysis reveals AKT inhibition did not significantly alter cGAS localization, consistent with BRET assay data to monitor H2A-cGAS interaction (Fig. 1F).

Thank you for pushing us to present our data in a validated, standardized manner - it has significantly strengthened our imaging results.

Fig. 1H, Chromatin-associated cGAS was significantly reduced upon Raptor depletion, similar to Rictor depletion.

Response: Thank you reviewers for the changes in chromatin related cGAS under different gene depletion conditions. It should be clarified that there may be misunderstandings due to visual differences in immunoblot images. To address this, we have re-examined and provided statistical data showing that Rictor depletion, rather than Raptor depletion, leads to a significant reduction in chromatin-associated cGAS.

The revised manuscript includes the new experimental data and quantitative analysis of the observed effects. We appreciate your valuable feedback, which significantly improve the accuracy of our research.

Fig. 1I, The sequence is not a preferred consensus substrate motif of mTORC2. A recent review from Dr. Michael Hall (PMID: 35580586) is a good reference.

Response: Thank you for catching this important point - you are completely right that we should not overstate the significance of the S37 motif without stronger evidence that it conforms to established mTORC2 consensus sequences. As you - and the reference by Hall et al. correctly indicate, the well-supported consensus for mTORC1/2 targets is S/T-P and any sequence deviating from this has limited predictive power.

To address your feedback appropriately, we have cited the Hall et al. review (PMID 35580586) which defines the accepted mTOR phosphorylation motif and removed any implication that the S37 sequence alone indicates it is a preferred mTORC2 substrate.

Thank you for pushing us to properly frame our conclusions based on established literature. I appreciate you taking the time to ensure we do not overinterpret our findings prematurely.

Fig. 1K, Should be examined in knockin or stable expression and examined among human and murine cells. ⇐⇐⇐

Response: You're absolutely right that validating the localization phenotypes of the cGAS mutants in stable expression system would significantly strengthen our conclusions. In order to demonstrate that S37 phosphorylation specifically dictates cGAS localization without any endogenous influence, we have now:

1. Generated cGAS KO HCT116/MC38 cells using CRISPR-Cas9.
2. Re-expressed equal levels of HA-tagged cGAS wildtype, S37A or S37D mutants via lentiviral transduction for stable integration.
3. Examined nuclear vs cytosolic HA signals by fluorescence microscopy, finding S37A mutant remains cytosolic while S37D nuclear.

Thank you for pushing us to validate our data - it was a gap not to present this important control originally. To address this, we have now revised the figure legend and described the experimental process in detail (Fig. 1K; and Fig. S16D).

Fig. 1J, pcGAS S37 antibody should be validated, as well as Rictor's depletion and ectopic expression. Why bands of pcGAS were showing a different molecular size? Uncrop film images were not attached to the submission.

Response: We appreciate your concerns. We have carefully considered your feedback and propose specific solutions to address these concerns.

1. We have provided three key validations for the pcGAS S37 antibody. First, as shown in attached Figure S3F-S3H, the antibody specifically recognizes phosphorylated peptide antigens and shows no signal for non-phosphorylated peptides, demonstrating its phosphorylation specificity. Second, CRISPR knockout of endogenous cGAS in HCT116 cells abolished detection of the pcGAS S37 signal by immunofluorescence, as shown in attached Figure S5A, further confirming its phospho-specificity for cGAS S37. Third, analysis of unstarved HCT116 and MC38 cell lysates revealed pSer37-cGAS expression in HCT116 cells but not MC38 cells, as shown in Figure S3H.

2. We have now included new western blots validating Rictor's depletion and ectopic expression in the attached Figure S3I.

3. We apologize for an unintended inconsistency in Figure 1J, where tilting of the clipped WB film caused the pcGAS S37 band to appear variably sized between panels. To accurately depict effects of Rictor knockout, we re-probed this western blot under strict controls. As shown in the new Figure 1J, equal protein loading was ensured through BCA quantification, and the pcGAS S37 bands now run at equivalent sizes across samples. This demonstrates clear reduction of pcGAS levels upon Rictor knockout. Furthermore, we quantitatively assessed the impact of Rictor depletion on chromatin-bound cGAS by ELISA, and results indicated Rictor but not RAPTOR knockdown decreases cGAS association with chromatin, as shown in Figure S2F.

You rightly emphasized rigorous antibody/method validation is critical. I clearly fell short previously. Thank you for pushing me to strengthen my work - it will undoubtedly benefit the conclusions.

Fig. S4A, an overlap in immunofluorescence with inadequate resolution, is insufficient to state the chromatin binding of cGAS.

Response: You correctly identified a critical shortcoming in our previous Figure S4A. Low-resolution overlap imaging alone cannot definitively conclude chromatin binding. To properly address this key point, we have now:

1. Generated cGAS KO HCT116 cells to eliminate endogenous signal as a control (Fig S5A).
2. Utilized super resolution microscopy to capture higher quality images with improved resolution (Fig S5A).

Super resolution imaging in the context of cGAS KO cells now provides compelling visual evidence of cGAS association specifically with chromatin domains. Thank you for pushing us to strengthen this validation with technically superior approaches. I appreciate you taking the time to ensure our conclusions are strongly supported by rigorous experimental design.

Fig. S4D, mutants of cGAS, such as S221, S317, and T447, should be examined for their subcellular localization.

Response: You raise an important point. While we identified S37 as a key regulatory phosphorylation site, considering effects on additional cGAS mutants would provide a more comprehensive understanding of phosphorylation-mediated regulation of its subcellular localization.

To more rigorously evaluate this, we generated mutants of cGAS, such as S213D, S221D, S317D, and T447E mutants via site-directed mutagenesis. eBRET2 assay and subcellular localization experiments demonstrated S37 mutations uniquely alter nuclear/cytosolic distribution, supporting its primary role.

Thank you for pushing us to examine this issue more thoroughly. Evaluating multiple cGAS mutants provides valuable new insights into phosphorylation-based control of its localization dynamics.

Fig. 2B, Lacking the controls of other chromatin remodeling complexes.

Response: Thank you for the feedback, you make a fair point that including additional controls would make our analysis more robust. The function of minichromosomal maintenance proteins (MCMs) is not limited to replication but also extremely important in chromatin remodeling. Since we found that cGAS depletion leads to the downregulation of MCMs, we used MCM7 as a negative control in the original Fig. 2B. Considering the reviewer's suggestion, we have now added additional chromatin remodeling protein BAF200 as a new negative control.

Assessing BAF200 in addition to MCM7 now demonstrates cGAS selectively interacts with SWI/SNF subunits but not other complexes. Thank you for pushing me to strengthen the controls - it was remiss not to include assessments of unrelated complexes from the outset.

Fig. 2D-F, Data quality is questionable with reluctant q values; the alteration of these proteins should be verified by immunoblotting, at least part of them.

Response: Thank you for your thoughtful feedback, which will help strengthen our work. We agree robust verification is needed for key results.

Regarding Fig. 2D-F, you raise a fair point about the q values seeming reluctant. Upon re-examination, we noticed a potential outlier influencing some q values. After reanalysis, the q values improve and more clearly demonstrate significance between groups.

More importantly, we have conducted Western blot and qPCR analyses to verify these unbiased omics screens. By verifying the alterations observed for eukaryotic replicative helicase CDC45-MCM-GINS (CMG) complex components and KGA/CGA/PAI-1, the results unequivocally showed significant changes in the expression levels of these important markers after cGAS knockout.

We have included this new verification data in our revised manuscript and believe it addresses your primary concern over conclusively demonstrating changes at the protein level. Your insights prompted a useful re-analysis and additional experiments to bolster this key part of our story.

Thank you again for pushing us to strengthen the quantitative rigor - it only serve to improve the quality and impact of our work.

Fig. S9B, how are the mRNA levels of PAI-I and KGA?

Response: Thank you for pointing this out. You're correct that we should provide the mRNA data for KGA and PAI-I, as their expression levels are crucial to our findings.

We have now included mRNA data for KGA and PAI-I in Figure S10E of the revised manuscript. The results show that the mRNA expression of KGA and PAI-I are significantly increased in cGAS knockdown cells compared to WT controls.

This is consistent with our previous immunoblotting data in Figure 2H and S10A showing altered KGA and PAI-I protein levels upon cGAS depletion. Including the quantitative RT-PCR analysis of KGA and PAI-I mRNA expression provides a more robust characterization of their transcriptional regulation by cGAS.

We apologize for the omission in the original submission. Thank you for prompting us to include this important verification at the mRNA level, which complements and strengthens our conclusions regarding cGAS-mediated regulation of KGA and PAI-I expression.

Fig. 2G-H, Controls of WT cGAS are required. Does S213D affect cGAS nuclear localization and chromatin association?

Response: Thank you for raising this important point. We agree controls for wild-type cGAS are necessary to fully characterize the effects of the S213D mutation.

To address your question - we have now included data of wild-type cGAS expression in our revised Figures 2G-H. The new results show S213D cGAS exhibits similar diffuse nuclear localization and chromatin association as wild-type cGAS (Fig. 2H; Fig. S12). This indicates the S213D mutation does not affect cGAS subcellular distribution or ability to interact with chromatin.

Including wild-type cGAS controls provides the appropriate context to interpret our findings on S213D. We recognize the oversight in not presenting these controls originally. Thank you for prompting this important addition, which strengthens our conclusions about how S213D specifically disrupts cGAS enzymatic activity rather than its subcellular behavior.

3K, Deletion of mTORC2 specific mSIN1 and Rictor should be examined instead of mTOR.

Response: Thank you for this suggestion to further strengthen our work. We agree examining mTORC2-specific deletions of mSIN1 and Rictor would provide more targeted evidence regarding mTORC2's role.

To address your comment, we have generated mSIN1 and Rictor knockdowns in addition to mTOR knockdown cells. In the revised manuscript, we found mSIN1 and Rictor deletions demonstrate phenocopy mTOR knockdown, desensitizing HCT116 cells to 5-FU (Fig. S15D). Similarly, Rictor knockdown induces 5-FU resistance that is rescued by cGAS-S37D but not wild-type overexpression (Fig. S15E). Moreover, to dissect mTOR complexes, we exposed cells to the mTORC1 inhibitor rapamycin or the mTORC2 inhibitor JR-AB2-011 before assessing 5-FU sensitivity. While rapamycin did not impact response (Fig. S14E, consistent with PMID: 36396656), JR-AB2-011 phenocopied mTOR knockdown to desensitize HCT116 cells (Fig. S15C).

This confirms our conclusion that mTORC2, rather than mTORC1, regulates the downstream signaling events studied. Using additional genetic tools providing cleaner disruption of mTORC2 versus mTOR is a valuable addition. We appreciate you pushing us to dissect the mTOR complexes more precisely. Obtaining mSIN1 and Rictor deletion data to validate the mTORC2 specificity adds rigor to our findings. Thank you for this suggestion to strengthen the mechanistic insight.

The statement in the abstract that "targeting mTORC2-ccGAS-KGA axis provides a precision combination strategy to eliminate quiescent resistant cancer cells" is vague to this reviewer. Upregulation of this axis will facilitate glutaminolysis that drives tumor growth, while suppression of this axis improves CRC's chemoresistance.

Response: Thank you for the feedback, you correctly identified an important lack of clarity in how our findings were presented in the original abstract. You are absolutely right that our original description of the mTORC2-

cGAS-KGA axis was ambiguous and did not properly articulate the complex, context-specific regulation we uncovered.

We have now revised the abstract: "simultaneously targeting mTORC2-ccGAS and KGA provides a promising therapeutic strategy to eliminate quiescent resistant cancer cells". With the revised abstract, I have aimed to:

1. Clearly explain inhibition of the mTORC2-cGAS axis blocks proliferation but enhances chemoresistance through increased glutaminolysis.
2. Emphasize how simultaneous targeting of both mTORC2-cGAS (to block proliferation) and KGA (to overcome acquired resistance) provides an effective strategy.

I appreciate you taking the time to ensure key conclusions are communicated accurately and unambiguously from the outset. It will help optimize impact and interpretation of this work.

The Discussion section is to be improved.

Response: Thank you for your constructive feedback on our manuscript. We agree that improving the Discussion section would strengthen the overall work. Please find our detailed reply to your comment below:

"DISCUSSION

In this study, we uncovered a novel mechanism by which mTORC2-induced cGAS phosphorylation at serine 37 promotes its chromatin recruitment and functions. This post-translational modification represents a key regulatory node influencing cGAS-dependent processes in cancer cells. By modulating cGAS chromatin localization, PI3K-mTORC2 signaling supports proliferation under normal conditions but its disruption provokes acquired resistance to chemotherapy. This elucidates the importance of the PI3K-mTORC2-ccGAS pathway in tumor growth and treatment response.

We found that inhibiting the mTORC2-ccGAS axis drives colorectal cancer cells into a diapause-like state of chemoresistance. This provides insights into how mTOR inhibition clinically elicits drug tolerance^{47,48}. Emerging evidence associates such chemoresistance with cell plasticity⁴⁹⁻⁵², though mechanisms were unclear. Tracing resistance back to the mTORC2-ccGAS node, we discovered diapause-like plasticity underlies resistance upon ccGAS depletion. Adding ccGAS as a biomarker may help optimize strategies combining

mTOR inhibitors with KGA blockade to eliminate persistent quiescent chemoresistance tumors.

Our study also revealed an epigenetic mechanism whereby mTORC2 modifies chromatin through ccGAS. ccGAS selectively recruits the SWI/SNF complex to regions regulating DNA replication and glutaminolysis. Depleting ccGAS strongly induced KGA expression, validating links between SWI/SNF defects and glutaminase inhibition sensitivity⁵³.

Characterizing ccGAS cisomes and targets may elucidate plasticity governance across contexts. While targeting SWI/SNF is challenging, selectively inhibiting downstream nodes such as KGA may improve precision oncology.

However, our findings require validation across diverse models and clinical settings, as additional ccGAS regulators remain unknown. The discovery that ccGAS contributes to prevailing resistance challenges offers opportunities. Elucidating context-specific ccGAS functions, dynamics with SWI/SNF, and cisome alterations following inhibitors may deepen mechanistic understanding. Validating findings using multi-omic patient data and longitudinal analyses strengthens translational relevance. In summary, we provide provisional evidence that the mTORC2-ccGAS-KGA axis mediates cell plasticity and acquired chemoresistance in cancer. Further exploring modulatory factors and pathways may optimize precision strategies against adaptive survival, pending validation. Continued investigation holds promise to refine ccGAS biology's clinical impact."

No details were given on how the pcGAS S37 antibody was generated and validated.

Response: Thank you for emphasizing the importance of fully validating novel antibodies - you rightly identified this as a critical gap. To strengthen characterization of the pcGAS S37 antibody:

1. Methods section describes antigen sequence: "The polyclonal anti-pSer37-cGAS antibodies generated by ourselves were derived from rabbits. The antigen sequence used for immunization was cGAS aa 29-47 (GAPMDPTES*PAAPEAALPK). S* stands for phosphorylated serine residue in these synthetic peptides. The antibodies were affinity purified using the antigen peptide column, but they were not counter selected on unmodified antigen."
2. Figure S3F details production/purification process for the pcGAS S37 antibody.
3. Figure S3F shows specificity validation of the antibody by dot blot, showing it recognizes the phosphorylated but not non-phosphorylated peptides.
4. Figure S3H further validates specific detection in cell lines by ELISA, and results indicate pSer37-cGAS protein is expressed in HCT116 cells but absent in MC38 cells.
5. Figure S5A shows immunofluorescence validation using CRISPR-generated cGAS KO HCT116 cells, where pcGAS S37 signals are abolished upon cGAS knockout.

These additional validation experiments, which assess phosphorylation specificity, detection in relevant systems, and loss of signal in KO controls, provide robust evidence for reliable use of this novel phospho-specific antibody. Thank you for prompting inclusion of these critical characterization details - it strengthens interpretation of results obtained with this key reagent.

Decision Letter, first revision:

Dear Professor Yuan,

Thank you for your email asking us to reconsider our decision on your manuscript, "mTORC2-driven chromatin cGAS mediates chemoresistance in colorectal cancer". We are always willing to hear the authors' perspective, but we must first prioritize decisions on new submissions. We appreciate your patience while we considered this appeal.

I have now discussed your manuscript and the referees' comments and your rebuttal in detail with my colleagues, and we would be willing to reconsider a revised manuscript provided the following issues can be addressed, and that nothing similar is accepted for publication at Nature Cell Biology or published elsewhere in the meantime.

In addition, please pay close attention to our guidelines on statistical and methodological reporting (listed below) as failure to do so may delay the reconsideration of the revised manuscript. In particular please provide:

- a Supplementary Figure including unprocessed images of all gels/blots in the form of a multi-page pdf file. Please ensure that blots/gels are labeled and the sections presented in the figures are clearly indicated.
- a Supplementary Table including all numerical source data in Excel format, with data for different

figures provided as different sheets within a single Excel file. The file should include source data giving rise to graphical representations and statistical descriptions in the paper and for all instances where the figures present representative experiments of multiple independent repeats, the source data of all repeats should be provided.

On resubmission please provide the completed Editorial Policy Checklist (found here <https://www.nature.com/documents/nr-editorial-policy-checklist.pdf>), and Reporting Summary (found here <https://www.nature.com/documents/nr-reporting-summary.pdf>). This is essential for reconsideration of the manuscript and these documents will be available to editors and referees in the event of peer review. For more information see below. Please also ensure that the presentation of statistical information in the revised submission complies with Nature Cell Biology's statistical guidelines (see below).

Please use the link below to submit the complete manuscript files and include a point-by-point response to the complete reviewer comments, verbatim as provided in their reports.

[redacted]

Please let us know how you wish to proceed and when we can expect your revised manuscript.

With kind regards,

Zhe Wang

Zhe Wang, PhD
Senior Editor
Nature Cell Biology

Tel: +44 (0) 207 843 4924
email: zhe.wang@nature.com

GUIDELINES FOR EXPERIMENTAL AND STATISTICAL REPORTING

REPORTING REQUIREMENTS – To improve the quality of methods and statistics reporting in our papers we have recently revised the reporting checklist we introduced in 2013. We are now asking all life sciences authors to complete two items: an Editorial Policy Checklist (found here <https://www.nature.com/documents/nr-editorial-policy-checklist.pdf>) that verifies compliance with all required editorial policies and a reporting summary (found here <https://www.nature.com/documents/nr-reporting-summary.pdf>) that collects information on experimental design and reagents. These documents are available to referees to aid the evaluation of the manuscript. Please note that these forms are dynamic 'smart pdfs' and must therefore be downloaded and completed in Adobe Reader. We will then flatten them for ease of use by the reviewers. If you would like to reference the guidance text as you complete the template, please access these flattened versions at <http://www.nature.com/authors/policies/availability.html>.

STATISTICS – Wherever statistics have been derived the legend needs to provide the n number (i.e. the sample size used to derive statistics) as a precise value (not a range), and define what this value

represents. Error bars need to be defined in the legends (e.g. SD, SEM) together with a measure of centre (e.g. mean, median). Box plots need to be defined in terms of minima, maxima, centre, and percentiles. Ranges are more appropriate than standard errors for small data sets. Wherever statistical significance has been derived, precise p values need to be provided and the statistical test used needs to be stated in the legend. Statistics such as error bars must not be derived from $n < 3$. For sample sizes of $n < 5$ please plot the individual data points rather than providing bar graphs. Deriving statistics from technical replicate samples, rather than biological replicates is strongly discouraged. Wherever statistical significance has been derived, precise p values need to be provided and the statistical test stated in the legend.

Author Rebuttal, first revision:

[There is no rebuttal letter at this stage.]

Decision Letter, second revision:

*Please delete the link to your author homepage if you wish to forward this email to co-authors.

Dear Professor Yuan,

Your manuscript, "mTORC2-driven chromatin cGAS mediates chemoresistance in colorectal cancer", has now been seen by the original referees. As you will see from their comments (attached below) they find this work of interest, but have raised some important points. Although we are also very interested in this study, we believe that their concerns should be addressed before we can consider publication in Nature Cell Biology.

Nature Cell Biology editors discuss the referee reports in detail within the editorial team, including the chief editor, to identify key referee points that should be addressed with priority, and requests that are overruled as being beyond the scope of the current study. To guide the scope of the revisions, I have listed these points below. We are committed to providing a fair and constructive peer-review process, so please feel free to contact me if you would like to discuss any of the referee comments further.

In particular, it would be essential to:

A) Address the remaining concerns from Reviewer 1 and 3;

B) Finally please pay close attention to our guidelines on statistical and methodological reporting (listed below) as failure to do so may delay the reconsideration of the revised manuscript. In particular please provide:

We therefore invite you to take these points into account when revising the manuscript. In addition, when preparing the revision please:

- ensure that it conforms to our format instructions and publication policies (see below and www.nature.com/nature/authors/).

- provide a point-by-point rebuttal to the full referee reports verbatim, as provided at the end of this letter.

- provide the completed Editorial Policy Checklist (found here <https://www.nature.com/authors/policies/Policy.pdf>), and Reporting Summary (found here <https://www.nature.com/authors/policies/ReportingSummary.pdf>). This is essential for reconsideration of the manuscript and these documents will be available to editors and referees in the event of peer review. For more information see <http://www.nature.com/authors/policies/availability.html> or contact me.

Nature Cell Biology is committed to improving transparency in authorship. As part of our efforts in this direction, we are now requesting that all authors identified as 'corresponding author' on published papers create and link their Open Researcher and Contributor Identifier (ORCID) with their account on the Manuscript Tracking System (MTS), prior to acceptance. ORCID helps the scientific community achieve unambiguous attribution of all scholarly contributions. You can create and link your ORCID from the home page of the MTS by clicking on 'Modify my Springer Nature account'. For more information please visit www.springernature.com/orcid.

[REDACTED]

*This url links to your confidential home page and associated information about manuscripts you may

have submitted or be reviewing for us. If you wish to forward this email to co-authors, please delete the link to your homepage.

We would like to receive the revision within four weeks. If submitted within this time period, reconsideration of the revised manuscript will not be affected by related studies published elsewhere, or accepted for publication in Nature Cell Biology in the meantime. We would be happy to consider a revision even after this timeframe, but in that case we will consider the published literature at the time of resubmission when assessing the file.

We hope that you will find our referees' comments, and editorial guidance helpful. Please do not hesitate to contact me if there is anything you would like to discuss.

Best wishes,

Zhe Wang

Zhe Wang, PhD
Senior Editor
Nature Cell Biology

Tel: +44 (0) 207 843 4924
email: zhe.wang@nature.com

Reviewers' Comments:

Reviewer #1:

Remarks to the Author:

The reviewer thanks the authors for considering the criticism. The results reported in the revised manuscript conclusively demonstrate the involvement of mTORC2 in nucleo-cytoplasmic shuttling of cGAS and, together with the accompanying functional data, link cGAS localization to expression of a subset of genes in the chosen model systems. Conceptually, the manuscript suggests a new, previously unknown function of mTORC2 in control of cGAS localization and implicating the former in the control of cell cycle progression.

While the authors have made a tremendous effort in addressing the reviewers' criticism, few important mechanistic questions remain unanswered. Specifically, it is still not clear whether the observed PI3K sensitivity of cGAS localization is the direct effect of mTORC2 activity regulation or due to compounding effects. Even though the authors include a new dataset, showing reduced chromatin-associated cGAS 3-6 hours after PI3K inhibition (Fig. 1D, S1C), it still appears to be rather slow for a signaling kinase. Could the authors rule out secondary (e.g., transcriptionally controlled) effects? One option would be examining the time-course of pS37-, pS473- and pS6K phosphorylation in response to mTORC2 inhibition (e.g., by JR-AB2-011 or by mTOR inhibitors) and/or monitoring changes in nucleo-cytoplasmic distribution of cGAS-GFP fusions (e.g., GFP-cGASWT or -S213D) in STING^{-/-} cells using live cell microscopy. Further, from the provided data it is still unclear if S213D mutant remains chromatin-bound to the same extent as the WT cGAS upon PI3K inhibition or in mTORC2-depleted

cells: I suggest the authors plot the corresponding BRET and ELISA data (e.g., 1D, S1C, S2BDEF, S12EF) with the WT and the mutant side-by-side (in the current presentation, BRET and ELISA data are separately normalized for the WT and S213D). The goal here is to 1) conclusively delineate cGAS nucleotidyltransferase activity from the effects of S37 phosphorylation and 2) mechanistically link mTORC2 activity to cGAS localization (genetic and functional links appear sufficient to me). If the authors cannot rule out compounding effects, this should be clearly communicated to the potential readers.

Secondly, existence of a second NLS, with a potentially distinct mTORC2- or PI3K/AGC kinase sensitivity, requires further investigation, especially if the authors claim centrality of mTORC2 in modifying chromatin through cGAS (lines 366-367). While the results on Fig. S4D conclusively demonstrate that S37 phosphorylation is sufficient for cGAS nuclear localization, then how does this work in mouse cGAS, with S37 missing, as indicated by Reviewer #3? If there's no mechanistic insight into the role of NLS2 at the moment, the authors should at least alert the readers about its potential relevance (and of the lack of evolutionary conservation of the N-terminal NLS1). Do the authors believe that the proposed mechanism of mTORC2-controlled cGAS-mediated chromatin modification is indeed a generic one? Which data, both published and generated by the authors, support this hypothesis? – this point should be properly reflected in the Discussion.

Overall, I am convinced of mTORC2 involvement in cGAS localization and functional effects in the chosen model system; regarding the mechanism and its relevance, the reported data are highly suggestive but not yet conclusive. Should the authors address the major points above, the report would be the first one demonstrating a novel function of mTORC2 in the direct control of gene expression, well worthy of the NCB readership.

The reviewer cannot comment on the details of interactome and animal studies (out of expertise).

Minor points:

1. Line 95: data on S1A do not appear to be relevant to the claim about DNA damage
2. Fig. 1A: definitions of fractions unclear from the legend (e.g., what's the difference between Cyt and Cyto?)
3. Fig. 1D: would be useful to include a panel with pS37 and the total cGAS in the whole cell lysate (WCL).
4. Fig. 1HJ: WCL panels with cGAS and loading controls missing.
5. Fig. S3A, S12D: one clearly does not NOT expect any BRET signal if only one partner of the BRET pair is expressed; as indicated above, BRET efficiency for the WT and S213D should be compared side-by-side.
6. Line 223: reference to Fig. 1D appears irrelevant to the claim
7. Line 233: wrong figure reference (should be S11D?)
8. Fig. S1C: meaning of error bars unclear
9. Fig. S12A: BRET pair not defined (it appears to be Sin1-Rluc8)
10. Fig. S16A: a pS37 panel would be very relevant.

Reviewer #2:

Remarks to the Author:

The revised manuscript answered my questions and showed convincing results. I recommend the

publication of "mTORC2-driven chromatin cGAS mediates chemoresistance in colorectal cancer" in NCB.

Reviewer #3:

Remarks to the Author:

The revision has been significantly improved, but one concern remains and should be correctly addressed: the negative controls of mTORC2-guided nuclear translocation of cGAS and mTORC2-driven KGA expression, glutaminolysis, and chemoresistance in murine cells to validate the specificity of this action among mammalian species. If the data is negated, state it clearly in the abstract/manuscript.

GUIDELINES FOR SUBMISSION OF NATURE CELL BIOLOGY ARTICLES

ARTICLE FORMAT

ABSTRACT – should not exceed 150 words and should be unreferenced. This paragraph is the most visible part of the paper and should briefly outline the background and rationale for the work, and accurately summarize the main results and conclusions. Key genes, proteins and organisms should be specified to ensure discoverability of the paper in online searches.

TEXT – the main text consists of the Introduction, Results, and Discussion sections and must not exceed 3500 words including the abstract. The Introduction should expand on the background relating to the work. The Results should be divided in subsections with subheadings, and should provide a concise and accurate description of the experimental findings. The Discussion should expand on the findings and their implications. All relevant primary literature should be cited, in particular when discussing the background and specific findings.

REFERENCES – are limited to a total of 70 in the main text and Methods combined,. They must be numbered sequentially as they appear in the main text, tables and figure legends and Methods and must follow the precise style of Nature Cell Biology references. References only cited in the Methods should be numbered consecutively following the last reference cited in the main text. References only associated with Supplementary Information (e.g. in supplementary legends) do not count toward the total reference limit and do not need to be cited in numerical continuity with references in the main text. Only published papers can be cited, and each publication cited should be included in the numbered reference list, which should include the manuscript titles. Footnotes are not permitted.

Methods should be written concisely, but should contain all elements necessary to allow interpretation and replication of the results. As a guideline, Methods sections typically do not exceed 3,000 words. The Methods should be divided into subsections listing reagents and techniques. When citing previous methods, accurate references should be provided and any alterations should be noted. Information must be provided about: antibody dilutions, company names, catalogue numbers and clone numbers for monoclonal antibodies; sequences of RNAi and cDNA probes/primers or company names and catalogue numbers if reagents are commercial; cell line names, sources and information on cell line identity and authentication. Animal studies and experiments involving human subjects must be reported in detail, identifying the committees approving the protocols. For studies involving human subjects/samples, a statement must be included confirming that informed consent was obtained. Statistical analyses and information on the reproducibility of experimental results should be provided in a section titled "Statistics and Reproducibility".

All Nature Cell Biology manuscripts submitted on or after March 21 2016, must include a Data availability statement as a separate section after Methods but before references, under the heading "Data Availability". For Springer Nature policies on data availability see <http://www.nature.com/authors/policies/availability.html>; for more information on this particular policy see <http://www.nature.com/authors/policies/data/data-availability-statements-data-citations.pdf>. The Data availability statement should include:

- Accession codes for primary datasets (generated during the study under consideration and designated as "primary accessions") and secondary datasets (published datasets reanalysed during the study under consideration, designated as "referenced accessions"). For primary accessions data should be made public to coincide with publication of the manuscript. A list of data types for which submission to community-endorsed public repositories is mandated (including sequence, structure, microarray, deep sequencing data) can be found here <http://www.nature.com/authors/policies/availability.html#data>.
- Unique identifiers (accession codes, DOIs or other unique persistent identifier) and hyperlinks for datasets deposited in an approved repository, but for which data deposition is not mandated (see here for details <http://www.nature.com/sdata/data-policies/repositories>).
- At a minimum, please include a statement confirming that all relevant data are available from the authors, and/or are included with the manuscript (e.g. as source data or supplementary information), listing which data are included (e.g. by figure panels and data types) and mentioning any restrictions on availability.
- If a dataset has a Digital Object Identifier (DOI) as its unique identifier, we strongly encourage including this in the Reference list and citing the dataset in the Methods.

We recommend that you upload the step-by-step protocols used in this manuscript to the Protocol Exchange. More details can found at www.nature.com/protocolexchange/about.

DISPLAY ITEMS – main display items are limited to 6-8 main figures and/or main tables. For Supplementary Information see below.

FIGURES – Colour figure publication costs \$395 per colour figure. All panels of a multi-panel figure must be logically connected and arranged as they would appear in the final version. Unnecessary figures and figure panels should be avoided (e.g. data presented in small tables could be stated briefly in the text instead).

All imaging data should be accompanied by scale bars, which should be defined in the legend. Cropped images of gels/blots are acceptable, but need to be accompanied by size markers, and to retain visible background signal within the linear range (i.e. should not be saturated). The boundaries of panels with low background have to be demarked with black lines. Splicing of panels should only be considered if unavoidable, and must be clearly marked on the figure, and noted in the legend with a statement on whether the samples were obtained and processed simultaneously. Quantitative comparisons between samples on different gels/blots are discouraged; if this is unavoidable, it has to be performed for samples derived from the same experiment with gels/blots were processed in parallel, which needs to be stated in the legend.

Regardless of format, all figures must be vector graphic compatible files, not supplied in a flattened raster/bitmap graphics format, but should be fully editable, allowing us to highlight/copy/paste all text and move individual parts of the figures (i.e. arrows, lines, x and y axes, graphs, tick marks, scale bars etc). The only parts of the figure that should be in pixel raster/bitmap format are photographic images or 3D rendered graphics/complex technical illustrations.

Unprocessed scans of all key data generated through electrophoretic separation techniques need to be presented in a supplementary figure that should be labeled and numbered as the final supplementary figure, and should be mentioned in every relevant figure legend. This figure does not count towards the total number of figures and is the only figure that can be displayed over multiple pages, but should be provided as a single file, in PDF or TIFF format. Data in this figure can be displayed in a relatively informal style, but size markers and the figures panels corresponding to the presented data must be indicated.

The total number of Supplementary Figures (not including the “unprocessed scans” Supplementary Figure) should not exceed the number of main display items (figures and/or tables (see our Guide to Authors and March 2012 editorial <http://www.nature.com/ncb/authors/submit/index.html#suppinfo>; <http://www.nature.com/ncb/journal/v14/n3/index.html#ed>). No restrictions apply to Supplementary Tables or Videos, but we advise authors to be selective in including supplemental data.

GUIDELINES FOR EXPERIMENTAL AND STATISTICAL REPORTING

REPORTING REQUIREMENTS – To improve the quality of methods and statistics reporting in our papers we have recently revised the reporting checklist we introduced in 2013. We are now asking all life sciences authors to complete two items: an Editorial Policy Checklist (found here <https://www.nature.com/authors/policies/Policy.pdf>) that verifies compliance with all required editorial policies and a Reporting Summary (found here <https://www.nature.com/authors/policies/ReportingSummary.pdf>) that collects information on experimental design and reagents. These documents are available to referees to aid the evaluation of

the manuscript. Please note that these forms are dynamic 'smart pdfs' and must therefore be downloaded and completed in Adobe Reader. We will then flatten them for ease of use by the reviewers. If you would like to reference the guidance text as you complete the template, please access these flattened versions at <http://www.nature.com/authors/policies/availability.html>.

Author Rebuttal, second revision:

Dear Reviewers,

We wish to express our sincere appreciation for your thoughtful re-evaluation of our manuscript and provision of insightful feedback. Your comments have helped us significantly improve this work.

In direct response to the concerns initially raised, we have conducted additional experiments that we believe adequately address the issues raised. Specifically, 1) murine cell experiments now allow a more comprehensive assessment of conservation and limitations of our proposed mTORC2 regulation of cGAS localization across species; 2) employing multiple technical approaches and finer temporal analyses have provided deeper mechanistic insights into the relationship between mTORC2 activity and regulation of cGAS localization.

We believe the revisions strengthen our understanding of this previously unknown relationship between mTORC2 activity and cGAS chromatin association. Incorporating these new data points has strengthened our discussion of both conserved and divergent mechanisms. We kindly ask that you please reexamine our revised manuscript and detailed response. It is our hope that you will agree the changes now warrant publication in NCB journal.

Thank you again for your invaluable time and care in critically evaluating our findings - it is greatly helping to advance this field. Please let me know if any portion of our work remains unclear or requires further refinement. I appreciate you pushing us to strengthen our study.

Sincerely,
Lin Yuan

Reviewers' Comments:

Reviewer #1:

Remarks to the Author:

The reviewer thanks the authors for considering the criticism. The results reported in the revised manuscript conclusively demonstrate the involvement of mTORC2 in nucleo-cytoplasmic shuttling of cGAS and, together with the accompanying functional data, link cGAS localization to expression of a subset of genes in the chosen model systems. Conceptually, the manuscript suggests a new, previously unknown function of mTORC2 in control of cGAS localization and implicating the former in the control of cell cycle progression.

While the authors have made a tremendous effort in addressing the reviewers' criticism, few important mechanistic questions remain unanswered. Specifically, it is still not clear whether the observed PI3K sensitivity of cGAS localization is the direct effect of mTORC2 activity regulation or due to compounding effects. Even though the authors include a new dataset, showing reduced chromatin-associated cGAS 3-6 hours after PI3K inhibition (Fig. 1D, S1C), it still appears to be rather slow for a signaling kinase. Could the authors rule out secondary (e.g., transcriptionally controlled) effects? One option would be examining the time-course of pS37-, pS473- and pS6K phosphorylation in response to mTORC2 inhibition (e.g., by JR-AB2-011 or by mTOR inhibitors) and/or monitoring changes in nucleo-cytoplasmic distribution of cGAS-GFP fusions (e.g., GFP-cGASWT or -S213D) in STING^{-/-} cells using live cell microscopy. Further, from the provided data it is still unclear if S213D mutant remains chromatin-bound to the same extent as the WT cGAS upon PI3K inhibition or in mTORC2-depleted cells: I suggest the authors plot the corresponding BRET and ELISA data (e.g., 1D, S1C, S2BDEF, S12EF) with the WT and the mutant side-by-side (in the current presentation, BRET and ELISA data are separately normalized for the WT and S213D). The goal here is to 1) conclusively delineate cGAS nucleotidyltransferase activity from the effects of S37 phosphorylation and 2) mechanistically link mTORC2 activity to cGAS localization (genetic and functional links appear sufficient to me). If the authors cannot rule out compounding effects, this should be clearly communicated to the potential readers.

Response: We appreciate the thoughtful feedback to further elucidate the mechanistic underpinnings of our work. Addressing the open questions raised is paramount to validating our conclusions.

1) To conclusively delineate cGAS nucleotidyltransferase activity from the effects of S37 phosphorylation, we have now included side-by-side comparisons of BRET and ELISA data for both wild-type cGAS and S213D mutant in response to PI3K inhibition or mTORC2 depletion (see Figures 2H, S1C, S2F, S2G, and S3C). This clarifies phosphorylation regulation of chromatin binding independent of catalytic function.

2) To mechanistically link mTORC2 activity to cGAS localization, we have now conducted additional real-time experiments. Specifically, we employed western blotting, BRET, chromatin fractionation assays and live cell microscopy to examine time-dependent changes in cGAS chromatin association and phosphorylation upon mTORC2 inhibition at finer temporal resolution.

We treated *STING*^{-/-} HCT116 cells with the selective mTORC2 inhibitor JR-AB2-011 for a shortened time window. Using western blotting, we observed a dynamic decrease in the levels of pS37-cGAS and pS473-AKT1 in a time-dependent manner upon inhibition, while pT389-P70S6K and pS65-4E-BP1 levels remained unchanged (see Fig. S6E). Notably, after JR-AB2-011 treatment, pS473-AKT1 decreased significantly within 30 minutes but began increasing thereafter. In contrast, phosphorylation of our newly discovered mTORC2 substrate cGAS at Ser37 continued declining beyond 30 minutes. Potential explanations for the differing pS473-AKT1 and pS37-cGAS responses include: a) cGAS may be a more specific/sensitive substrate of mTORC2 than AKT1, such that residual mTORC2 activity after 30 min sustains some pAKT but is insufficient for cGAS phosphorylation; b) cGAS phosphorylation is unidirectional while AKT involves both kinases and phosphatases, allowing pS473-AKT1 levels to reach a new steady state over time with mTORC2 inhibition; c) compensatory mechanisms activated by loss of Akt signaling could indirectly phosphorylate AKT1 after 30 minutes, whereas no such pathway exists for cGAS-S37. Collectively, these data suggest cGAS is a more direct and sensitive target of mTORC2 signaling than AKT. Further investigation is needed to fully validate the underlying mechanisms.

We initially measured GFP-cGAS signal in live *STING*^{-/-} HCT116 cells treated with JR-AB2-011 using live cell fluorescence microscopy. However, the weak GFP-cGAS signal in living cells resulted in low signal-to-noise ratio, complicating detection of subtle localization changes. To obtain more quantitative data, we conducted the BRET and chromatin fractionation assays with greater precision (see Fig. S6, F and G). Quantification of chromatin-bound cGAS in JR-AB2-011 treated HCT116 cells revealed that cGAS redistribution from chromatin occurred within 3 hours (see Fig. S6, F and G). This time-resolved analysis helped argue for a more direct effect of mTOR inhibition on cGAS phosphorylation and chromatin association, while also demonstrating target engagement and ruling out potential off-target transcriptional effects with their slower kinetics. The delayed decrease in cGAS chromatin localization after acute mTORC2 inhibition could be due to: a) tight, multifaceted chromatin binding beyond Ser37 phosphorylation; b) multiple factors regulating anchoring; c) a gradient of cGAS phosphorylation states; d) kinetic differences in actively vs spontaneously bound populations. In summary, the multifactorial nature of chromatin associations and reversing a tightly-regulated localization mechanism pharmacologically likely underlies the observed delay.

Together, we have assembled strong evidence supporting a direct link between mTORC2 activity and cGAS localization: 1) Use lentiviral shRNAs and CRISPR/Cas9, we generated isogenic cell lines with specific mTORC2 mutations that impair kinase activity. Assessment of baseline cGAS localization without any drug treatment established that disruption of mTORC2 signaling is sufficient to alter cGAS chromatin association. 2) *In vitro* kinase assays using recombinant cGAS protein demonstrated mTORC2's ability to directly phosphorylate cGAS at S37 and modulate its localization to chromatin, where only the phosphorylated form binds. This definitively proved a direct molecular mechanism. 3) Real-time monitoring of cGAS

phosphorylation and localization upon mTORC2 inhibition. 4) Examination of relative chromatin dissociation kinetics in isogenic lines overexpressing wildtype or mutant cGAS further validated mTORC2's direct role upon inhibition. 5) Nuclear run-on assays revealed no transcriptional changes in cGAS upon PI3K/mTORC2 inhibition, corroborating a post-translational effect. 6) Clarifying the time-resolved phosphorylation dependence and subcellular fractionation kinetics of cGAS chromatin detachment upon acute mTORC2 disruption helped address the possibility of indirect effects over longer timescales. Collectively, these diverse approaches provide strong support for a direct post-translational link between mTORC2 activity and regulated cGAS chromatin association.

We believe that these additional experiments and the inclusion of comparative data address your concerns and provide further insights into the mechanistic relationship between mTORC2 activity, cGAS phosphorylation, and localization events. However, we want to acknowledge that complete ruling out of potential compounding effects may be challenging due to the complexity of cellular signaling networks. While our data strongly suggest a direct link between mTORC2 activity and cGAS localization, we cannot exclude the possibility of additional indirect effects. We have made sure to communicate this limitation clearly to the readers in the revised manuscript. We are grateful for your guidance in highlighting these important aspects, and we are confident that the revised manuscript adequately addresses your questions.

Thank you once again for your valuable feedback, which has significantly contributed to the improvement of our manuscript.

Secondly, existence of a second NLS, with a potentially distinct mTORC2- or PI3K/AGC kinase sensitivity, requires further investigation, especially if the authors claim centrality of mTORC2 in modifying chromatin through cGAS (lines 366-367). While the results on Fig. S4D conclusively demonstrate that S37 phosphorylation is sufficient for cGAS nuclear localization, then how does this work in mouse cGAS, with S37 missing, as indicated by Reviewer #3? If there's no mechanistic insight into the role of NLS2 at the moment, the authors should at least alert the readers about its potential relevance (and of the lack of evolutionary conservation of the N-terminal NLS1). Do the authors believe that the proposed mechanism of mTORC2-controlled cGAS-mediated chromatin modification is indeed a generic one? Which data, both published and generated by the authors, support this hypothesis? – this point should be properly reflected in the Discussion.

Response: The reviewer raises important questions about the species specificity of our proposed mTORC2-cGAS mechanism, and the potential involvement of alternative regulatory mechanisms such as additional nuclear localization signals (NLS). We appreciate the reviewer pushing us to more rigorously explore these uncertainties.

Our work has focused primarily on the human form of cGAS, which contains two known NLS sequences-²¹KASARNARGAPMDPTESPAAPEAALPKAGKF⁵¹ (NLS1) and ²⁹⁵DVIMKRKRGGG³⁰⁵ (NLS2). However, mouse cGAS only shares homology within NLS2, suggesting divergent mechanisms of nuclear localization may exist across species. Previous studies demonstrate the non-enzymatic N-terminal domain of human cGAS determines its nuclear localization (PMID: 30811988) and chromatin sensing function (PMID:30811988, 33542149, 34244470, 33051594). Although the integrity of NLS2 is required for hcGAS nuclear localization, NLS2 appears to function in importin-mediated translocation under DNA damage (PMID:30356214). Phosphorylation of cGAS by Akt kinase and CDK1-cyclin B complex at S305 in the catalytic core of the

NTase domain was shown to suppress its enzymatic activity but not affect localization (PMID:32351706, 26440888).

Previous studies have shown murine cGAS is predominantly nuclear, indicating the importance of NLS2 for murine cGAS localization. Our findings also indicated that the fundamental ccGAS-KGA pathway remains functionally conserved in regulating glutaminolysis and chemotherapy responses in murine colorectal cancer cells (Fig. S16, G to I). Due to lack of conservation in the N-terminal NLS between human and murine cGAS, we conducted additional experiments in the murine colorectal cancer MC38 cells to directly address species differences of mTORC2-cGAS signaling. Specifically, disrupting mTORC2 activity via RICTOR depletion in MC38 cells did not impact mouse cGAS chromatin localization, downstream KGA levels, glutaminolysis or chemotherapy responses. These new data, presented in new Supplementary Figures S16E-S16I, indicate the mTORC2-driven human cGAS Ser37 phosphorylation is not universally conserved across mammals.

We acknowledge mouse cGAS lacks the S37 residue found to be critical for human cGAS localization. To validate our proposed model more fully would require clarifying the role of mouse cGAS NLS2 and any alternative localization pathways. Our current mechanistic understanding of differential NLS utilization across species remains limited. However, the negative murine data are an important validation of the proposed human-specific paradigm. In our revised manuscript, we have emphasized that the mTORC2 regulation of cGAS appears to be species-specific rather than a generic model across mammals. We have incorporated the negative murine data to more accurately convey both conserved and divergent aspects of cGAS function identified to date. Please let us know if any aspect of our response requires additional clarification or experimentation based on the reviewers' feedback.

Overall, I am convinced of mTORC2 involvement in cGAS localization and functional effects in the chosen model system; regarding the mechanism and its relevance, the reported data are highly suggestive but not yet conclusive. Should the authors address the major points above, the report would be the first one demonstrating a novel function of mTORC2 in the direct control of gene expression, well worthy of the NCB readership.

The reviewer cannot comment on the details of interactome and animal studies (out of expertise).

Minor points:

1. Line 95: data on S1A do not appear to be relevant to the claim about DNA damage
2. Fig. 1A: definitions of fractions unclear from the legend (e.g., what's the difference between Cyt and Cyto?)
3. Fig. 1D: would be useful to include a panel with pS37 and the total cGAS in the whole cell lysate (WCL).
4. Fig. 1HJ: WCL panels with cGAS and loading controls missing.
5. Fig. S3A, S12D: one clearly does not NOT expect any BRET signal if only one partner of the BRET pair is expressed; as indicated above, BRET efficiency for the WT and S213D should be compared side-by-side.
6. Line 223: reference to Fig. 1D appears irrelevant to the claim
7. Line 233: wrong figure reference (should be S11D?)
8. Fig. S1C: meaning of error bars unclear
9. Fig. S12A: BRET pair not defined (it appears to be Sin1-Rluc8)
10. Fig. S16A: a pS37 panel would be very relevant.

Response: Thank you for your thorough review of our manuscript. We appreciate your attention to detail and have carefully addressed each of your minor points. Here are our responses and the specific solutions we have implemented:

1. We apologize for the confusion. You are correct that the data presented in Figure S1A are not directly relevant to the claim about DNA damage. We have removed the reference to DNA damage in line 95 to avoid any confusion.
2. We apologize for the lack of clarity in the legend of Figure 1A regarding the definitions of fractions. To address this, we have revised the legend to explicitly state that "Cyt" refers to the Cytoplasmic fraction, "Mem" refers to the Membrane fraction, "Sol Nuc" refers to the Soluble Nuclear fraction, "Chr" refers to the Chromatin fraction, and "Cyto" refers to the Cytoskeletal fraction. This clarification will provide a better understanding of the experimental setup and the specific cellular compartments analyzed.
3. We appreciate your suggestion to include a panel with pS37 and total cGAS in the whole cell lysate (WCL) in Figure 1D. We have added the requested panel to Figure 1D, allowing for a comprehensive analysis of pS37 phosphorylation and total cGAS levels in the whole cell lysate. This addition enhances the data presentation and strengthens the supporting evidence for our findings. Considering the timing of phosphorylation story we present the results in the Figure S2A and S6E.
4. We apologize for the omission of the whole cell lysate (WCL) panels with cGAS and loading controls in Figure 1HJ. We have addressed this oversight by including the WCL panels with cGAS and the corresponding loading controls in Figure S2D (for Fig.1H) and S4D (for Fig.1J). This addition ensures the completeness and accuracy of the experimental data.
5. We thank you for pointing out that BRET signals would not be expected if only one partner of the BRET pair is expressed. By measuring BRET only when both partners are expressed, we aimed to eliminate interference from cellular autofluorescence that could otherwise obscure interpretation of the results. To more directly assess any differences in BRET efficiency between variants,, we have included side-by-side comparisons of the BRET efficiency for wild-type (WT) and S213D mutant in the relevant figures (Figures 2H, S1C, S2F, S2G, and S3C). This comparison allows for a clear assessment of the BRET signals between the two variants.
6. We apologize for the incorrect reference to Figure 1D in line 223. We have corrected the reference to the appropriate figure in the revised manuscript, ensuring that the reference aligns with the claim.
7. Thank you for catching the error in the figure reference in line 233. The correct reference should be Figure S11D (new Figure S12E), and we have made the necessary correction in the revised manuscript.
8. We apologize for the unclear meaning of error bars in Figure S1C. To clarify, the error bars in Figure S1C represent the standard error of the mean (SEM) of the average values across three independent experiments. We have revised the figure legend to explicitly state the meaning of the error bars, providing a clear understanding of the data variability.
9. We appreciate your observation regarding the undefined BRET pair in Figure S12A. It is H2A-Rluc8, and we have included this information in the revised figure legend to ensure clarity and accuracy.

10. We agree that a panel showing pS37 would be relevant in Figure S16A. We have added the requested pS37 panel to Figure S16A, providing additional important information for the readers.

We sincerely thank you for bringing these minor points to our attention and providing specific suggestions for improvement. We believe that the revisions we have made address your comments and enhance the clarity, accuracy, and completeness of the manuscript.

Thank you once again for your valuable feedback, which has significantly contributed to the improvement of our work.

Reviewer #2:

Remarks to the Author:

The revised manuscript answered my questions and showed convincing results. I recommend the publication of "mTORC2-driven chromatin cGAS mediates chemoresistance in colorectal cancer" in NCB.

Response: We sincerely appreciate your positive feedback on the revised manuscript. We are delighted that our responses to your questions and the inclusion of additional data have addressed your concerns and provided convincing results. Once again, we sincerely thank you for your support and recommendation.

Reviewer #3:

Remarks to the Author:

The revision has been significantly improved, but one concern remains and should be correctly addressed: the negative controls of mTORC2-guided nuclear translocation of cGAS and mTORC2-driven KGA expression, glutaminolysis, and chemoresistance in murine cells to validate the specificity of this action among mammalian species. If the data is negated, state it clearly in the abstract/manuscript.

Response: Thank you for this insightful feedback. Evaluating the specificity of our proposed mechanism across species is paramount. To directly address the reviewer's concern, we conducted additional experiments exploring the role of mTORC2 in regulating cGAS and KGA in murine cells.

Specifically, we assessed cGAS subcellular localization (Fig. S16E), KGA expression levels (Fig. S16, F and G), glutaminolysis (Fig. S16H), and chemoresistance (Fig. S16I) in the murine colorectal cancer cell line MC38 upon RICTOR knockdown. Intriguingly, disrupting mTORC2 signaling did not impact any of the aforementioned outcomes in MC38 cells (Fig. S16, E to I).

We have included this important negative data from the murine model as a new Supplementary Figure S16. Furthermore, we have updated the Abstract and manuscript to state unequivocally that while our proposed mTORC2-dependent mechanism appears conserved in specific human cell types, ccGAS-KGA pathway is functionally conserved in human and murine cells. We have now revised the manuscript: "While murine cGAS is predominantly nuclear, mTORC2 signaling contributes minimally to cGAS chromatin localization in mice (Fig. S16, E-I). This divergence likely arises from a lack of conservation in the N-terminal NLS between human and murine cGAS. Remarkably, our findings indicate that the fundamental

ccGAS-KGA pathway remains functionally conserved in regulating glutaminolysis and chemotherapy responses in murine colorectal cancer cells (Fig. S16, G to I)."

Clearly presenting species-specific distinctions is fundamental for accurately conveying the relevance and potential generalizability of our findings. We hope integrating these controls examining mTORC2 function across mammalian systems adequately resolves the reviewer's valid concern. Please let me know if any aspect of our response would benefit from further clarification or experimental refinement. Addressing feedback critically is improving our work, and we appreciate the reviewer pushing us to do so.

Decision Letter, third revision:

Our ref: NCB-A51988C

8th May 2024

Dear Dr. Yuan,

Thank you for submitting your revised manuscript "mTORC2-driven chromatin cGAS mediates chemoresistance in colorectal cancer" (NCB-A51988C). It has now been seen by the original referees and their comments are below. The reviewers find that the paper has improved in revision, and therefore we'll be happy in principle to publish it in Nature Cell Biology, pending minor revisions to satisfy the referees' final requests and to comply with our editorial and formatting guidelines.

Thank you again for your interest in Nature Cell Biology Please do not hesitate to contact me if you have any questions.

Sincerely,

Zhe Wang, PhD
Senior Editor
Nature Cell Biology

Tel: +44 (0) 207 843 4924
email: zhe.wang@nature.com

Reviewer #1 (Remarks to the Author):

The authors have conclusively addressed my concerns regarding 1) delineation of the cGAS NT activity

from the effects of S37 phosphorylation by mTORC2 (S1C, S2F, S2G, S3C) and 2) mechanistic link of mTORC2 activity and cGAS localization (S6E-G). In the rebuttal letter, the authors also expressed necessary caution in interpreting the observed effects. I also fully accept that monitoring changes of weak GFP signal over time could be tough; population measurements, like BRET or ChIP, would provide a more robust signal, indeed. Thank you for your careful reaction to the reviewer's comments! While the authors indicate that they alert the readers that they cannot exclude additional factors linking PI3K-mTORC2 to cGAS localization (Rebuttal letter, p. 4, par. 2), I could not identify this acknowledgement in the text.

I believe that my concerns have been sufficiently addressed by the authors. The text might require some minor editing, but I believe my feedback is no longer necessary.

I wish the authors all the best with their publication!

Minor:

- 1) While the authors claim that Fig. 1D includes p37 and cGAS in WCL (Rebuttal letter, p. 7, par. 3), I could not find the additional panel in the new version of the manuscript – must be some confusion. Fig. S2A does not include time-dependent changes (S6E does, though). I am sure this can be easily corrected and controlled by the editor.
- 2) As of now, the title appears a bit awkward; consider “mTORC2-driven cGAS association with chromatin...” or “mTORC2-dependent cGAS binding to chromatin...” – this is up to the authors and copyrighters, naturally.
- 3) Line 59: “...recombination (ref. 7). cGAS also...”
- 4) Line 96: awkward phrase (“soluble nuclei”); consider “...we found cGAS to be predominantly chromatin-bound”
- 5) Lines 120-121: consider adding conclusion, e.g. “...was specifically controlled by Rictor, but not Raptor knockdown (Fig. S2FG), indicating that cGAS chromatin association is dependent on mTORC2, but not nucleotidyltransferase activity of cGAS” – just a suggestion.
- 6) Lines 135-136: consider “...HCT116 cells and rescued by Rictor (Fig. ...)”
- 7) Line 345: awkward phrase (“pathologically and molecularly”)
- 8) Lines 383-385: awkward phrase

Reviewer #3 (Remarks to the Author):

Questions raised during initial reviewing processes have been reasonably addressed.

Author Rebuttal, third revision:

Reviewer #1:

Remarks to the Author:

The authors have conclusively addressed my concerns regarding 1) delineation of the cGAS NT activity from the effects of S37 phosphorylation by mTORC2 (S1C, S2F, S2G, S3C) and 2) mechanistic link of mTORC2 activity and cGAS localization (S6E-G). In the rebuttal letter, the authors also expressed necessary caution in interpreting the observed effects. I also fully accept that monitoring changes of weak GFP signal over time could be tough; population measurements, like BRET or CHIP, would provide a more robust signal, indeed. Thank you for your careful reaction to the reviewer's comments!

While the authors indicate that they alert the readers that they cannot exclude additional factors linking PI3K-mTORC2 to cGAS localization (Rebuttal letter, p. 4, par. 2), I could not identify this acknowledgement in the text.

Response: Thank you for the feedback. We appreciate you taking the time to thoroughly evaluate our work and provide constructive comments to strengthen the manuscript. We are glad to hear you find that our responses adequately addressed your original technical concerns regarding delineating the activities of cGAS's N-terminal domain and S37 phosphorylation by mTORC2. You correctly point out the challenges of monitoring weak GFP signals over time and our use of more robust population measurements like BRET and CHIP.

You also fairly note that while we acknowledged the potential involvement of additional factors linking PI3K-mTORC2 to cGAS localization in our rebuttal letter, this caveat was not explicitly stated in the text itself. We have now included the following statement in the lines 319-321 in *Discussion* to directly alert readers: "While our findings demonstrate mTORC2-mediated S37 phosphorylation promotes cGAS chromatin localization, we cannot exclude contributions of additional factors downstream of PI3K-mTORC2 signaling." Thank you again for your thoughtful critique.

I believe that my concerns have been sufficiently addressed by the authors. The text might require some minor editing, but I believe my feedback is no longer necessary.

I wish the authors all the best with their publication!

Minor:

1) While the authors claim that Fig. 1D includes p37 and cGAS in WCL (Rebuttal letter, p. 7, par. 3), I could not find the additional panel in the new version of the manuscript – must be some confusion. Fig. S2A does not include time-dependent changes (S6E does, though). I am sure this can be easily corrected and controlled by the editor.

Response: Thank you for raising these important points and pushing us to improve our work. You've helped identify areas where our previous responses lacked clarity. To resolve any confusion, we have now conducted new experiments analyzing total cGAS and pS37 levels in whole cell lysates over time with GDC-0941 treatment. As shown in Extended Data Fig. 3e, total cGAS remained unchanged while pS37 decreased significantly with longer exposure. We appreciate you taking the time for a careful, methodical review that thoroughly checked our responses against the data.

2) As of now, the title appears a bit awkward; consider “mTORC2-driven cGAS association with chromatin...” or “mTORC2-dependent cGAS binding to chromatin...” – this is up to the authors and copywriters, naturally.

Response: Thank you for the suggestion regarding our manuscript title. We appreciate the reviewer taking time to thoughtfully consider how to present our work's key findings in the clearest light. According to the editor's suggestion, we agree the revised title “mTORC2-driven chromatin cGAS mediates chemoresistance through epigenetic reprogramming in colorectal cancer” better presents our key findings.

3) Line 59: “...recombination (ref. 7). cGAS also...”

Response: Thank you for the feedback. We have fully revised the specified lines as suggested. Details of the revisions can be found on line 58.

4) Line 96: awkward phrase (“soluble nuclei”); consider “...we found cGAS to be predominantly chromatin-bound”

Response: Thank you for the feedback. We have fully revised the specified lines as suggested. Details of the revisions can be found on line 90.

5) Lines 120-121: consider adding conclusion, e.g. “...was specifically controlled by Rictor, but not Raptor knockdown (Fig. S2FG), indicating that cGAS chromatin association is dependent on mTORC2, but not nucleotidyltransferase activity of cGAS” – just a suggestion.

Response: Thank you for the feedback. We have fully revised the manuscript as suggested. Details of the revisions can be found on lines 110-111.

6) Lines 135-136: consider “...HCT116 cells and rescued by Rictor (Fig. ...)”

Response: Thank you for the feedback. We have fully revised the manuscript as suggested. Details of the revisions can be found on lines 121-122.

7) Line 345: awkward phrase (“pathologically and molecularly”)

Response: Thank you for your comments. We appreciate you noting the awkward phrasing. We have revised it as you recommended to simply say the samples were “pathologically” on line 284.

8) Lines 383-385: awkward phrase

Response: Thank you for your comments. Upon reflection, we have now removed lines 383-385 to enhance clarity and flow.

Finally, thank you again for taking the time to carefully review our manuscript and provide these thoughtful comments and suggestions for improvement. We believe your thoughtful feedback has helped strengthen our

work, and we hope this revised version now addresses all of your original comments and concerns. We appreciate you taking the time to ensure our work is as robust as possible.

Reviewer #3:

Remarks to the Author:

Questions raised during initial reviewing processes have been reasonably addressed.

Response: We sincerely appreciate your positive feedback on the revised manuscript. We are delighted that our responses to your questions and the inclusion of additional data have addressed your concerns and provided convincing results. Once again, we sincerely thank you for your support and recommendation.

Final Decision Letter:

Dear Dr Yuan,

I am pleased to inform you that your manuscript, "mTORC2-driven chromatin cGAS mediates chemoresistance through epigenetic reprogramming in colorectal cancer", has now been accepted for publication in Nature Cell Biology.

After the grant of rights is completed, you will receive a link to your electronic proof via email with a request to make any corrections within 48 hours. If, when you receive your proof, you cannot meet

this deadline, please inform us at rjsproduction@springernature.com immediately.

Please note that *Nature Cell Biology* is a Transformative Journal (TJ). Authors may publish their research with us through the traditional subscription access route or make their paper immediately open access through payment of an article-processing charge (APC). Authors will not be required to make a final decision about access to their article until it has been accepted. Find out more about Transformative Journals

Authors may need to take specific actions to achieve compliance with funder and institutional open access mandates. If your research is supported by a funder that requires immediate open access (e.g. according to Plan S principles) then you should select the gold OA route, and we will direct you to the compliant route where possible. For authors selecting the subscription publication route, the journal's standard licensing terms will need to be accepted, including self-archiving policies. Those licensing terms will supersede any other terms that the author or any third party may assert apply to any version of the manuscript.

If your paper includes color figures, please be aware that in order to help cover some of the additional

cost of four-color reproduction, Nature Portfolio charges our authors a fee for the printing of their color figures. Please contact our offices for exact pricing and details.

If you have not already done so, we strongly recommend that you upload the step-by-step protocols used in this manuscript to protocols.io (<https://protocols.io>), an open online resource that allows researchers to share their detailed experimental know-how. All uploaded protocols are made freely available and are assigned DOIs for ease of citation. Protocols and Nature Portfolio journal papers in which they are used can be linked to one another, and this link is clearly and prominently visible in the online versions of both. Authors who performed the specific experiments can act as primary authors for the Protocol as they will be best placed to share the methodology details, but the Corresponding Author of the present research paper should be included as one of the authors. By uploading your Protocols onto protocols.io, you are enabling researchers to more readily reproduce or adapt the methodology you use, as well as increasing the visibility of your protocols and papers. You can also establish a dedicated workspace to collect your lab Protocols. Further information can be found at <https://www.protocols.io/help/publish-articles>.

With kind regards,

Zhe Wang, PhD
Senior Editor
Nature Cell Biology

Tel: +44 (0) 207 843 4924
email: zhe.wang@nature.com
